# Programmable melanoma-targeted radio-immunotherapy via fusogenic liposomes functionalized with multivariate-gated aptamer assemblies

Xijiao Ren[1,7], Rui Xue[2,7], Yan Luo[3,7], Shuang Wang[2], Xinyue Ge[2], Xuemei Yao[2], Liqi Li[4], Junxia Min [5], Menghuan Li [2] ✉, Zhong Luo [1,2] ✉ & Fudi Wang [5,6] ✉

Radio-immunotherapy exploits the immunostimulatory features of ionizing radiation (IR) to enhance antitumor effects and offers emerging opportunities for treating invasive tumor indications such as melanoma. However, insufficient dose deposition and immunosuppressive microenvironment (TME) of solid tumors limit its efficacy. Here we report a programmable sequential therapeutic strategy based on multifunctional fusogenic liposomes (Lip@AUR-ACP-aptPD-L1) to overcome the intrinsic radio-immunotherapeutic resistance of solid tumors. Specifically, fusogenic liposomes are loaded with gold-containing Auranofin (AUR) and inserted with multivariate-gated aptamer assemblies (ACP) and PD-L1 aptamers in the lipid membrane, potentiating melanoma-targeted AUR delivery while transferring ACP onto cell surface through selective membrane fusion. AUR amplifies IR-induced immunogenic death of melanoma cells to release antigens and damage-associated molecular patterns such as adenosine triphosphate (ATP) for triggering adaptive antitumor immunity. AUR-sensitized radiotherapy also upregulates matrix metalloproteinase-2 (MMP-2) expression that combined with released ATP to activate ACP through an "and" logic operation-like process (AND-gate), thus triggering the in-situ release of engineered cytosine-phosphate-guanine aptamer-based immunoadjuvants (eCpG) for stimulating dendritic cell-mediated T cell priming. Furthermore, AUR inhibits tumor-intrinsic vascular endothelial growth factor signaling to suppress infiltration of immunosuppressive cells for fostering an anti-tumorigenic TME. This study offers an approach for solid tumor treatment in the clinics.

Radiotherapy (RT) is an antitumor modality that employs high-energy X ray or subatomic particles to destroy tumor cells, which is commonly used for the treatment of a variety of solid tumor indications due to its good cost-effectiveness, high treatment compliance, and curative/palliative benefit[1–3]. Recent studies reveal that radiotherapy also has the potential to substantially modify the tumor ecosystem to exert multifaceted immunostimulatory effects including induction of immunogenic tumor cell death, tumor-associated antigen presentation, and activation of tumor-specific effector T cells, thus offering potential synergy with various immunotherapeutic modality for

enhanced antitumor efficacy[3–6]. Indeed, clinical insights collectively confirm that combining radiotherapy with immunotherapy could convey significant improvement on the overall survival benefit of melanoma patients without inducing obvious side effects. For instance, preconditioning tumors with IR (peri-induction radiotherapy) could activate the immune system and facilitate the recognition and elimination of tumor cells by the sequentially administered immunotherapeutic modalities. On the other hand, there are reports that the local IR treatment of tumors following immunotherapy (post-escape radiotherapy) could potentially remodel the transmission electron microscopic (TEM) to reverse immunoresistance and prevent tumor immune escape[7–9]. Overall, these emerging radio-immunotherapies have demonstrated unique advantages compared with conventional antitumor therapies including systemic antitumor effects and long-lasting antitumor immune memory, which are highly favorable for treating invasive and refractory solid tumor indications such as melanoma[10–12]. However, solid tumors possess multiple intrinsic traits that may undermine the efficacy of radio-immunotherapy[13–15]. Typically, the actual deposition of ionizing radiation (IR) in tumor tissues is usually insufficient, which requires dangerously high IR doses to achieve significant tumor inhibition effects and thus elevates the RT-associated side effects[16–19]. Furthermore, the immunosuppressive TME will substantially impair the T cell-mediated antitumor immunity despite the IR-triggered immunostimulatory effects[20–22]. Therefore, new treatment strategies with cooperative radiosensitization and anti-tumorigenic TME immunomodulatory capabilities are urgently needed to overcome these challenges, which hold promise to augment the therapeutic potency of radio-immunotherapy for robust and persistent tumor inhibition.

The excessive presence of immunosuppressive cell populations such as myeloid-derived suppressor cells (MDSCs) and regulatory T cells (Tregs) in TME is a major driver of tumor immune escape[23,24]. Notably, tumor cells frequently express abundant VEGF to recruit MDSCs and Tregs to TME as well as stimulating their proliferation thereafter, which is recognized as a crucial promoter of tumor immunoresistance and a potential target for clinical exploitation[25–27]. Auranofin (AUR) is a gold coordination compound that has been long approved by FDA for treating rheumatoid arthritis in the clinics. Interestingly, it has demonstrated multiple therapeutically favorable bioactivities in recent studies and been increasingly repurposed for tumor treatment[28–30]. Recent studies reveal that AUR could abolish VEGF-dependent pro-tumorigenic immunosignaling pathways through inhibiting ERK1/2-HIF-1α axis in tumor cells for enhancing the tumor-infiltration and cytotoxic potential of antitumor T cells[28,31–34]. Moreover, due to the complexation with high-Z gold (I) species, AUR treatment could significantly enhance IR deposition in tumor cells for effective radiosensitization[35–38]. Therefore, tumor-targeted AUR treatment could be a promising strategy for boosting radio-immunotherapy efficacy in the clinical context.

Aptamer is a class of synthetic oligonucleotide ligands with antibody-like binding behavior with designated molecular targets[39–41], which has attracted broad interest for therapeutic applications due to the high binding affinity/specificity and may fulfill a variety of functional roles including signaling mediators and targeting ligands, which are particularly favorable in the field of antitumor immunotherapeutics[42–46]. For example, CpG ODN (CpG oligonucleotide) is a clinically tested aptamer-based immune adjuvant that can promote DC activation via triggering toll-like receptor 9 (TLR9) immune signaling to stimulate the downstream adaptive immune reactions[47–49]. Alternatively, there is abundance evidence that PD-L1-targeting aptamers could bind with PD-L1-overexpressing tumor cells for efficient PD-L1 antagonization[33,50,51]. Notably, the versatile aptamer chemistry allows the further modular integration of multiple chemically-tailored aptamer units to introduce logic-gate bioresponsive reactivity without altering their original biological

functions[52–54]. It is thus anticipated that implementing programmable aptamer assemblies into therapeutic systems could be a practical approach for regulating their biointeractions and potentiating cooperative therapeutic combinations. Indeed, there are already reports that aptamer-based logic-gated nanosystems could convey programmable diagnostic or therapeutic activities, which may substantially improve their controllability and precision in vitro or in vivo[39,55–57].

In this work, we report a multivariate-gated aptamer assembly-modified AUR-loaded fusogenic liposome for boosting melanoma-targeted radio-immunotherapy. 5′ end of commercial CpGs are conjugated with a DNA sequence that can bind to the 5′ end region of ATP-binding aptamer (aptATP) to afford eCpG. Meanwhile, MMP-2-degradable peptide nucleic acid (PNA) sequences that can bind to 3′ end region of aptATP are synthesized and complexed to aptATP-eCpG to form duplex assemblies (ACP). The 3′ ends of aptATP and PD-L1-binding aptamer (aptPD-L1) are modified with cholesterol for insertion into the lipid bilayers of fusogenic liposomes, while AUR is pre-dissolved into lipid precursors before liposome formation. The fusogenic liposomes (Lip@AUR-ACP-aptPD-L1) are prepared through a simple film-hydration method. Lip@AUR-ACP-aptPD-L1 can bind with PD-L1-overexpressing melanoma cells and fuse with the cytoplasmic membrane, which not only anchors the ACP assemblies onto melanoma cell surface but also enables targeted AUR delivery. ACP operates as an AND-gate in biological environment, which shows negligible responsiveness to separate ATP or MMP-2 stimulation and can only be dissociated when both ATP and MMP-2 are at a high level. AUR contents substantially enhance the IR dose accumulation in melanoma cells and induces efficient immunogenic cell death (ICD), releasing abundant tumor-derived antigens and damage-associated molecular patterns (DAMPs) such as ATP into TME while also inducing MMP-2 upregulation, thus creating input signals for triggering eCpG release from ACP and enhancing DC-mediated cross-priming of antitumor T cells. In addition, AUR blocks ERK1/2-HIF-1α-VEGF axis in tumor cells to inhibit tumor-infiltrating immunosuppressive cells. These effects act cooperatively to abolish melanoma growth and prevent its metastasis or recurrence (Fig. 1). This work presents a programmable sequential strategy to enhance the radio-immunotherapeutic efficacy against invasive melanomas.

## Results
### Multivariate-gated activation of aptamer assembly
The multivariate-gated activation mode of the ACP assembly is an essential perquisite for enhancing the radio-immunotherapeutic efficacy of the liposomal nanoformulation, which is crucial for enabling optimal immunostimulation in post-IR melanomas with spatial-temporal precision while minimizing the potential side effects. To obtain the bioresponsive multi-component aptamer assemblies, we first synthesized eCpG, aptATP, PmP, and aptPD-L1 via established procedures as the basic components, of which the complementary binding affinity between aptATP/eCpG and aptATP/PmP pairs provided the mechanistic basis for assembly formation (Fig. 2a). It is important to note that the primary ATP binding sequence in aptATP was simultaneously occupied by the GGAGTATTGC segments in the 5′ end of eCpG and the AGGAA-GG-TAAGA segments located near the MMP-2-cleavable peptidic chain in PNA[39,55]. Notably, to avoid the potential negative impact of cholesterol modification on the structural and biochemical features of aptATP and aptPD-L1 aptamers, multiple base T units were added at the 3′ end of the aptamer sequences as a functional handle. NUPACK simulation of secondary structures of these engineered aptamers showed no changes in the structure and △G of the aptamers (Supplementary Fig. 1), confirming successful aptamer modification without altering their designated biological functions. To ensure effective eCpG detachment from aptATP/eCpG complexes under ATP competition, we proactively constructed aptamer assemblies with different aptATP/eCpG ratios and tested their

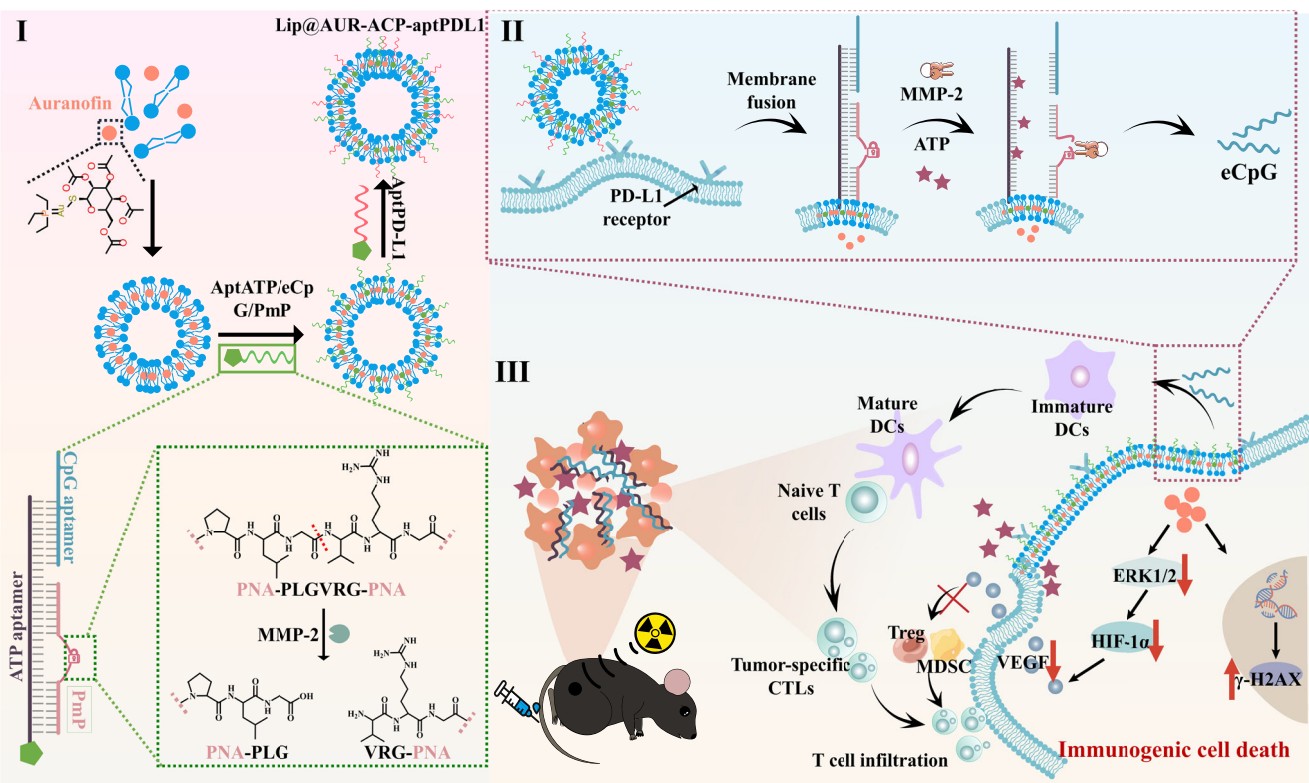

**Fig. 1 | Schematic illustration of Lip@AUR-ACP-aptPD-L1 construction and its radio-immunotherapeutic effect.** (I) Schematic depiction of the assembly process of ACP and construction of Lip@AUR-ACP-aptPD-L1. As the primary ATP binding sequence in aptATP was simultaneously occupied by the GGAGTATTGC segments in the 5′ end of eCpG and the AGGAA-GG-TAAGA segments located near the MMP-2-cleavable peptidic chain in PNA, the ACP assemblies have high stability in physiological environment with negligible eCpG leakage. (II) Schematic representation of the AND-gate release of eCpG from ACP assembly in Lip@AUR-ACP-aptPD-L1 in the context of IR treatment. (III) Lip@AUR-ACP-aptPD-L1 mediates sequential radiosensitization of melanoma cells and anti-tumorigenic remodeling of tumor immune microenvironment, potentiating enhanced radio-immunotherapeutic efficacy.

responsiveness to ATP treatment. Comparative PAGE analysis under graded ATP concentrations showed that aptamer assemblies at the aptATP/eCpG ratio of 2:1 presented enhanced sensitivity to ATP competition to trigger efficient eCpG release, which was used as the standard condition for subsequent experiment (Fig. 2b). Although DNA-PAGE analysis could intuitively illustrate the changes in aptATP/eCpG binding state, it is unable to provide quantitative data for objective analysis. Hence, we also carried out quantitative fluorescence analysis for the samples, revealing that aptATP has a complexation efficiency of 97.07% with eCpG (Supplementary Fig. 4c), ascribing to the molecular specificity of the complementary sequences thereof. The ATP-responsiveness of aptATP/eCpG complex was further profiled by PAGE assay, which showed that treating aptATP/eCpG complexes with an ATP concentration of 0.05 μM was sufficient to induce significant eCpG release (Fig. 2c), emphasizing the necessity for the implementation of the AND-gate eCpG release function to avoid premature eCpG leakage at background concentrations. Next, the aptATP/eCpG complexes were sequentially integrated with PNA at an aptATP: PNA ratio of 1:1.5, leading to the formation of duplex structures (ACP) with robust stability under physiological conditions. Similarly, quantitative fluorescence analysis showed that the assembly efficiency of ACP with aptATP, eCpG, and PNA was 95.52% (Supplementary Fig. 4c), indicating that the assembly process is highly modular with molecular precision. According to Fig. 2d, the eCpG release from ACP assembly under sole 200 nM ATP treatment was almost negligible (orange frame); similarly, treating ACP with only 10 nM MMP-2 also failed to induce obvious eCpG release (blue frame). It is worth noting that eCpG was potently released from ACP under the combined treatment of 200 nM ATP and 10 nM MMP-2 (Fig. 2d red

frame). Comparative analysis on eCpG release profiles immediately suggested that PNA complexation inhibited the ATP recognition and binding capability of aptATP and validated the multivariate-gated eCpG release behavior, which is beneficial for minimizing accidental ACP activation in response to tissue-intrinsic ATP stimulation at background levels.

## Construction and characterization of the fusogenic liposomes

Liposomes are a well-tested pharmaceutical technology with easy production and high cost-effectiveness, which have already been used to formulate a myriad of therapeutic substances in the clinics for enhanced delivery. Here the liposomal nanosubstrates were synthesized through the self-assembly of DMPC, DSPE-PEG$_{2000}$, DOTAP, and AUR, thus endowing cytoplasm membrane fusion and long-circulating stability while also achieving spontaneous AUR loading. Due to the proactive modification of cholesterol on the 3′ position of aptATP and aptPD-L1, the multivariate-gated ACP assembly and tumor-targeting aptPD-L1 could be facilely inserted into the lipid bilayers for non-invasive modification to form Lip@AUR-ACP-aptPD-L1 (Fig. 2a). To provide an intuitive demonstration of the nanoscale morphology of the liposomal products, the Lip@AUR-ACP-aptPD-L1 samples were observed by TEM imaging analysis and the results indicated that the liposomes have uniform spherical morphology and high monodispersity (Fig. 2e). However, TEM imaging only showed the liposome morphology under dried conditions. To characterize the size distribution of the liposomes under biomimetic solution environment, the liposomes were further dispersed in biomimetic buffers for DLS analysis. The imaging results were consistently supported by the quantitative DLS analysis, revealing an average diameter of around

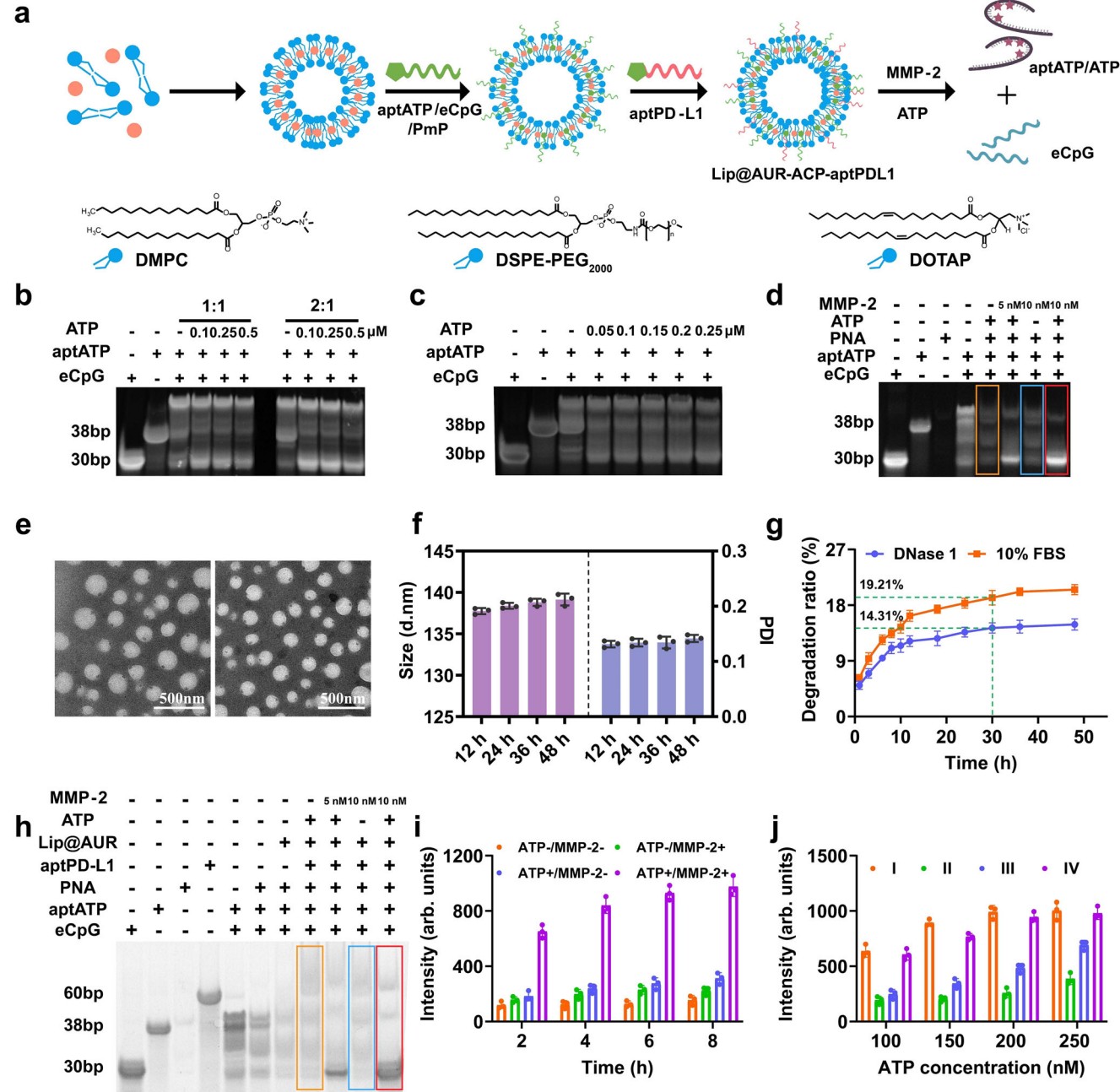

**Fig. 2 | Physicochemical characterization of the fusogenic liposomes.**
**a** Preparation process and lipid composition of Lip@AUR-ACP-aptPD-L1. **b** DNA-PAGE analysis regarding eCpG release from aptATP/eCpG complex in response to different ATP concentrations (*n* = 3 experimental replicates). **c** Impact of competitive ATP binding on aptATP/eCpG complex via DNA-PAGE analysis (aptATP:eCpG = 2:1) (*n* = 3 experimental replicates). **d** DNA-PAGE analysis regarding eCpG release from the ACP assembly (aptATP: eCpG: PNA = 2:1:3) with 200 nM ATP and 5 nM or 10 nM MMP-2 (*n* = 3 experimental replicates). The yellow box under 200 nM ATP, the blue box under 10 nM MMP-2, and the red box under 200 nM ATP and 10 nM MMP-2. **e** TEM results of Lip@AUR-ACP-aptPD-L1 stained with 4% phosphotungstic acid (*n* = 3 experimental replicates). **f** The stability of Lip@AUR-ACP-aptPD-L1 in pH7.4 PBS buffer at 12–48 h by DLS analysis. The purple color represents the size and the blue color represents the polydispersity index (PDI).

**g** Degradation assessment of Lip@AUR-AC^Cy5^P-aptPD-L1 in DNase 1 or 10% FBS through 50 h incubation. **h** DNA-PAGE analysis of eCpG release from Lip@AUR-AC^Cy5^P-aptPD-L1 with 200 nM ATP and 5 nM or 10 nM MMP-2 (*n* = 3 experimental replicates). The yellow box under 200 nM ATP, the blue box under 10 nM MMP-2 and the red box under 200 nM ATP and 10 nM MMP-2. **i** Fluorescence analysis of eCpG^Cy5^ release from Lip@AUR-ACP-aptPD-L1 with or without 200 nM ATP/10 nM MMP-2 stimulus input. **j** Fluorescence analysis of eCpG^Cy5^ release from different liposome formulations under different ATP concentrations. I: Lip@AUR-AC^Cy5^-aptPD-L1, II: Lip@AUR-AC^Cy5^P-aptPD-L1, III: Lip@AUR-AC^Cy5^P-aptPD-L1 + 5 nM MMP-2, IV: Lip@AUR-AC^Cy5^P-aptPD-L1 + 10 nM MMP-2. Data are presented as mean values ± SEM (*n* = 3 experimental replicates for (**f**, **g**) and (**i**, **j**)). Source data are provided as a Source Data file.

130 nm for the final liposome products (Supplementary Table 4 and Supplementary Fig. 2a) as well as a polydispersity index of around 0.12 (Supplementary Table 4 and Supplementary Fig. 2b), confirming the morphological homogeneity of the liposomes thereof. Zeta potential

analysis showed that pristine Lip had an average surface charge of around 38.16 mV due to the positively charged status of DOTAP contents (Supplementary Table 4 and Supplementary Fig. 2c), while the zeta potential of Lip@AUR-ACP-aptPD-L1 dropped significantly to

−10.71 mV, supporting the successful immobilization of the negatively-charged aptamers. We also found that the Lip@AUR-ACP-aptPD-L1 nanoformulation presented good loading capacity for the therapeutic contents. Here the AUR contents in the liposomal systems were measured using both ICP and fluorescence spectroscopic analysis. From a technical perspective, ICP has higher limit of detection (LOD) but superior interference control, while fluorescence spectroscopy has lower LOD but is more susceptible to background noises in complex samples. Therefore, the two techniques were combined to accurately profile the AUR loading in the liposomal system. Specifically, ICP and quantitative fluorescence analysis showed that the AUR could be efficiently loaded into the fusogenic liposomes at a high efficiency of around 88.29%, and the relative AUR ratio in the final Lip@AUR-ACP-aptPD-L1 was around 4.98% (Supplementary Table 4, Supplementary Fig. 2d, e and Supplementary Fig. 3). Due to the presence of the cholesterol tails, the liposomal integration of ACP assembly and aptPD-L1 was highly efficient with a loading efficiency of 86.5% and 81.0%, respectively, while the average number of ACP assembly and aptPD-L1 on a single liposome was 109 and 51 based on fluorescence spectroscopy (Supplementary Fig. 5). DLS analysis also revealed that Lip@AUR-ACP-aptPD-L1 has favorable stability in aqueous solution with no noticeable size changes after incubation for 2 days (Fig. 2f). Alternatively, quantitative fluorescence analysis showed that the degradation rate of ACP in Lip@AUR-ACP-aptPD-L1 was only 19.21% or 14.31% after incubation in 10% FBS or DNase 1 for 30 h (Fig. 2g). These data confirmed the relative stability of Lip@AUR-ACP-aptPD-L1 in biomimetic buffers, which is beneficial for improving the therapeutic index of AUR and ACP after systemic administration.

## Multivariate AND-gate operation of Lip@AUR-ACP-aptPD-L1

To test if the multivariate AND-gate operation of the ACP assembly was maintained after the integration into Lip@AUR-ACP-aptPD-L1, the liposomes were processed with different stimulus inputs and then the corresponding release efficiency of eCpG was monitored by DNA-PAGE, and fluorescence spectroscopic analysis, which allowed the accurate profiling of the eCpG release behavior under different conditions. Consistent with the observations of vehicle-free ACP assemblies in Fig. 2d, the combinational treatment of 200 nM ATP and 10 nM MMP-2 induced efficient eCpG release from Lip@AUR-ACP-aptPD-L1 according to the DNA-PAGE analysis (Fig. 2h), which was also supported by the results of the quantitative fluorescence spectroscopic analysis that eCpG release rate from Lip@AUR-AC$^{Cy5}$P-aptPD-L1 reached around 92.48% after 2 h incubation with 200 nM ATP and 10 nM MMP-2 (Supplementary Fig. 4d). Contrastingly, eCpG release remained at a low level when the ATP and MMP-2 inputs are not simultaneously available (Fig. 2i). These observations collectively supported the multivariate AND-gate operation of ACP assemblies was well maintained after integration into the fusogenic liposomal platform. Alternatively, Lip@AUR-AC$^{Cy5}$-aptPD-L1 showed high sensitivity to ATP input even in the absence of MMP-2, where the eCpG release rate reached around 63.51% under a low ATP concentration of 100 nM (Fig. 2j), immediately suggesting the compromised release control due to the lack of the engineered PNA segments. Considering the universal ATP release and MMP-2 upregulation in IR-treated tumors in the clinics, it is anticipated that the multivariate AND-gate operation of the liposome-integrated ACP assemblies could remain stable in response to ATP in physiological environment at background concentrations while enabling superior spatiotemporal control over their eCpG-dependent immunomodulatory activity, supporting its potential utility for post-IR immunostimulation.

## Cell-nano-interaction modes of Lip-ACP-aptPD-L1

To investigate the targeting ability of aptPD-L1 or eCpG in TME under clinically relevant conditions in vitro, we synthesized aptPD-L1 or eCpG with fluorescent 5′-FAM tags, thus allowing the accurate profiling of the

cellular distribution of the aptamer species in the B16F10/splenocyte co-incubation system. Concrete evidence confirms that B16F10 cells have strong similarities with human melanoma cells in terms of upregulated PD-L1, VEGF, and HIF-1α expression[58], hyperglycolysis metabolism trait[59] and pathological traits including high invasiveness and metastasis risk[60,61], which is a widely used model system in melanoma research. Flow cytometric results immediately suggested that the amount of aptPD-L1 bound to B16F10 cell surface was 410% higher than splenocytes, which was in line with the elevated PD-L1 expression status of melanoma cells compared with their normal counterparts or immune cells (Fig. 3a). Alternatively, eCpG showed preferential binding to DCs that was 220% higher than other cells (Fig. 3a). The fusion of Lip-ACP-aptPD-L1 with cytoplasmic membrane would trigger the anchoring of the ACP assemblies on tumor cell surface, which is crucial for enabling the AND-gate logic operation of ACP in IR-treated melanomas. To monitor the kinetic features of the membrane retention of the fusogenic liposomes, we systematically monitored the fluorescence distribution patterns of different Dil-labeled liposomes after incubation for 3/6/12/18 h in the co-culture system of B16F10 cells and mouse splenocytes. As shown in Fig. 3b and Supplementary Figs. 6, 8, both Lip@Dil and Lip@Dil-ACP showed low fusogenic capacity according to the CLSM results. It is notable that the fusogenic capacity of Lip@Dil-ACP was even lower than Lip@Dil, which was attributed to the electrostatic repulsion between the negatively-charged ACP and cytoplasmic membrane that hinders the interaction of liposomal and cytoplasmic membranes. Notably, Lip@Dil-ACP-aptPD-L1 showed evidently superior fusogenic capacity than both Lip@Dil and Lip@Dil-ACP in the co-culture system, ascribing to the integration of melanoma-targeting aptPD-L1 components. The aptPD-L1-boosted membrane fusion of liposomes is consistent with recent findings that the presence of cell-binding ligands could facilitate the fusion of the liposomes and cytoplasmic membrane through enhancing the direct interaction in between[62–65]. Based on the data above, the time interval between liposome administration and IR treatment for the in vitro and in vivo experiments was set to 12 h to ensure that sufficient ACP assemblies were still anchored on tumor cell surface, thus maximizing the eCpG release into the post-IR TME. The tumor-targeted binding and uptake capability of the Lip-AC$^{Cy5}$P-aptPD-L1 liposomes was further validated using tumor spheroid model, evidenced by the strong Cy5 fluorescence in the Lip-AC$^{Cy5}$P-aptPD-L1 group (Fig. 3c).

## Liposome-mediated radiosensitization and the associated immunogenic effects

Immunofluorescence imaging of PD-L1 with fluorescently-labeled antibodies revealed a general positive correlation between IR dose and PD-L1 expression, which was in accordance with the observations in previous reports[66–68] and confirmed the promotional effect of IR on PD-L1 expression (Supplementary Fig. S9). The visual trends above were further analyzed quantitatively using flow cytometry, which revealed that the PD-L1 abundance in the AUR + 8 Gy group was more than 9-fold higher than the AUR + 0 Gy and PBS + 0 Gy groups. These data collectively supported the capacity of IR to upregulate PD-L1 expression in a dose-dependent manner, necessitating the incorporation of additional immunostimulatory modalities to compensate the potential negative impact on post-IR immune responses. Meanwhile, cytotoxicity assay on B16F10 cells or NIH3T3 cells revealed an optimal dosage of Lip@AUR-aptPD-L1 at 40 μg·mL$^{-1}$ for melanoma treatment, based on a balanced consideration of adverse toxicity and therapeutic potency (Supplementary Fig. 10). To test if the liposome-delivered Au-containing AUR could enhance the IR susceptibility of melanoma cells, we incubated B16F10 cells under different conditions of liposomal nanosamples with or without IR treatment. B16F10 cells showed significant resistance to radiotherapy that their survival rate was still around 90% under the IR dose of 4 Gy (Supplementary Fig. 11a). In contrast, the combined treatment of Lip@AUR-aptPD-L1 liposomes

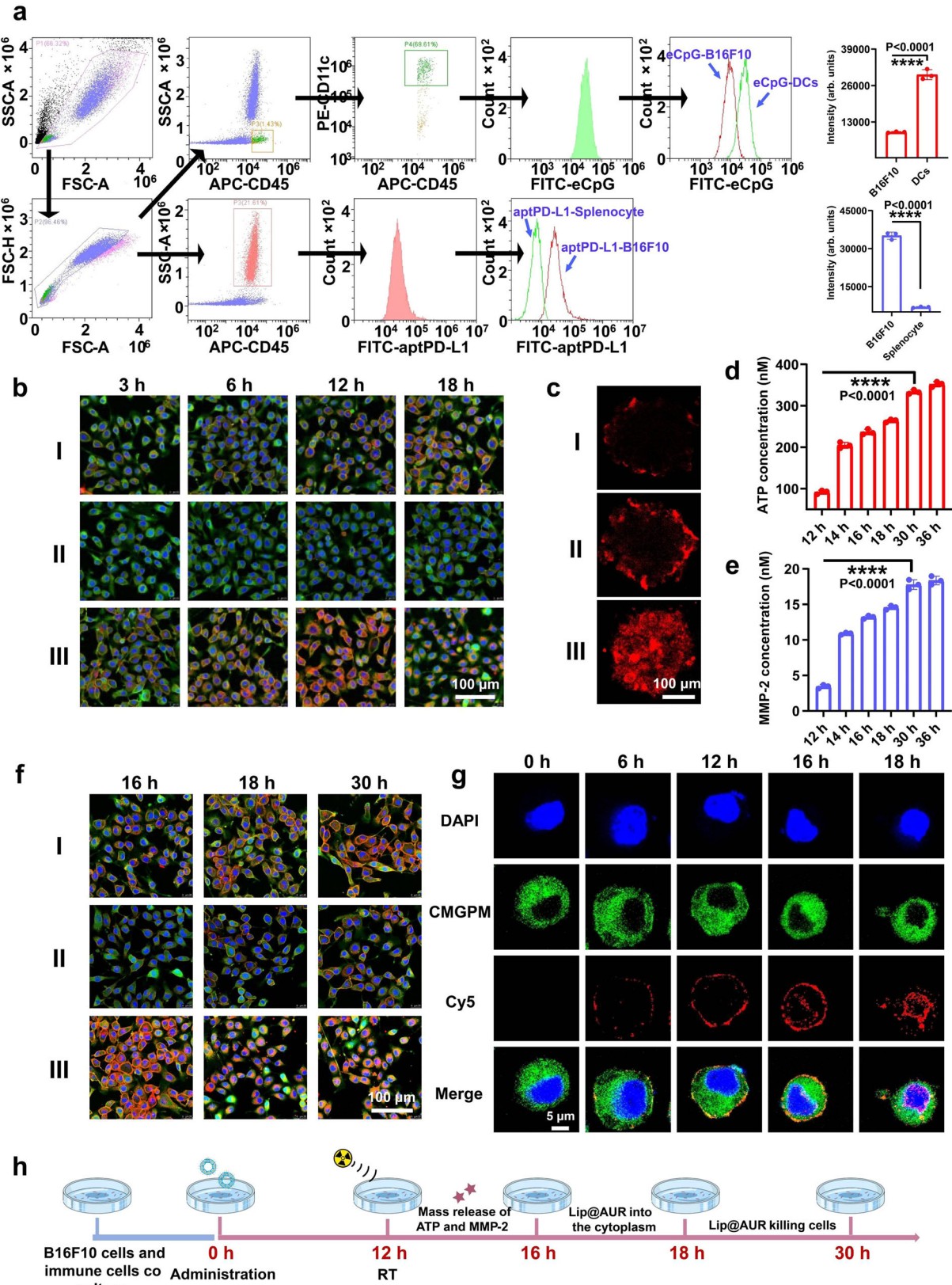

and 4 Gy IR caused significant melanoma inhibition effect, of which the survival rate dropped to only around 65% at 24 h post treatment, evidently supporting the radiosensitization effect of AUR-containing liposomes (Supplementary Fig. 11a). It is also observed that the Lip@AUR-aptPD-L1 + 4 Gy IR treatment induced negligible negative impact on the immune cell populations with a relative viability

decrease of less than 10% (Supplementary Fig. 11b), while the combined treatment of 8 Gy IR and Lip@AUR-aptPD-L1 liposomes caused a 21% reduction in splenocyte survival and the changes were statistically significant, immediately suggesting the importance to lower the necessary IR doses for maximizing the post-IR immunostimulatory effect with optimal safety. It is also of interest to note that

**Fig. 3 | Melanoma targeting and membrane fusion performance of Lip@AUR-ACP-aptPD-L1 in vitro based on B16F10/splenocyte co-incubation system.**
**a** Targeting ability of eCpG and aptPD-L1 with the designated cell targets in the B16F10/splenocyte co-incubation system by flow cytometry. **b** Fusion status of different liposomes to B16F10 cell membranes in the co-culture system at 3, 6, 12, or 18 h of incubation by CLSM (*n* = 3 experimental replicates). I: Lip@Dil, II: Lip@Dil-ACP, III: Lip@Dil-ACP-aptPD-L1. Red: Dil. Green: Invitrogen CellMask™ Green plasma membrane stain. Blue: DAPI. **c** Tumor sphere assay on the targeting ability of different samples at 12 h incubation (*n* = 3 experimental replicates). I: AC^Cy5^P, II: Lip-AC^Cy5^P, III: Lip-AC^Cy5^P-aptPD-L1. **d, e** Time-dependent changes in ATP and MMP-2 abundance in vivo with Lip@AUR-ACP-aptPD-L1 + 4 Gy IR treatment. (**f**) Fusion status of different liposomes to B16F10 cell membranes in the co-culture system at

16, 18 or 30 h of incubation by CLSM (*n* = 3 experimental replicates). 4 Gy IR treatment was applied at 12 h. I: Lip@Dil+IR, II: Lip@Dil-ACP + IR, III: Lip@Dil-ACP-aptPD-L1 + IR. **g** Time-dependent melanoma-targeted membrane fusion performance of Cy5-Lip@AUR-ACP-aptPD-L1 in vivo. 4 Gy IR treatment was applied after 12 h post intravenous injection (*n* = 3 experimental replicates). **h** Schedule of the combinational Lip@AUR-ACP-aptPD-L1 + 4 Gy IR treatment set-up in vitro. Data are presented as mean values ± SEM (n = 3 experimental replicates for (**a**), *n* = 3 mice for (**d**–**e**)). Statistical analysis in (**a**) and (**d**–**e**) was carried out via one-way ANOVA method. * indicates significance at *p* < 0.05, ** indicates significance at *p* < 0.01, *** indicates significance at *p* < 0.001, **** indicates significance at *p* < 0.0001. Source data are provided as a Source Data file.

Lip@AUR-aptPD-L1 liposomes induced slight melanoma inhibition effects even without IR treatment, which was ascribed to the intrinsic anti-tumor activity of AUR and also consistent with the observations in recent reports (Supplementary Fig. 10), although the changes were not therapeutically appreciable due to the low loading amount of AUR[69–71]. Based on a balanced consideration of AUR-enabled radiosensitization and potential risk of immunosuppression according to the above data, the final IR dose for in vitro and in vivo tests was set to 4 Gy. According to the optimized treatment schedule above, Lip@AUR-aptPD-L1 showed significant improvement in the co-culture system even under the low IR dose of 4 Gy according to MTT assay (Supplementary Fig. 12). Next, we measured the total ATP and MMP-2 release in vivo at 12/14/16/18/30/36 h incubation with Lip@AUR-ACP-aptPD-L1 + 4 Gy IR treatment, which eventually reached a plateau after 30 h incubation (Fig. 3d, e). It is also observed that the B16F10-associated Dil fluorescence in the Lip@Dil-ACP-aptPD-L1 with 4 Gy IR group was gradually translocated to the cytoplasm after 16 h incubation (Fig. 3f and Supplementary Fig. 13), which was similar to the result in Fig. 3b. To test if the parameters of the in vitro experiment could be used to design the experimental set-up for in vivo analysis, Cy5-labeled liposomes (Cy5-Lip@AUR-ACP-aptPD-L1) were synthesized and administered into B16F10 tumor-bearing mice through tail vein injection. The time of injection was set as the start of the treatment period (0 h) and IR (4 Gy) was applied at 12 h. Melanoma tissues were extracted at 0 h, 6 h, 12 h, 16 h, and 18 h, which were pulverized, filtered, and lysed with RBC lysis buffer for single-cell CLSM imaging. Images in Fig. 3g showed that substantial amount of Cy5-Lip@AUR-ACP-aptPD-L1 were already bound to the B16F10 cell membrane at 6 h, immediately suggesting the melanoma-targeting effect of the liposomes in vivo. Indeed, apparent Cy5 fluorescence was detected in the cytoplasm of B16F10 cells at 16 h, and at 18 h almost all the Cy5 fluorescence was enriched in the cytoplasm while the membrane-bound Cy5 fluorescence decreased to a negligible level. The melanoma cell membrane fusion performance of Lip@AUR-ACP-aptPD-L1 in vivo was consistent with our in vitro observations and the time interval between liposome administration and IR exposure was determined at 12 h for maximizing eCpG release into the post-IR TME.

Based on the kinetic insights described above, the treatment schedule of Lip@AUR-aptPD-L1 in vitro was established and shown in Fig. 3h to ensure balanced AUR-mediated IR sensitization/VEGF inhibition and logic operation of ACP. The crosstalk between tumor cells and immunosuppressive cells is a major driver of the immunosuppressive TME. There is already clinical evidence that VEGF secreted by melanoma cells could recruit MDSCs and Tregs to TME for suppressing the effector function of CTLs, thus contributing to their immune escape. Interestingly, recent reports reveal that AUR could demonstrate potent VEGF suppressing capability through inhibiting ERK1/2-HIF-1α signaling activity in tumor cells[72–74]. Indeed, we have carried out transcriptome sequencing on Lip@AUR-aptPD-L1-treated B16F10 cells to screen the treatment-induced impact on various immune-related signaling pathways, and the KEGG enrichment analysis results

immediately suggested that Lip@AUR-aptPD-L1 treatment pronouncedly inhibited the VEGF signaling pathways (Supplementary Figs. 14, 17). The VEGF-inhibiting function of AUR-incorporated liposomes was investigated in greater detail via western blot assay. As shown in Supplementary Fig. 18, sole IR treatment induced significant activation of the ERK1/2-HIF-1α-VEGF axis, which was attributed to the oxygen-consumption effect of IR and consistent with the clinical data in previous reports[75–78]. Similar trends in the activation status of ERK1/2-HIF-1α-VEGF signaling pathway were also observed in those non-AUR-containing groups including Lip+IR, Lip-aptPD-L1 + IR, and Lip-ACP-aptPD-L1 + IR, suggesting their inability to suppressive VEGF expression in melanoma cells. In contrast, Lip@AUR-aptPD-L1 + IR and Lip@AUR-ACP-aptPD-L1 + IR both induced obvious inhibition on pERK1/2, HIF-1α, and VEGF regardless of the IR treatment condition. The WB data on HIF-1α expression after different treatments were further supported by immunofluorescence imaging, which showed that the Lip@AUR-aptPD-L1 and Lip@AUR-ACP-aptPD-L1 treatments induced evident reduction in the HIF-1α-intrinsic red fluorescence compared with the other AUR-free modalities, again confirming that AUR component in the liposomes could inhibit HIF-1α expression in melanoma cells (Supplementary Fig. 19). The data above collectively confirmed that the AUR component in the Lip@AUR-ACP-aptPD-L1 liposomes could effectively inhibit VEGF expression in IR-treated melanoma cells through inhibiting ERK1/2-HIF-1α axis, offering potential opportunities to impede the recruitment of immunosuppressive cells into TME for restoring antitumor immunity.

The potential therapeutic benefit of liposome-induced VEGF suppression was evaluated using co-culture system of B16F10 cells and splenocytes. Flow cytometry analysis showed that fewer Tregs and MDSCs were recruited after Lip@AUR-aptPD-L1 + IR treatment, which were 18.47% and 9.44% lower than the control group (Supplementary Fig. 20), respectively, accompanied with increasing DC (14.17%, Supplementary Fig. 21b) and CD8+ T cell (12.23%, Supplementary Fig. 21a) infiltration. The results showed that AUR-mediated VEGF inhibition could potentially establish an anti-tumorigenic microenvironment by reducing Treg and MDSC infiltration. We further investigated if the Lip@AUR-aptPD-L1-mediated radiosensitization of melanoma cells could enhance their immunogenic feature and contribute to immunostimulation. Here we first monitored the cellular status of key DAMPs including ATP (Supplementary Fig. 22a), HMGB1 (Supplementary Fig. 22b), and CRT (Supplementary Fig. 22c) using the corresponding assay kits. Notably, untreated B16F10 cells showed negligible CRT expression as well as low levels of ATP and HMGB1 release, which is in accordance with their low immunogenic potential under common conditions. Low dose (4 Gy) IR treatment induced significant enhancement in CRT expression (140%) and ATP/HMGB1 release (170%/130%) (Supplementary Fig. 22), which was attributed to the IR-induced ICD of melanoma cells. However, the relative increase for the abundance of typical DAMPs in IR-treated B16F10 cells were modest at most due to ineffective radiotherapeutic effect. Remarkably, melanoma cells in the Lip@AUR-aptPD-L1 + IR group showed the greatest increase in CRT expression (370%) and ATP/HMGB1 secretion (570%/

310%) compared with the control group (Supplementary Fig. 22), which is in line with the pronounced radiosensitization effect of the Lip@AUR-aptPD-L1 liposomes. These observations evidently supported our hypothesis that the radiosensitization effect of the Lip@AUR-aptPD-L1 liposomes could induce pronounced ICD of melanoma cells and thus offer multifaceted therapeutic benefit. On one hand, the released DAMPs and tumor-associated neoantigens could stimulate the adaptive immune system to initiate antitumor immune responses. On the other hand, the enhanced ATP secretion could cooperate with IR up-regulated MMP-2 to trigger the AND-gate activation of the ACP assembly and release eCpG into TME for programmable sequential DC stimulation.

### AND-gate eCpG release and the immunostimulatory effects of liposomes

Extending from the IR-triggered liposome-augmented ICD of melanoma cells above, we further comprehensively investigated the immunostimulatory impact of liposome-sensitized melanoma radiotherapy in vitro. To start with, we evaluated if the molecular engineering of 5′ end of CpG ODN would alter its immunological activities via NUPACK analysis. As shown by the simulation results, the addition of the 10-base aptATP binding sequence caused no alterations in the structure of the stem-loop domain (Fig. 4a, b). Subsequently, we employed 3D model-based molecular dock analysis to further profile the complexation of pristine CpG ODN and eCpG with TLR9 proteins. The binding sequence of CpG ODN to TLR9 is base 6-11 (GACGTT) that directly complexes to 337Arg and 338Lys on TLR9 while also presenting indirect interaction with 347Lys, 348Arg and 353His (Fig. 4c, d), which was consistent with the structural analysis in previous reports[79–81]. Interestingly, eCpG bond to TLR9 through the same GACGTT sequence with identical amino acid interaction, immediately suggesting that the addition of aptATP-binding sequence at the 5′ end of CpG induced negligible impact on its TLR9-binding behavior. We further prepared Cy5 labeled eCpG and tested their binding with TLR9-positive DCs (Fig. 4e). Notably, eCpG showed comparable TLR9-binding affinity to pristine CpG ODN and was capable of substantially promoting DC maturation (51.10%) (Fig. 4f), while mutating the CG bases in the GACGTT sequence induced significant reduction in the DC-binding capacity of the aptamers and failed to induce significant changes in DC maturation ratio after co-incubation. Meanwhile, we detected that pretreating eCpG with the complementary sequence (CTGCAA) of the TLR9-binding domain also impaired their complexation with TLR9-positive DCs and abolished their pro-DC maturation function (20.80%) (Fig. 4f). These results collectively supported that the molecularly engineered eCpG successfully expanded its nanointegrative functionality without impairing its DC-stimulatory activity.

Next, we investigated if the Lip-ACP-aptPD-L1 liposomes could stimulate the DC populations through mediating AND-gate eCpG release in B16F10 cells. To monitor the cellular distribution of eCpG, the eCpG molecules were labeled by Cy5 for fluorescent tracking, of which the samples were denoted as Lip-AC^Cy5-aptPD-L1 (PNA-free) and Lip-AC^Cy5P-aptPD-L1 (PNA complexed). By referring to the ATP and MMP-2 abundance in B16F10 cells treated with 4 Gy IR (Fig. 3d, e), Lip-AC^Cy5-aptPD-L1 and Lip-AC^Cy5P-aptPD-L1 were used to incubate B16F10 cells with 0 nM ATP + 5 nM MMP-2, 100 nM ATP + 5 nM MMP-2, 0 nM ATP + 10 nM MMP-2 or 200 nM ATP + 10 nM MMP-2 for 2 h (Fig. 4g, h). Notably, Lip-AC^Cy5-aptPD-L1 only showed partial eCpG release under the incubation condition of 100 nM ATP + 5 nM MMP-2, while Lip-AC^Cy5P-aptPD-L1 did not demonstrate obvious eCpG release under the same incubation condition. In comparison, both Lip-AC^Cy5-aptPD-L1 and Lip-AC^Cy5P-aptPD-L1 showed almost complete eCpG release under the incubation conditions of 200 nM ATP + 10 nM MMP-2. In addition, fluorescence analysis of Cy5 on the cell membrane and supernatant of Lip-AC^Cy5P-aptPD-L1-treated B16F10 cells also showed that significant

proportion of eCpG^Cy5 was released after IR treatment (Supplementary Fig. 23). The eCpG release was further investigated in vivo by treating B16F10 tumor-bearing mouse models with Lip@AUR-AC^Cy5-aptPD-L1 or Lip@AUR-AC^Cy5P-aptPD-L1 via intravenous injection and 4 Gy IR at 12 h post injection, and the release of eCpG^Cy5 from the B16F10 surfaces was eventually monitored by CLSM at 16 h post intravenous injection. Specifically, most of eCpG^Cy5 of Lip@AUR-AC^Cy5-aptPD-L1 was released regardless of the IR treatment conditions, while eCpG^Cy5 of Lip@AUR-AC^Cy5P-aptPD-L1 was mostly retained on B16F10 tumors in the absence IR treatment but almost completely released after IR exposure (Fig. 4i). These observations immediately suggested the capacity of Lip-AC^Cy5P-aptPD-L1/Lip@AUR-AC^Cy5P-aptPD-L1 to release the eCpG payload under conditions resembling IR-treated tumors in a spatiotemporally coordinated manner and confirmed the necessity of the multivariate AND-gate operation of the ACP assembly for maximizing the immunostimulatory benefit while reducing potential adverse immune reactions. The time-dependent execution of the eCpG release from membrane-bound ACP assemblies was further investigated in vivo using B16F10 tumor-bearing mice and liposomes constructed with Cy5-labeled eCpG sequences. As shown in Fig. 4j, the B16F10-specific eCpG accumulation reached the maximum at 12 h post-injection but decreased to a negligible level at 18 h post-injection, suggesting the gradual release of eCpG from B16F10 cell surface during the 12–18 h period. As a result of the efficient AND-gate eCpG release, DCs in the Lip@AUR-ACP-aptPD-L1 + IR group showed the highest maturation ratio (CD11c + CD80 + CD86+) after 30 h incubation in vitro (Fig. 4k), indicating that 4 Gy IR successfully triggered the AND-gate eCpG release to promote DC maturation. To further investigate if the released eCpG could promote DC maturation in the post-IR TME, we also isolated DC populations from the extracted tumors for flow cytometric analysis. According to the data in Fig. 4l, m, the relative frequency of mature DCs showed no obvious changes in the 0–12 h period post-injection but started to increase rapidly after the time point of 16 h, which eventually reached a plateau after 30 h. These observations confirmed that eCpG released through the AND-gate operation of the membrane-bound ACP assemblies efficiently promoted DC maturation in the post-IR TME in a time-sequenced manner, which is conducive for promoting the adaptive antitumor immune response in IR-treated melanomas.

We further studied whether the liposome-augmented IR-induced ICD of melanoma cells and the cooperative AND-gate eCpG release could evoke adaptive immunity to achieve effective radio-immunotherapy against melanomas. It is well-established that tumor cells undergoing ICD would release tumor-associated immunogenic materials for the processing and recognition by tumor-infiltrating antigen-presenting cells for mediating the downstream immune reactions. Firstly, to verify whether Lip@AUR-ACP-aptPD-L1 could cause ICD in co-culture system, we detected the expressions of HMGB1 and CRT in B16F10 cells. The confocal microscope results showed that Lip@AUR-ACP-aptPD-L1 + IR group significantly decreased the expression of HMGB1 in the nucleus as well as significantly increased the expression of CRT on the cell membrane (Supplementary Fig. 24), indicating that Lip@AUR-ACP-aptPD-L1 combined with 4 Gy IR could successfully induce significant tumor immunogenic death. Indeed, flow cytometric analysis on the extracted immune cell populations from the co-incubation system showed that the combined Lip@AUR-ACP-aptPD-L1 + IR treatment substantially improved the maturation and antigen-presentation capacity of DC population, where the frequencies of CD11c + CD80 + CD86+ (Fig. 5a and Supplementary Figs. 26a, 27a) and CD11c+MHC-II+ (Supplementary Fig. 25a) DCs have increased by 31.76% and 34.44% compared with the control group and obviously higher than all other groups. In line with the enhanced activation status of DCs, the Lip@AUR-ACP-aptPD-L1 + IR group showed a substantial expansion of the CD4 + CD8+ T cell populations to around 77.66% (Fig. 5b and Supplementary Figs. 26b, 27b), while the

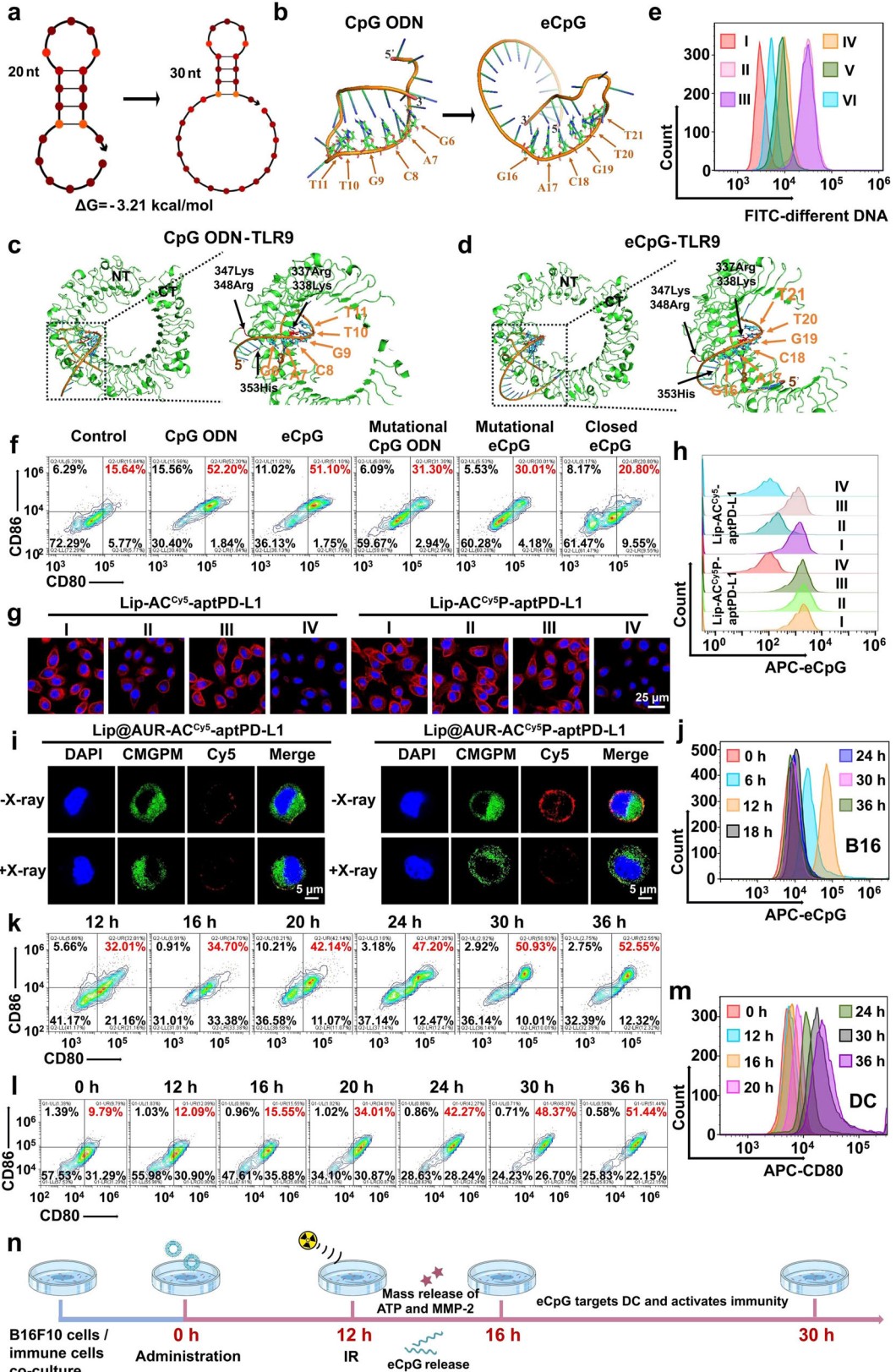

frequency of IFN-γ + CD8+ (Supplementary Fig. 25b) T cells had also increased to 45.81%, suggesting effective DC-mediated priming of antitumor T cells thereof. In addition, the secretion of key immune-related molecular markers in the co-incubation system was analyzed by ELISA assay to indicate the alteration in the immune composition, and the results revealed that the secretion levels of pro-inflammatory

cytokines and chemokines including IFN-γ (Fig. 5c), TNF-α (Fig. 5d), CXCL10 (Fig. 5e) and IL-2 (Fig. 5f) in the Lip@AUR-ACP-aptPD-L1 + IR group were the highest among all groups, which have increased by 4.2-fold, 6.3-fold, 4.3-fold and 5.5-fold compared to the control group, respectively. In contrast, the secretion levels of anti-inflammatory cytokines including IL-4 (Supplementary Fig. 28a), IL-10

**Fig. 4 | Evaluation on AND-gate release of eCpG from Lip@AUR-ACP-aptPD-L1 for DC stimulation. a** NUPACK analysis on eCpG secondary structure. **b–d** Molecular docking analysis on the TLR9-binding behaviors of CpG ODN and eCpG. **e** Flow cytometry analysis on the target binding of different DNA sequences to DCs (*n* = 3 experimental replicates). I: Control, II: CpG ODN, III: eCpG, IV: mutated CpG ODN, V: mutated eCpG, VI: closed eCpG. **f** Stimulatory impact of various DNA sequences to DC maturation by flow cytometry analysis (*n* = 3 experimental replicates). I: Control, II: CpG ODN, III: eCpG, IV: mutated CpG ODN, V: mutated eCpG, VI: closed eCpG. **g** The CLSM analysis regarding the effect of PNA complexation on eCpG$^{Cy5}$ release from membrane-bound aptamer assemblies under different conditions (*n* = 3 experimental replicates). I: 0 nM ATP + 5 nM MMP-2, II: 100 nM ATP + 5 nM MMP-2, III: 0 nM ATP + 10 nM MMP-2, IV: 200 nM ATP + 10 nM MMP-2. **h** Flow cytometry analysis regarding eCpG$^{Cy5}$ release from membrane-bound aptamer assemblies under different conditions (*n* = 3 experimental replicates). I: 0 nM ATP + 5 nM MMP-2, II: 100 nM ATP + 5 nM MMP-2, III: 0 nM ATP + 10 nM MMP-2, IV: 200 nM ATP + 10 nM MMP-2. **i** CLSM analysis regarding PNA complexation on eCpG$^{Cy5}$ released from membrane-bound aptamer assemblies at 16 h incubation in vivo with three mice per group. **j** Time-dependent analysis on eCpG$^{Cy5}$ release from Lip@AUR-AC$^{Cy5}$P-aptPD-L1 in vivo with three mice per group. **k** Time-dependent evaluation on DC maturation status (CD11c + CD80 + CD86+) after Lip@AUR-ACP-aptPD-L1 + 4 Gy IR treatment (*n* = 3 experimental replicates). **l, m** Time-dependent evaluation on DC maturation status (CD11c + CD80 + CD86+) after Lip@AUR-ACP-aptPD-L1 + 4 Gy IR treatment in vivo with three mice per group. **n** Treatment schedule for the B16F10-mouse splenocyte co-incubation system for the evaluation of the immunostimulatory effects.

(Supplementary Fig. 28b), and TGF-β (Supplementary Fig. 28c) in the Lip@AUR-ACP-aptPD-L1 + IR group were the lowest among all groups, which have decreased by 72%, 77%, and 75% compared with the control group, respectively. Extending from the mechanistic evaluations above, we then systematically evaluated the antitumor efficacy of the liposome-augmented radio-immunotherapy using B16F10/mouse splenocyte co-incubation system. According to the flow cytometric data, the death rate of B16F10 cells in Lip@AUR-ACP-aptPD-L1 + IR group reached around 76.78%, which was almost 9-fold higher than the PBS + IR group (Fig. 5g and Supplementary Fig. 26c, 27c). Consistently, MTT data showed that Lip@AUR-ACP-aptPD-L1 + IR group presented the lowest B16F10 survival rate of only around 17.98% (Supplementary Fig. 29). It is also of interest to note that B16F10 cells in the Lip@AUR-ACP-aptPD-L1 + IR group showed elevated γ-H2AX levels, a typical marker of IR-induced DNA damage, according to immunochemical staining and western blotting analysis (Fig. 5h, Supplementary Fig. 26d, 27d and Supplementary Fig. 30), again validating the therapeutic contribution of AUR-mediated radiosensitization. These observations are immediate evidence that the Lip@AUR-ACP-aptPD-L1 could enhance the radio-immunotherapeutic efficacy against melanoma cells in vitro through a programmable sequential manner.

**Investigations on IR treatment parameters on the radio-immunotherapeutic efficacy in vivo**

It is well-established in clinical practice that IR doses and fractionations have significant impact on their tumoricidal and immunostimulatory effects. Therefore, we comprehensively studied the effect of these IR-related parameters on the eventual radio-immunotherapeutic efficacy. Herein, the B16F10 tumor-bearing C57BL/6J mice were subjected to the treatment of PBS and Lip@AUR-ACP-aptPD-L1 as well as different IR treatment schedules for 15 days. For the evaluation of the dose-dependent impact of IR treatment on the radio-therapeutic efficacy, the melanoma sites were treated with 0 Gy, 2 Gy, 4 Gy, and 8 Gy IR with a 5-day interval through the 15-day treatment period (3 times in total) (Supplementary Fig. 32a). Notably, sole IR treatment under all dose conditions did not induce obvious tumor inhibition effect, while the coordinated treatment of Lip@AUR-ACP-aptPD-L1 and IR significantly improved the melanoma inhibition efficacy, where the average tumor weight in the Lip@AUR-ACP-aptPD-L1 + 4 Gy and Lip@AUR-ACP-aptPD-L1 + 8 Gy groups was only 0.25 g and 0.22 g with no statistical difference in between. These observations were consistent with the TUNEL staining results, which revealed that the Lip@AUR-ACP-aptPD-L1 + 4 Gy and Lip@AUR-ACP-aptPD-L1 + 8 Gy groups had the largest dead melanoma cell populations (Supplementary Figs. 31, 32b–d). Flow cytometric analysis on the immunocomposition in the extracted tumors showed that the frequencies of DCs, CD4+ T cells, CD8+ T cells, Tregs, and MSDCs in the PBS + IR groups all showed no significant changes, indicating that there was no obvious alleviation of the immunosuppressive TME. However, the Lip@AUR-ACP-aptPD-L1 + 4 Gy and Lip@AUR-ACP-aptPD-L1 + 8 Gy treatments both induced

significant expansion of the DC, CD4+ T cell and CD8+ T cell populations, while the tumor-residing Treg and MSDC populations have markedly decreased, suggesting the successful remodeling of the TME into an immunoactivated state (Supplementary Fig. 32e–h). These data also confirmed the superior suitability of the Lip@AUR-ACP-aptPD-L1 + 4 Gy for melanoma treatment due to its potent radio-immunotherapeutic efficacy at a relatively lower IR dose.

The impact of IR fractionation on the melanoma inhibition and immunostimulation effects was also studied in vivo, where the B16F10 tumor-bearing mice were treated with 4 Gy IR for 0 time, 1 time (at the start of the treatment period), 3 times (with a 4-day interval) and 5 times (with a 2-day interval) through the 15-day treatment period (Supplementary Fig. 34a). Notably, treating the melanoma-bearing mice with IR for 0 time and 1 time induced no obvious alleviation of the tumor burden even with Lip@AUR-ACP-aptPD-L1 pretreatment. In comparison, the combinational treatment of Lip@AUR-ACP-aptPD-L1 + 4 Gy × 3 and Lip@AUR-ACP-aptPD-L1 + 4 Gy × 5 drastically inhibited tumor growth with a low average final tumor weight of 0.26 g and 0.24 g and pronounced melanoma cell death under TUNEL staining (Supplementary Figs. 33, 34b–d). Flow cytometric analysis on the extracted tumors revealed that the TME remodeling effect of the combinational Lip@AUR-ACP-aptPD-L1 + IR treatment showed a general positive correlation with the times of IR exposure, where increasing the rounds of IR treatment at 4 Gy could elevate the frequency of tumor-residing DCs and CD4+/CD8+ T cells while reducing immunosuppressive Treg and MSDC populations (Supplementary Fig. 34e–h). Nevertheless, it is notable that there is no significant difference between the antimelanoma and immunostimulatory effects of Lip@AUR-ACP-aptPD-L1 + 4 Gy × 3 and Lip@AUR-ACP-aptPD-L1 + 4 Gy × 5 groups. These observations again validated the feasibility of Lip@AUR-ACP-aptPD-L1 + 4 Gy × 3 treatment as the standard condition for melanoma treatment in vivo, which may elicit optimal anti-melanoma efficacy with reduced IR exposure.

**Therapeutic evaluation of Lip@AUR-ACP-aptPD-L1 in vivo**

The in vivo therapeutic evaluation of the liposomal systems was further carried out on B16F10 melanoma-bearing C57BL/6J mice, which have marked resemblance with melanomas on real life patients in terms of pathologic, metabolic, and immunological traits[82]. To monitor the pharmacokinetic properties of the liposomal systems in vivo, AUR, Lip@AUR or Lip@AUR-aptPD-L1 were injected into C57BL/6J mice through the tail vein with three mice per group, then AUR content in tail venous blood of mice was detected by HPLC at scheduled time points, which could quantitatively determine the AUR levels with molecular precision and thus indicate the blood retention time of the injected samples. The Lip@AUR-aptPD-L1 liposomes showed significantly longer blood circulation time compared with AUR, of which the blood half-life has increased by 5-fold and reached around 8 h (Supplementary Fig. 35). The liposome-enhanced blood retention capacity of the therapeutic contents is particularly important for their subsequent radio-immunotherapeutic activities, which may facilitate

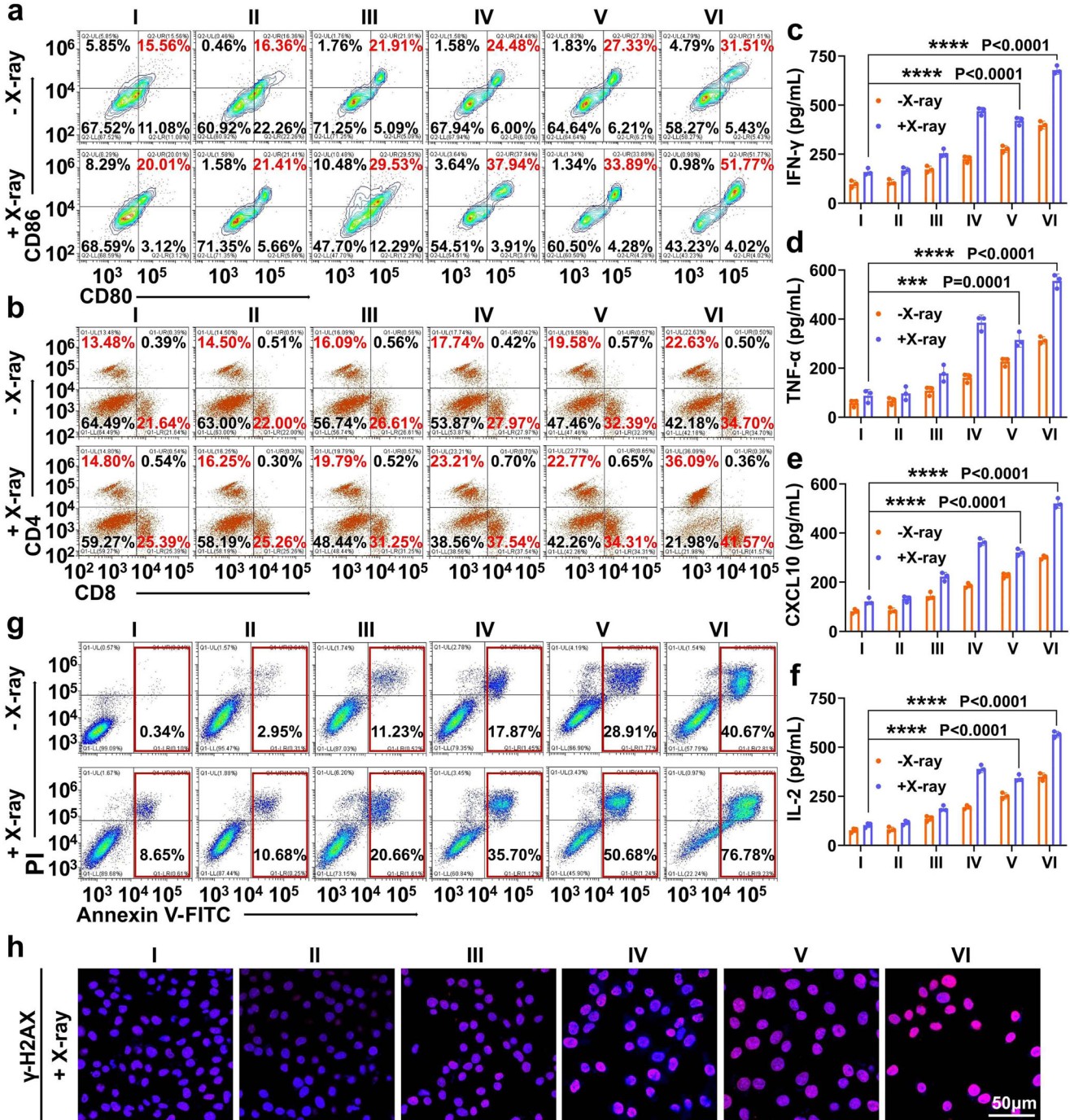

**Fig. 5 | Immunostimulatory effect of combined Lip@AUR-ACP-aptPD-L1 and 4 Gy IR treatment in vitro. a** Flow cytometry analysis on the maturation status (CD11c + CD80 + CD86+) of DCs in the co-incubation system after different treatments (*n* = 3 experimental replicates). **b** Flow cytometry analysis on T cell activation status (CD3 + CD4 + CD8+) in the co-incubation system after different treatments (*n* = 3 experimental replicates). **c–f** Secretion levels of immunostimulatory markers including IFN-γ, TNF-α, CXCL10, and IL-2 in the supernatants of the co-culture system after different treatments. **g** Flow cytometry analysis on the apoptosis of B16F10 cells after different treatments in co-culture system (*n* = 3 experimental

replicates). **h** γ-H2AX immunofluorescence of IR-treated B16F10 cells after different sample treatments (*n* = 3 experimental replicates). I: PBS, II: Lip, III: Lip-aptPD-L1, IV: Lip-ACP-aptPD-L1, V: Lip@AUR-aptPD-L1, VI: Lip@AUR-ACP-aptPD-L1. Data in (**c–f**) are presented as mean values ± SEM (*n* = 3 experimental replicates). Statistical analysis in (**c–f**) was carried out via one-way ANOVA method. * indicates significance at *p* < 0.05, ** indicates significance at *p* < 0.01, *** indicates significance at *p* < 0.001, **** indicates significance at *p* < 0.0001. Source data are provided as a Source Data file.

the interaction of the liposomes with melanoma tissues. Meanwhile, we also profiled the systemic distribution of the liposomes by measuring the AUR abundance in specific organs and tissues via ICP test in vivo using B16F10 tumor-bearing C57BL/6J mouse model, on account of the wide applicability of ICP analysis for tracking trace elements in complex samples. The comparative analysis of AUR deposition patterns immediately suggested that Lip@AUR-aptPD-L1 liposomes predominantly accumulated in the B16F10 tumors with a relative ratio of around 46% after 24 h of administration (Supplementary Fig. 36). In contrast, non-targeting Lip@AUR liposomes were

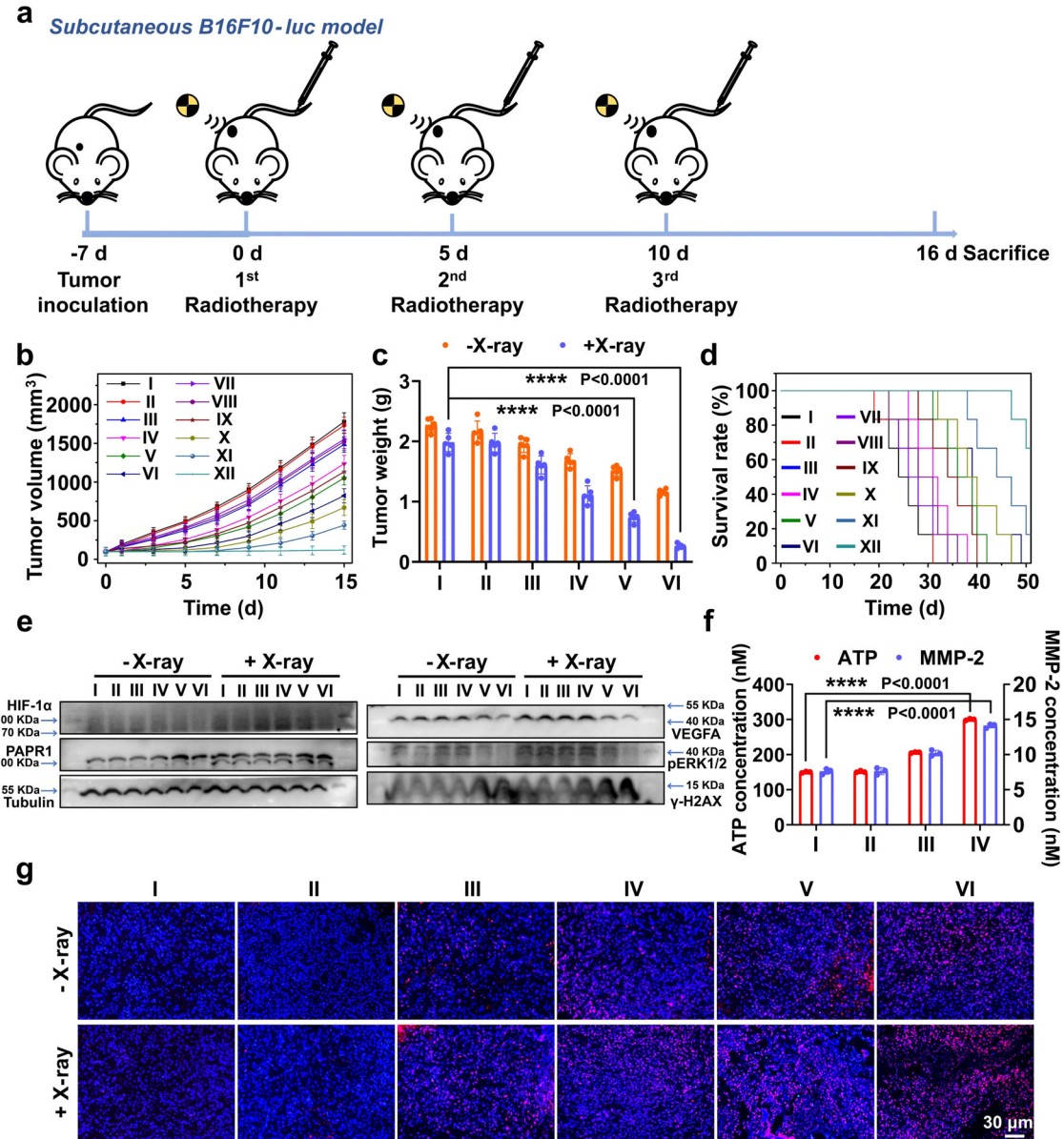

**Fig. 6 | Anti-tumor effects of Lip@AUR-ACP-aptPD-L1-augmented radio-immunotherapy in vivo. a** Schematic representation of the treatment protocol for B16F10-luc tumor-bearing mice. **b** Tumor volume analysis throughout the treatment period. I: PBS, II: Lip, III: Lip-aptPD-L1, IV: Lip-ACP-aptPD-L1, V: Lip@AUR-aptPD-L1, VI: Lip@AUR-ACP-aptPD-L1, VII: PBS + IR, VIII: Lip+IR, IX: Lip-aptPD-L1 + IR, X: Lip-ACP-aptPD-L1 + IR, XI: Lip@AUR-aptPD-L1 + IR, XII: Lip@AUR-ACP-aptPD-L1 + IR. **c** Tumor weight analysis at the end of the treatment period. I: PBS, II: Lip, III: Lip-aptPD-L1, IV: Lip-ACP-aptPD-L1, V: Lip@AUR-aptPD-L1, VI: Lip@AUR-ACP-aptPD-L1. **d** Survival analysis of mice after different treatments. I: PBS, II: Lip, III: Lip-aptPD-L1, IV: Lip-ACP-aptPD-L1, V: Lip@AUR-aptPD-L1, VI: Lip@AUR-ACP-aptPD-L1, VII: PBS + IR, VIII: Lip + IR, IX: Lip-aptPD-L1 + IR, X: Lip-ACP-aptPD-L1 + IR, XI: Lip@AUR-aptPD-L1 + IR, XII: Lip@AUR-ACP-aptPD-L1 + IR. **e** Western blotting on the expression levels of related proteins in the tumor tissues with five mice per group. I: PBS, II: Lip, III: Lip-aptPD-L1, IV: Lip-ACP-aptPD-L1, V: Lip@AUR-aptPD-L1, VI: Lip@AUR-ACP-aptPD-L1. **f** The release level of ATP and MMP-2 in B16F10 tumors after different treatment. I: PBS + IR, II: Lip+IR, III: Lip@AUR + IR, IV: Lip@AUR-aptPD-L1 + IR. **g** TUNEL staining of tumor tissue samples after treatment with five mice per group. I: PBS, II: Lip, III: Lip-aptPD-L1, IV: Lip-ACP-aptPD-L1, V: Lip@AUR-aptPD-L1, VI: Lip@AUR-ACP-aptPD-L1. Data are presented as mean values ± SEM (*n* = 5 mice for (**b**–**c**), *n* = 6 mice for (**d**), *n* = 3 mice for (**f**)). Statistical analysis in (**c**, **f**) was carried out via one-way ANOVA method. * indicates significance at *p* < 0.05, ** indicates significance at *p* < 0.01, *** indicates significance at *p* < 0.001, **** indicates significance at *p* < 0.0001. Source data are provided as a Source Data file.

mostly detected in mouse kidney, attributing to the liposome clearance capacity of the mononuclear phagocyte system (MPS) therein. The observations validated our hypothesis that the incorporation of aptPD-L1 ligands enables efficient and guided delivery of the liposomes to melanomas after systemic administration. Next, we tested the inhibition effect of the liposome-augmented radio-immunotherapy against B16F10-luc tumors in vivo (Fig. 6a). Mice treated with non-drug-loaded liposomes showed rapid tumor growth similar to the PBS-only control group due to the lack of antitumor function, in which the

average tumor volume reached around 1750 mm³ after 15-day of treatment (Fig. 6b and Supplementary Fig. 37a). Sole IR treatment induced modest inhibition on melanoma growth with a final tumor volume of around 1550 mm³, which was slightly lower than the control group and suggested the innate radiotherapeutic resistance of melanomas (Fig. 6b and Supplementary Fig. 37a). Similarly, treating melanomas with Lip-aptPD-L1 also only induced slight antitumor effect (1490 mm³), attributing to the low aptPD-L1 dosage as well as the immunosuppressive TME. Remarkably, the combination of Lip@AUR-

ACP-aptPD-L1 and 4 Gy IR induced the highest melanoma inhibition among all groups, of which the final tumor volume was only around 95 mm³ (Fig. 6b and Supplementary Fig. 37a). Analysis of tumor weight revealed the same trend that the Lip@AUR-ACP-aptPD-L1 + IR group showed the lowest final tumor weight of around 0.26 g (Fig. 6c). Resulting from the treatment-ameliorated tumor burdens, mice in Lip@AUR-ACP-aptPD-L1 + IR group presented the longest average survival time with a median survival period of more than 50 days (Fig. 6d). H&E and TUNEL-based histological analysis on the extracted tumor tissue slices showed that the combined Lip@AUR-ACP-aptPD-L1 + IR treatment induced severe apoptosis in melanoma cells (Fig. 6g and Supplementary Fig. 37b), further substantiating its anti-tumor potency in vivo. Overall, these observations confirmed that combining Lip@AUR-ACP-aptPD-L1 with low-dose IR treatment enabled efficient elimination of melanoma cells in vivo. The biochemical alterations in the extracted tumor samples were further analyzed to clarify the mechanism underlying the liposome-mediated programmable radio-immunotherapeutic effects. Notably, WB analysis revealed that tumors in the Lip@AUR-ACP-aptPD-L1 + IR group presented significant enhancement in the expression levels of γ-H2AX and PARP1 (Fig. 6e and Supplementary Fig. 38), evidently supporting the AUR-mediated radiosensitization effect by enhancing the IR-dependent DNA damage in melanoma cells. Meanwhile, treating mice with AUR-containing samples such as Lip@AUR-aptPD-L1 and Lip@AUR-ACP-aptPD-L1 inhibited key mediators in the ERK1/2/HIF-1α/VEGF pathway in melanoma cells at varying degrees (Fig. 6e), which was consistent with the trends in vitro. Immunofluorescence analysis showed that the Lip@AUR-ACP-aptPD-L1-augmented radio-immunotherapy of melanomas induced evident increases in the tumor abundance of typical DAMPs including CRT (Supplementary Fig. 39a) and HMGB1 (Supplementary Fig. 39b), supporting our hypothesis that the liposome-mediated radiosensitization effect could promote IR-induced ICD of melanoma cells in vivo. Quantitative analysis further demonstrated that the liposome-amplified radiotherapeutic effects caused significant upregulation of ATP and MMP-2 by 2.1-fold and 1.9-fold compared with PBS + IR group in melanoma tissues (Fig. 6f), thus enabling the AND-gate release of eCpG into the tumor tissues for DC stimulation. As the combined result of these immunostimulatory traits, the Lip@AUR-ACP-aptPD-L1 + IR treatment substantially enhanced the overall immune cell infiltration (CD45+) in the melanoma tissues by about 11.99% (Fig. 7a). Specifically, the frequency of mature DCs (CD11c + CD80 + CD86+/CD11c + MHC-II+) in the melanoma tissues in the Lip@AUR-ACP-aptPD-L1 + IR group has increased by more than 30% compared with the control group (Fig. 7b and Supplementary Fig. 41a). Meanwhile, the Lip@AUR-aptPD-L1 + IR group also showed drastically lower frequency of tumor-infiltrating immunosuppressive cells including Tregs (7.31%) (Supplementary Fig. 40a) and MDSCs (1.78%) (Supplementary Fig. 40b), which was in line with the VEGF-inhibiting function of AUR incorporated liposomes. Owing to the liposome mediated stimulation of DCs and inhibition of immunosuppressive cell populations, mice in the Lip@AUR-ACP-aptPD-L1 + IR group showed enhanced tumor infiltration of CD4+/CD8+ T cells that was 37.78% higher than the control group (Fig. 7c), accompanied with a significant expansion of IFN-γ + CD8+ T cells by 30.19% (Supplementary Fig. 41b). The flow cytometric results regarding the tumor infiltration status of various immune cell populations were also consistently supported by the immunofluorescence assay based on relevant markers and CCK8 assay of T cells (Fig. 7d and Supplementary Figs. 42, 43). Extending from the treatment-induced changes in the immunocomposition of the melanoma tissues, tumors in the Lip@AUR-ACP-aptPD-L1 + IR group showed the highest enhancement in the secretion levels of pro-inflammatory cytokines and chemokines including IFN-γ (4.4-fold), TNF-α (6.2-fold), CXCL10 (4.5-fold) and IL-2 (5.6-fold) as well as the greatest reduction in the secretion levels of anti-inflammatory cytokines including IL-4 (25%), IL-10 (24%) and TGF-

β (24%), indicating that the liposome-augmented radio-immunotherapy has significantly boosted the adaptive immune responses for eliminating the melanoma cells in vivo (Fig. 7e–h and Supplementary Fig. 44). In addition to the therapeutic evaluations above, we also comprehensively studied the biocompatibility of the liposomes in vivo from a translational perspective. Notably, mice receiving combinational liposome+IR treatment showed no significant weight loss compared to the PBS-only control group, which not only confirmed the non-toxicity of the liposomes and low-dose IR but also indicated that the liposome-enhanced radio-immunotherapy induced no serious systemic adverse immune reactions (Supplementary Fig. 45). Alternatively, various samples were injected into the B16F10 tumor-bearing C57BL/6J mice through intravenous route for 15 days of treatment, afterwards the slices of major organs were stained by hematoxylin and eosin (H&E) for histological inspections, which revealed that the liposomal samples did not induce obvious damage to major mouse organs regardless of the IR treatment conditions (Supplementary Fig. 46). The histocompatibility of the liposome-enhanced radio-immunotherapy was manifold. On one hand, the melanoma-targeting effect of the liposomes and low IR dose could limit the collateral damage to non-specific organs and tissues. On the other hand, the multivariate-gated operation of the ACP assembly in TME could further improve the spatial-temporal controllability of the eCpG-dependent immunostimulatory effect and reduce the risk of systemic adverse immune responses. These above results suggested that Lip@AUR-ACP-aptPD-L1 liposomes could be a safe and effective option for melanoma radioimmunotherapy.

## Lip@AUR-ACP-aptPD-L1-augmented radio-immunotherapy induced robust systemic antitumor immunity and built immune memory

To investigate if the combinational treatment of Lip@AUR-ACP-aptPD-L1 and low-dose IR could induce robust and long-lasting antitumor immunity to offer systemic protection against invading melanomas, we have developed bilateral B16F10-luc-bearing mouse model for evaluating the therapeutic activities. To construct the bilateral melanoma mouse models, B16F10 cells were first inoculated into the right flank of the mice to establish the primary tumors, while B16F10 cells were later injected into the left flank after 15 days of incubation to create the secondary tumors (Fig. 8a). Mice in the Lip@AUR-ACP-aptPD-L1 + IR group showed the smallest tumor sizes for secondary tumors (88 mm³) (Fig. 8b), indicating the pronounced inhibitory effect thereof. Owing to the efficient treatment-induced melanoma inhibition, mice in the Lip@AUR-ACP-aptPD-L1 + IR group also presented the highest survival time (median survival: 52 days) among all groups (Fig. 8c). Flow cytometry analysis of extracted tumor samples showed a significant increase in the frequency of mature DCs in both primary (Supplementary Fig. 47a) and distal (Fig. 8d) B16F10 tumors in the Lip@AUR-ACP-aptPD-L1 + IR group, which has increased by 32.33% and 31.90% compared with the control group. Consistent with the immunoregulatory role of DCs as the primary APC populations for activating the CTL-mediated adaptive antitumor immunity, the infiltration status of CD8+ T cells in the primary (Supplementary Fig. 47b) and secondary tumors (Fig. 8e) of the Lip@AUR-ACP-aptPD-L1 + IR group was the highest among all groups, indicating that the combined Lip@AUR-ACP-aptPD-L1 and low-dose IR treatment successfully evoked potent systemic antitumor immune responses to eliminate the distal tumors. In addition, we have detected a significant expansion of CD62L-CD44+ memory CD8+ T cells in the melanoma tissue samples according to flow cytometric analysis (Fig. 8f). The results confirmed that the Lip@AUR-ACP-aptPD-L1 + IR-augmented radio-immunotherapy could substantially promote the formation of memory T cells to establish robust antitumor immune memory, which is beneficial for preventing melanoma metastasis and post-treatment relapse.

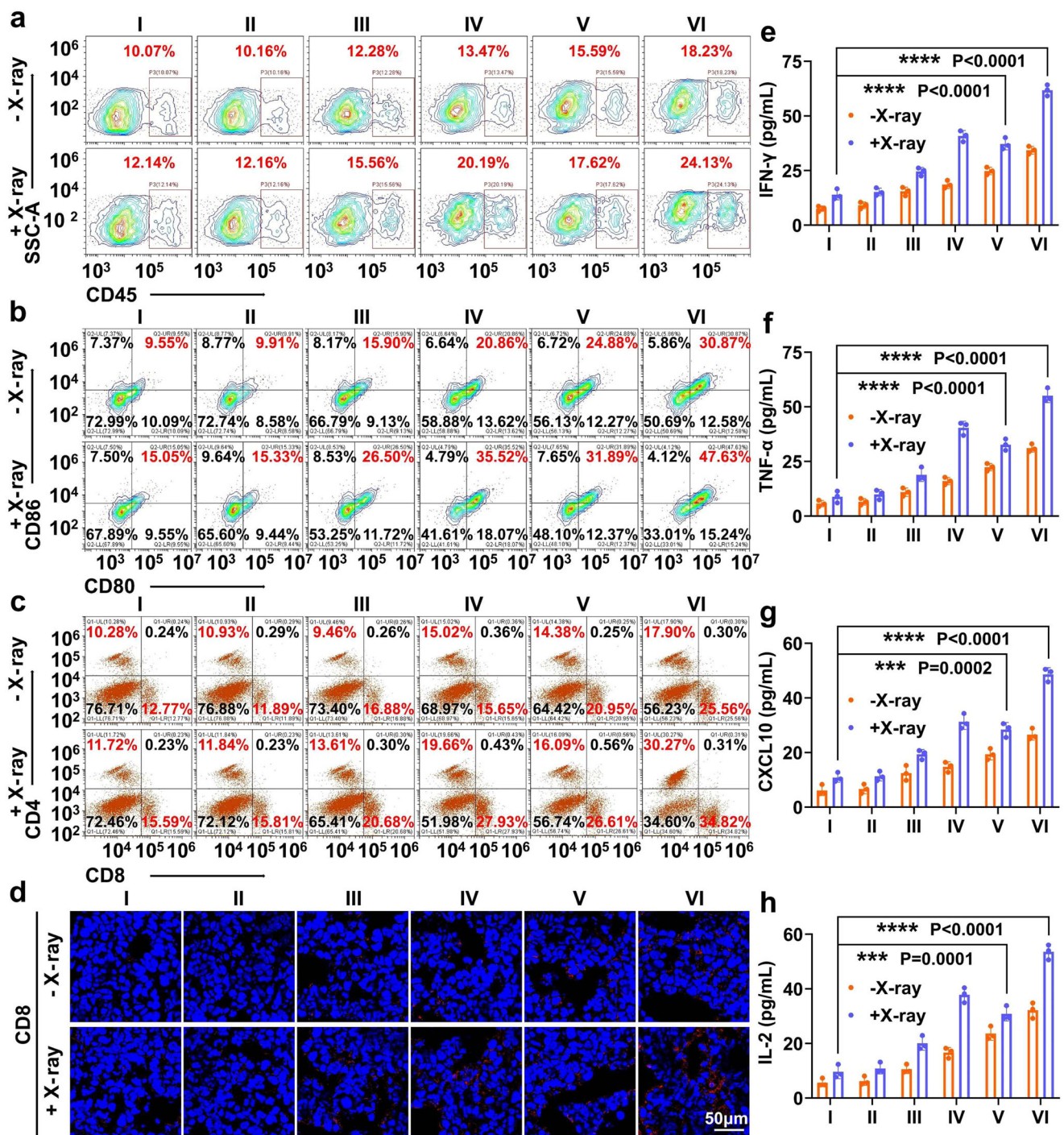

**Fig. 7 | Mechanism underlying Lip@AUR-ACP-aptPD-L1-augmented radio-immunotherapy in vivo. a–c** Flow cytometry analysis on the infiltration of total immune cells (CD45+), DCs (CD11c + CD80 + CD86+), and effector T cells (CD3 + CD4 + CD8+) at the tumor site after different groups treatment with five mice per group. I: PBS, II: Lip, III: Lip-aptPD-L1, IV: Lip-ACP-aptPD-L1, V: Lip@AUR-aptPD-L1, VI: Lip@AUR-ACP-aptPD-L1. **d** Immunofluorescence images of the extracted tumors showing infiltration of CD8+ T cells after different groups treatment with five mice per group. I: PBS, II: Lip, III: Lip-aptPD-L1, IV: Lip-ACP-aptPD-L1, V:

Lip@AUR-aptPD-L1, VI: Lip@AUR-ACP-aptPD-L1. **e–h** The secretion levels of IFN-γ, TNF-α, CXCL10, and IL-2 in mouse serum after treatment with I: PBS, II: Lip, III: Lip-aptPD-L1, IV: Lip-ACP-aptPD-L1, V: Lip@AUR-aptPD-L1, VI: Lip@AUR-ACP-aptPD-L1. Data are presented as mean values ± SEM (n = 3 mice for (**e–h**)). Statistical analysis in (**e–h**) was carried out via one-way ANOVA method. * indicates significance at $p < 0.05$, ** indicates significance at $p < 0.01$, *** indicates significance at $p < 0.001$, **** indicates significance at $p < 0.0001$. Source data are provided as a Source Data file.

## Discussion

In summary, we have developed melanoma-targeted fusogenic liposomal nanoformulations integrated with AUR and multivariate-gated aptamer assemblies for programmable sequential radio immunotherapy against melanomas. The liposomes could efficiently bind with PD-L1-overexpressing melanoma cells for rapid membrane fusion, which not only could allow the ACP assemblies to locate on the melanoma cell surface but also targeted AUR delivery. The gold-containing AUR could sensitize melanoma cells to incoming IR and facilitate their ICD even under a low IR dose of 4 Gy. This strategy allows the effective

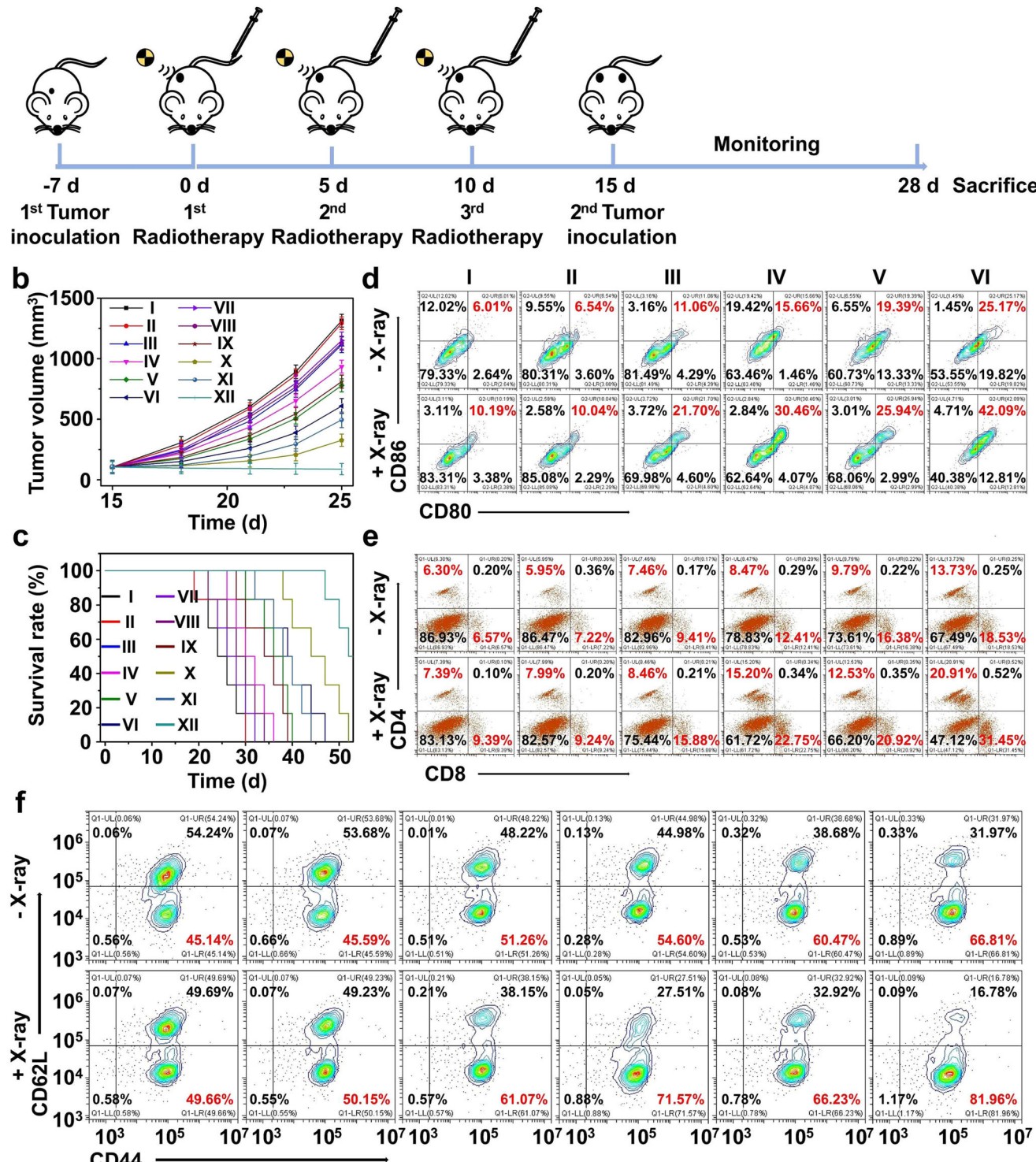

**Fig. 8 | Systemic anti-tumor immunity of Lip@AUR-ACP-aptPD-L1 to suppress distal B16F10 tumors. a** Schematic diagram of the treatment schedule for bilateral B16F10 tumor model. **b** Statistical analysis of distal B16F10 tumor volume during treatment. I: PBS, II: Lip, III: Lip-aptPD-L1, IV: Lip-ACP-aptPD-L1, V: Lip@AUR-aptPD-L1, VI: Lip@AUR-ACP-aptPD-L1, VII: PBS + IR, VIII: Lip + IR, IX: Lip-aptPD-L1 + IR, X: Lip-ACP-aptPD-L1 + IR, XI: Lip@AUR-aptPD-L1 + IR, XII: Lip@AUR-ACP-aptPD-L1 + IR. **c** Survival analysis of bilateral B16F10 tumor model-bearing mice. I: PBS, II: Lip, III: Lip-aptPD-L1, IV: Lip-ACP-aptPD-L1, V: Lip@AUR-aptPD-L1, VI: Lip@AUR-ACP-aptPD-L1, VII: PBS + IR, VIII: Lip+IR, IX: Lip-aptPD-L1 + IR, X: Lip-ACP-aptPD-L1 + IR, XI: Lip@AUR-aptPD-L1 + IR, XII: Lip@AUR-ACP-aptPD-L1 + IR. **d–f** Flow cytometry analysis on the infiltration levels of DCs (CD11c + CD80 + CD86+), effector T cells (CD3 + CD4 + CD8+), and memory CD8+ T cells (CD8 + CD44 + CD62L-) within the distal tumors after different groups treatment with three mice per group. I: PBS, II: Lip, III: Lip-aptPD-L1, IV: Lip-ACP-aptPD-L1, V: Lip@AUR-aptPD-L1, VI: Lip@AUR-ACP-aptPD-L1. Data are presented as mean values ± SEM (*n* = 3 mice for (**b**), *n* = 6 mice for (**c**)). Source data are provided as a Source Data file.

stimulation of melanoma immunogenicity while avoiding common IR-associated side effects such as collateral tissue damage or impairment of immune systems. The melanoma-specific sensitized radiotherapy would also trigger the release of abundant ATP as well as upregulate MMP-2 expression in the TME, which would induce the AND-gate activation of the ACP assembly to trigger eCpG for stimulating DCs maturation in a sequential manner, further expanding the tumor-infiltrating anti-tumor T cell populations for mounting potent adaptive immune responses. Meanwhile, the released AUR contents could also inhibit tumor-intrinsic ERK1/2/HIF-1α/VEGF pathway to suppress the migration of immunosuppressive cells into post-IR melanoma and thus maintain an anti-tumor tumor microenvironment. It is important to note that the nano-enabled programmable radio-immunotherapy could not only efficiently abolish melanoma growth but also orchestrate robust antitumor immune memory, which is beneficial for preventing melanoma metastasis or local relapse.

Our study reports that Lip@AUR-ACP-aptPD-L1 could substantially enhance the radio-immunotherapeutic efficacy against melanomas. However, more studies on melanoma samples from real-life patients are necessary to understand its pharmacological characteristics and general applicability, which will be the subjects of future research. Firstly, although we have demonstrated that Lip@AUR-ACP-aptPD-L1 liposomes can be anchored onto melanoma cell membranes for sufficiently long time to potentiate enhanced radio-immunotherapeutic potency, it is worth mentioning the delivery of the liposomes on real-life patients is a highly complex and dynamic process, and rigorous investigations of the bio-nanointeractions of the liposomes under clinically relevant conditions could substantially facilitate the optimization of treatment schedule. Secondly, it is important to note that phenotypic heterogeneity is a hallmark of solid tumors such as melanoma. Despite the potent melanoma inhibition efficacy of the Lip@AUR-ACP-aptPD-L1-enhanced radio-immunotherapy in vitro and in vivo, its efficacy against those minor melanoma cell populations with intrinsically low PD-L1 expression and the potential mechanisms are worth investigation in follow-up research.

The variations in the cell membrane dynamics, pharmacokinetic characteristics, and IR treatment parameters are major considerations regarding the potential clinical translation of the proposed Lip@AUR-ACP-aptPD-L1 systems for radio-immunotherapy against melanomas. Firstly, the fusion between the liposomes and melanoma cell membrane and the subsequent anchoring of ACP assemblies in real-life patients are profoundly affected by the molecular dynamics of cell membranes such as lipid composition, symmetry, and cytoskeleton organization, which could affect the scalability of the proposed therapeutic strategy and thus requires the personalized optimization of the liposome composition and treatment schedule to prolong the membrane retention of the anchored ACP constructs. Meanwhile, although the ACP constructs are assembled through the autonomous complexation between individual components, the potential competition from endogenous biomolecules remains to be investigated that may attenuate its tumor-specific biochemical reprogramming and TME remodeling efficacy, which could be solved through the rational optimization of the inter-component binding segments. Finally, the IR dose in this study was significantly lower than the ablative dose in the clinical setting, which may require further optimization for achieving balanced immunostimulatory effect, tumor cell damage, and safety on melanoma patients. Addressing these considerations may substantially enhance the clinical relevance of the radio-immunotherapeutic strategy in the present study.

## Methods

All studies in this research were conducted in accordance with relevant experimental and ethical regulations. C57BL/6J (female, 6-week-old) mice were purchased from Second Affiliated Hospital (Xinqiao Hospital) of the Army Medical University and housed in their animal center, of which the Laboratory Animal Production License Number was SCXK (Chongqing) 2022-0011. The experimental plans and procedures have been reviewed and approved by the Laboratory Animal Welfare and Ethics Committee of Army Medical University. The Laboratory Animal Use License Number of the animal experiment in this study is SYXK (Chongqing) 2022-0018.

### Materials

1,2-Dimyristoyl-sn-glycero-3-phosphocholine (DMPC), distearoyl phosphoethanola-mine-PEG$_{2000}$ (DSPE-PEG$_{2000}$), 1,2-dioleoyl-3-tri-methylam-monium-propane (DOTAP) were purchased from Meryer (Shanghai, China). Chloroform (CHCl$_3$) was purchased from Aladdin (Shanghai, China). Auranofin (AUR) was purchased from TargetMol (Shanghai, China). DNase 1 was purchased from Sangon (Shanghai, China). Peptide nucleic acid (PNA) was purchased from Tahepna (Hangzhou, China). Adenosine triphosphate (ATP) was purchased from Solarbio (Beijing, China). Recombinant matrix metalloproteinase-2 (MMP-2) was purchased from MedChemExpress (Shanghai, China). Annexin V-FITC apoptosis detection kit, DAPI, and TUNEL detection kit were purchased from Beyotime (Shanghai, China). ELISA kit of IFN-γ, TNF-α, CXCL-10, and granzyme were purchased from ABclonal (Wuhan, China). Red blood cell lysis buffer and Glycerol anhydrous were obtained from Solarbio (Beijing, China). DNA was purchased from Sangon (Shanghai, China), and the corresponding sequence information was provided in Supplementary Table 1. The information of antibodies was provided in Supplementary Tables 2, 3.

### Cell lines and animals

B16F10, NIH3T3, B16F10-luc cell lines were purchased from Shanghai Zeye Biotechnology Co., Ltd. with the catalog number of ZY-C6002M, ZY-C6050M, ZY-C6002M-L, respectively. The obtained cell lines were authenticated by Hoechst DNA staining, agar culture, and PCR-based assay with no signs of mycoplasma contamination. No commonly misidentified cell lines were used in this study.

C57BL/6J mice were housed in cages with five mice per cage and kept on in a regular 12 h light: 12 h dark cycle (9:00–21:00; 21:00–9:00). The temperature was $22 \pm 1\,°C$ and humidity was 40%-68%. All animal tests were carried out following the Animal Management Rules of the Ministry of Health of the People's Republic of China. According to the national and institutional guidelines, the maximum tumor size allowed was 2000 mm$^3$, and mice were euthanized when the tumor burden exceeded the threshold. B16F10-luc tumor cells ($1 \times 10^6$ cells) were injected subcutaneously into 6-week-old mice to establish B16F10-luc tumor C57BL/6J mouse model. The mice were cultured continuously until the tumor size reached 100 mm$^3$ and the body weight of mice was maintained at $17.5 \pm 0.3$ g.

### Synthesis of Lip@AUR

7.255 mg DMPC, 1.516 mg DSPE-PEG$_{2000}$, 1.963 mg DOTAP, 2 mg AUR were added into a clean 500 mL single-neck flask and dissolved by adding 33 mL chloroform, stirred, and ultrasonicated for 5 min. Lipid film was obtained by rotary evaporation at $26.7 \times g$ and 40 °C in a water bath overnight. The lipid membranes were rehydrated using 10 mL sterile PBS and ultrasonicated for 30 min. Impurities or aggregates were removed by centrifugation at $1000 \times g$ for 10 min. The liposomes were filtered through 0.22 μm membrane and repeatedly extruded by an extruder for about 10 times, followed by dialysis with an MWCO of 1000 Da for 2 days to obtain Lip@AUR.

### Construction of aptATP/eCpG/PNA (ACP) assembly

Moderate amount of DEPC water was added to solubilize the synthesized aptATP, eCpG, and PNA powder at 100 μM. aptATP, eCpG, PNA solutions were placed in clean 1.5 mL EP tubes. aptATP and eCpG samples were heat in 95°C oil bath for 10 min and then mixed in the

ratio of aptATP:eCpG=2:1, followed by further incubation in the oven at 42 °C for 1 h. PNA was added to aptATP/eCpG at the ratio of aptATP:PNA = 1:1.5 and heated in oil bath at 80–90 °C for 10 min. ACP assembly was obtained after incubating in oven at 42 °C for 1 h.

### Synthesis of Lip@AUR-ACP-aptPD-L1

Firstly, Lip@AUR was refrigerated at −80 °C and then freeze-dried in a freeze dryer to obtain liposome powder. The powder was rehydrated by DEPC water and mixed with ACP assemblies with the molarity ratio of lipid: aptATP = 80:2, and incubated in the oven at 37 °C for 4 h. aptPD-L1 powder was resuspended with DEPC water at 100 μM, and then aptPD-L1 was added at the molar ratio of lipid: aptATP: aptPD-L1 = 80:2:1. AptPD-L1 was incubated with Lip@AUR-ACP overnight in a 37 °C oven to obtain Lip@AUR-ACP-aptPD-L1. The product was frozen at −80 °C and then freeze-dried to obtain Lip@AUR-ACP-aptPD-L1 powder.

### DNA-PAGE analysis regarding aptamer binding and release

The formulation of 20% PAGE solution is as follows: 6.666 mL 30% acrylamide, 1 mL 10 × TBE buffer, 2.3 μL DEPC water, 50 μL 10% APS, 5 μL TEMED. After solidification, the corresponding samples were added to each hole and then electrophoresis was carried out at 140 V constant voltage. After electrophoresis, 0.29 g NaCl was dissolved in 50 mL deionized water and mixed with 5 μL GelRed. The gel was soaked in GelRed solution for 30 min and then taken out for observation with a gel imaging system.

### Characterization of liposome sample series

Lip, Lip@AUR, Lip@AUR-ACP, and Lip@AUR-ACP-aptPD-L1 were synthesized according to the above experimental procedure. The hydrodynamic size, polydispersity index, and zeta potential of these liposomes were detected by DLS. The concentration of AUR was measured by ICP and the relative encapsulation amount and encapsulation efficiency of AUR were measured via standard curve calibration.

### Loading and releasing of AUR

Firstly, 2% Triton X-100 solution was prepared with PBS, while 1 mg Lip@AUR-ACP-aptPD-L1 powder was dissolved in 1 mL PBS to afford Lip@AUR-ACP-aptPD-L1 solution. 100 μL Lip@AUR-ACP-aptPD-L1 solution was added into 900 μL 2% Triton X-100 solution and incubated at 37 °C for 1 h to lyse the liposomes and release AUR. The AUR release was detected by fluorescence spectrophotometer and quantified via standard curve calibration.

### The release of eCpG

For ease of understanding, Cy5 labeled eCpG was denoted as $eCpG^{Cy5}$, while the molecular complex of $eCpG^{Cy5}$ and aptATP was denoted as $AC^{Cy5}$. After the complementary binding with PNA, the aptamer assembly was denoted as $AC^{Cy5}P$. Finally, the aptamer-based ligands were inserted into liposomal membrane to afford Lip@AUR-$AC^{Cy5}$-aptPD-L1 or Lip@AUR-$AC^{Cy5}P$-aptPD-L1. Lip@AUR-$AC^{Cy5}$-aptPD-L1, Lip@AUR-$AC^{Cy5}P$-aptPD-L1, Lip@AUR-$AC^{Cy5}P$-aptPD-L1 + MMP-2 (5 nM) and Lip@AUR-$AC^{Cy5}P$-aptPD-L1 + MMP-2 (10 nM) groups were treated with ATP and then centrifuged under 1666.7 × g for 10 min to extract the supernatant. The release of $eCpG^{Cy5}$ was measured via fluorescence spectroscopy.

### Evaluation of ACP assembly construction

The standard curves of Cy5 labeled eCpG and PNA were obtained by fluorescence spectroscopy. $AC^{Cy5}$ and $ACP^{Cy5}$ were assembled with 4 nM aptATP, 2 nM eCpG, and 6 nM PNA and then filtered using an ultrafiltration tube (MWCO: 8000). Finally, the fluorescence intensity of Cy5 was detected by fluorescence spectroscopy.

### eCpG release patterns from Lip@AUR-ACP-aptPD-L1

The standard curves of Cy5 labeled eCpG was obtained by fluorescence spectroscopy. Lip@AUR-$AC^{Cy5}P$-aptPD-L1 was assembled with 4 nM aptATP, 2 nM eCpG, and 6 nM PNA, followed by incubation with 200 nM ATP and 10 nM MMP-2. The supernatant was obtained at different time points using an ultrafiltration tube (MWCO: 8000), afterwards the fluorescence intensity of Cy5 in the supernatant was detected by fluorescence spectroscopy.

### Loading analysis of eCpG and aptPD-L1

The synthesis of fluorescently labeled liposomes was generally the same with those fluorescence-free ones except that the original aptamers were replaced by Cy5-labeled eCpG or FAM-labeled aptPD-L1, leading to the formation of Lip@AUR-$AC^{Cy5}P$-aptPD-L1 or Lip@AUR-ACP-aptPD-L1$^{FAM}$. 56 μL Lip@AUR-$AC^{Cy5}P$-aptPD-L1 (5 mg·mL$^{-1}$) or Lip@AUR-ACP-aptPD-L1$^{FAM}$ (5 mg·mL$^{-1}$) aqueous solution was added to 1 × DNase 1 buffer solution and then treated with 20 U·mL$^{-1}$ DNase 1, incubated at 37 °C for 15 min and transferred to an ultrafiltration tube. After centrifugation at 3333.3 × g for 15 min, the supernatant was collected and fluorescence intensity of Cy5 or FAM was detected by fluorescence spectroscopy. $eCpG^{Cy5}$ or aptPD-L1$^{FAM}$ solution with different concentrations were configured to establish the standard curves via a fluorescence spectrophotometer. The aptamer concentrations in Lip@AUR-$AC^{Cy5}P$-aptPD-L1 or Lip@AUR-ACP-aptPD-L1$^{FAM}$ was quantified according to the standard curve, and then the load efficiency of $eCpG^{Cy5}$ or aptPD-L1$^{FAM}$ on liposomes was calculated accordingly.

### DNA degradation assay of Lip@AUR-$AC^{Cy5}P$-aptPD-L1

Lip@AUR-$AC^{Cy5}P$-aptPD-L1 was synthesized and placed in the PBS with 10% FBS or DNase 1. The supernatant was obtained at different time points using an ultrafiltration tube (MWCO: 5000), afterwards the fluorescence intensity of eCpG in the supernatant was detected by a fluorescence spectrophotometer.

### Morphological characterization of Lip@AUR-ACP-aptPD-L1

5 μL of Lip@AUR-ACP-aptPD-L1 solution was dropped on the carbon support film and dried naturally. Then the film was re-dyed with 4% phosphotungstic acid solution for 3 times (10 min each time) to observe its morphology with a transmission electron microscope.

### Cell culture

Mouse-derived melanoma cell line B16F10 was cultured in RPMI 1640 medium containing 10% fetal bovine serum (Gibco), penicillin (100 μg·mL$^{-1}$), and streptomycin (100 μg·mL$^{-1}$). Mouse embryonic fibroblasts NIH3T3 and B16F10-luc cell lines were cultured in high-glucose DMEM medium containing 10% fetal bovine serum (Gibco), penicillin (100 μg·mL$^{-1}$), and streptomycin (100 μg·mL$^{-1}$). The cells were cultured in a 37 °C constant temperature incubator containing 5% carbon dioxide.

### Extraction of splenocytes from C57BL/6J mice

Surgical tools were sterilized for 30 min by ultraviolet light on ultra-clean workbench. C57BL/6J mice were sacrificed and treated with 75% alcohol for 10 min. The murine spleens were dissected in cell strainer, which was placed into a six-well plate containing RPMI 1640 medium. The spleen was pulverized with the tip of the suction head of a sterile 5 mL syringe, and the strainer was removed after grinding until no obvious spleen tissue was found on the filter. The cells collected from the six-well plate were homogenized and transferred to a centrifuge tube, centrifuged at 666.7 × g for 5 min. The supernatant was discarded, the red blood cell lysate was added and mixed for 10 min, and the lysis was terminated by adding 7 times the volume of PBS. After centrifugation at 666.7 × g for 5 min, cells were collected.

## Toxicity evaluation of Lip@AUR-aptPD-L1

B16F10 cells or NIH3T3 cells were inoculated into 24-well plates with a density of $5 \times 10^4$ cells per well. When the cell confluence reached 80%, the different concentrations of Lip@AUR-aptPD-L1 were added into the above cells. After 30 h incubation, the cells were added with 500 μL serum-free fresh medium containing MTT reagent (0.5 mg·mL$^{-1}$) for 4 h in the dark. Afterwards, 300 μL DMSO was added into each well and homogenized, 100 μL of the added DMSO was extracted from each well for analysis. The OD values of the sample were measured at the wavelength of 490 nm using SpectraMax i3x microplate reader.

The cells were incubated with medium containing 40 μg·mL$^{-1}$ Lip-aptPD-L1 or Lip@AUR-aptPD-L1. After 12 h incubation, the cells were treated with 0, 2, 4, 8 or 16 Gy IR. After 24 h incubation, cell viability was tested via MTT assay as described above.

After placing spleen cells in the 12-well plate at a concentration of $1 \times 10^6$ per well, the cells were incubated with medium containing 40 μg·mL$^{-1}$ Lip@AUR-aptPD-L1. After 12 h incubation, the cells were treated with 0, 2, 4, 8, or 16 Gy IR. After 24 h incubation, 10% CCK-8 was added with the above cells for 2 h at 37 °C. The OD values of the samples were measured at the wavelength of 450 nm using a SpectraMax i3x microplate reader.

Co-culture system was constructed with the B16F10: splenocyte ratio of 1:10 and incubated with fresh medium containing different concentrations of Lip@AUR-aptPD-L1. After 12 h incubation, the cells were treated with 4 Gy IR. After 30 h incubation, spleen cells were removed with PBS and the viability of B16F10 cells was tested via MTT assay as described above.

## Toxicity evaluation of liposomes

The co-incubation system of B16F10 cells and mouse splenocytes was treated with different concentrations of Lip, Lip-aptPD-L1, Lip-ACP-aptPD-L1, Lip@AUR-aptPD-L1 or Lip@AUR-ACP-aptPD-L1. After 12 h incubation, the cells were treated with 4 Gy IR. After 30 h incubation, spleen cells were removed with PBS and the viability of B16F10 cells was tested via MTT assay as described above.

## Flow cytometric analysis on the receptor binding effect of aptPD-L1 and eCpG

B16F10 cells were mixed with splenocytes at a ratio of 1:10 and transferred to a 1.5 mL centrifuge tube. 170 nM aptPD-L1$^{FAM}$ and 360 nM eCpG$^{FAM}$ were added and incubated for 30 min with 5% BSA, followed by the addition of the corresponding antibodies (Entry 2 and 12, Supplementary Table 2) for 30 min incubation after washing with PBS. Flow cytometry was used to detect the binding status of aptPD-L1$^{FAM}$ and eCpG$^{FAM}$.

## Observation of membrane fusion behavior of the liposomes

Herein, orange-red probe Dil was loaded into the liposome instead of AUR for fluorescence tracking, of which the samples were denoted as Lip@Dil, Lip@Dil-ACP, and Lip@Dil-ACP-aptPD-L1. B16F10 cells were inoculated into confocal dishes at a density of $1 \times 10^5$ cells/well. The establishment of co-incubation system was the same as above. 40 μg·mL$^{-1}$ Lip@Dil, Lip@Dil-ACP or Lip@Dil-ACP-aptPD-L1 was added respectively into the co-culture system. At 3, 6, 12, and 18 h of incubation, cell samples were washed with PBS to remove the floating spleen cells, while the remaining B16F10 cells were first stained with Invitrogen CellMask™ Green plasma membrane stain for 15 min and then by DAPI for 10 min. Finally, the membrane fusion status of the liposomes with B16F10 cells was detected by laser confocal microscopy.

The co-incubation system of B16F10 cells and mouse splenocytes was treated with 40 μg·mL$^{-1}$ Lip@Dil, Lip@Dil-ACP or Lip@Dil-ACP-aptPD-L1. After 12 h incubation, the above cells were treated with 4 Gy IR. At 16, 18, and 30 h of incubation, while the remaining B16F10 cells were first stained with Invitrogen CellMask™ Green plasma membrane stain for 15 min and then by DAPI for 10 min. Finally, the membrane fusion status of the liposomes with B16F10 cells was detected by laser confocal microscopy.

## B16F10 tumor sphere assay for testing targeting effect

90 mg agarose gel was dissolved in 6 mL serum-free RPMI 1640 medium and sterilized at 115 °C for 30 min. 80 μL of the melted gel was added into sterile 96-well plates and cooled down naturally for solidification. The B16F10 cells were homogenized in RPMI 1640 medium containing 2.5% matrix gel and added into the wells at 5000 cells per well, of which the volume was 100 μL per well. The cells were cultured for about 7 days until pellets were formed under an optical microscope. AC$^{Cy5}$P, Lip-AC$^{Cy5}$P or Lip-AC$^{Cy5}$P-aptPD-L1 was added and incubated for 12 h, then cells were detached, centrifuged at $233.3 \times g$ for 5 min to remove matrix gel, cleaned with PBS for 3 times, and transferred to the confocal dish for detection by laser confocal microscopy.

## Evaluation of IR-mediated impact on ATP and MMP-2 expression in vivo

$1 \times 10^6$ B16F10 cells were injected subcutaneously into 6-week-old mice to establish B16F10-bearing C57BL/6J mouse model. The mice were cultured continuously until the tumor size reached 100 mm$^3$, then treated with Lip@AUR-ACP-aptPD-L1 (2 mg·kg$^{-1}$) by intravenous injection. The administration time was defined as 0 h. The mice received 4 Gy IR treatment at 12 h post intravenous injection. The tumors were dissected at 12, 14, 16, 18, 30, and 36 h post injection, then ground to the powder using liquid nitrogen, and then lysed with RIPA lysate (containing 1% PMSF) for 30 min at 4 °C and centrifuged at $4000 \times g$ for 10 min. The collected supernatant was used for detect the concentrations of ATP and MMP-2 in each tumor by the relevant kits.

## Evaluation of ATP and MMP-2 in tumors

B16F10-bearing C57BL/6J mouse models were treated with PBS, Lip, Lip@AUR or Lip@AUR-aptPD-L1 (2 mg·kg$^{-1}$), followed by 4 Gy IR at 12 h post intravenous injection. After 30 h post intravenous injection, the tumors were dissected and ground to the powder using liquid nitrogen, and then lysed with RIPA lysate (containing 1% PMSF) for 30 min at 4 °C and centrifuged at $4000 \times g$ for 10 min. The collected supernatant was used for detect the concentrations of ATP and MMP-2 in each tumor by the relevant kits.

## Observation of membrane fusion behavior of the liposomes in vivo

B16F10-bearing C57BL/6J mouse models were treated with Cy5-Lip@AUR-ACP-aptPD-L1 (2 mg·kg$^{-1}$) by intravenous injection. The administration time was defined as 0 h. The mice received 4 Gy IR treatment at 12 h post intravenous injection. The tumors were dissected at 0, 6, 12, 16, and 18 h post-injection, which were pulverized and filtered to extract the cells. The extracted cells were mixed with 5 mL red blood cell lysis buffer and stood for 10 min, and then centrifuged at $666.7 \times g$ for 5 min. Afterward, they were stained with Invitrogen CellMask™ Green plasma membrane stain for 15 min and then with DAPI for 10 min. Finally, the membrane fusion status of liposomes with B16F10 cells was detected by laser confocal microscopy.

## AUR induced secretion of critical DAMPs

B16F10 cells were inoculated into 12-well plates, and the initial cell density was $1 \times 10^5$ cells/well. When the cell confluence reached 80%, B16F10 cells were treated with Lip@AUR-aptPD-L1. After 12 h incubation, B16F10 cells were treated with 4 Gy IR. Then the cells were lysed with RIPA lysate (containing 1% PMSF) for 30 min at 4 °C and centrifuged at $4000 \times g$ for 10 min after 30 h incubation. The secretion of ATP, CRT, and HMGB1 in the collected supernatant was detected by the relevant kits.

## Impact of IR treatment on B16F10-intrinsic PD-L1 expression

After 12 h treatment with $20\,\mu g \cdot mL^{-1}$ AUR, B16F10 cells were treated with different radiation doses including 0 Gy, 2 Gy, 4 Gy, and 8 Gy. After 30 h incubation, the above samples were fixed with 4% paraformaldehyde for 30 min, followed by the addition of anti-PD-L1 antibody (Entry 28, Supplementary Table 2) and incubated at 4 °C overnight. Afterwards, FAM-labeled fluorescent secondary antibody was added and the cell samples were further incubated at room temperature for 2 h. The secondary antibody was removed and the cell nuclei were stained with DAPI for 10 min after washing with PBS. After cleaning, the immunofluorescence of PD-L1 was detected by confocal laser microscopy.

After 12 h treatment with $20\,\mu g \cdot mL^{-1}$ AUR, B16F10 cells were treated with different radiation doses including 0 Gy, 2 Gy, 4 Gy, and 8 Gy. After 30 h incubation, the cells were detached with trypsin and sealed with 5% BSA for 30 min. The cells were incubated with FITC-anti-PD-L1 antibody (Entry 29, Supplementary Table 2) at 4 °C for 30 min. After cleaning, the PD-L1 expression was detected by flow cytometry.

## Validation of AND-gate release of eCpG

B16F10 cells were treated with $40\,\mu g \cdot mL^{-1}$ Lip-AC$^{Cy5}$-aptPD-L1 or Lip-AC$^{Cy5}$P-aptPD-L1 and mixed with 0 nM ATP + 5 nM MMP-2, 100 nM ATP + 5 nM MMP-2, 0 nM ATP + 10 nM MMP-2 or 200 nM ATP + 10 nM MMP-2 for 2 h. The B16F10 cells were washed with PBS and the cell nuclei were stained with DAPI for 10 min. Finally, the Cy5 fluorescence of eCpG was detected by confocal laser microscopy.

For the flow cytometric analysis of eCpG release, B16F10 cells were inoculated into the 1.5 mL centrifuge tube with a density of $5 \times 10^4$ cells per tube. Then $40\,\mu g \cdot mL^{-1}$ Lip-AC$^{Cy5}$-aptPD-L1 or Lip-AC$^{Cy5}$P-aptPD-L1 was added into the above cells and treated with 0 nM ATP + 5 nM MMP-2, 100 nM ATP + 5 nM MMP-2, 0 nM ATP + 10 nM MMP-2 or 200 nM ATP + 10 nM MMP-2 for 2 h. The B16F10 cells were washed with PBS and then the Cy5 fluorescence of eCpG was quantified by flow cytometry.

## Observation of eCpG release in vivo

B16F10-bearing C57BL/6J mouse models were treated with Lip@AUR-AC$^{Cy5}$-aptPD-L1 or Lip@AUR-AC$^{Cy5}$P-aptPD-L1 ($2\,mg \cdot kg^{-1}$) by intravenous injection. The time point of administration was defined as 0 h, while 4 Gy IR treatment was applied at 12 h post intravenous injection. The tumors were extracted after 16 h post intravenous injection, pulverized, and filtered. The above cells were mixed with 5 mL red blood cell lysis buffer and stood for 10 min, and then centrifuged at $666.7 \times g$ for 5 min. Subsequently, the cells were first stained with Invitrogen CellMask™ Green plasma membrane stain for 15 min and then by DAPI for 10 min. After washing with PBS, single B16F10 cell was observed by laser confocal microscopy.

B16F10-bearing C57BL/6J mouse models were treated with Lip@AUR-AC$^{Cy5}$P-aptPD-L1 ($2\,mg \cdot kg^{-1}$) by intravenous injection. The time point of administration was defined as 0 h, while 4 Gy IR treatment was applied at 12 h post intravenous injection. The tumors were extracted at the time points of 0, 6, 12, 18, 24, 30, and 36 h, pulverized and filtered. The above cells were mixed with 5 mL red blood cell lysis buffer and stood for 10 min, and then centrifuged at $666.7 \times g$ for 5 min. Subsequently, the cells were stained with the corresponding antibodies (Entry 1, Supplementary Table 2) for 30 min and washed with PBS. Finally, the Cy5 fluorescence of eCpG was quantified by flow cytometry.

## Analysis of eCpG-mediated stimulation of DC maturation in vitro and in vivo

Lip@AUR-ACP-aptPD-L1 was added into the co-incubation system of B16F10 and mouse splenocytes and incubated for 12 h until 4 Gy IR was applied. At 12, 16, 20, 24, 30, or 36 h post liposome administration, the above cells were collected and added with the corresponding antibodies (Entry 1, 4, 7, and 15, Supplementary Table 2) for 30 min. After washing with PBS, the DC maturation status was detected by flow cytometry.

B16F10-bearing C57BL/6J mouse models were treated with Lip@AUR-ACP-aptPD-L1 ($2\,mg \cdot kg^{-1}$) by intravenous injection. The time point of administration was defined as 0 h, while 4 Gy IR treatment was applied at 12 h post intravenous injection. The tumors were extracted at the time points of 0, 6, 12, 18, 24, 30 and 36 h, pulverized and filtered. The above cells were mixed with 5 mL red blood cell lysis buffer and stood for 10 min, and then centrifuged at $666.7 \times g$ for 5 min. Subsequently, the cells were stained with the corresponding antibodies (Entry 1, 4, 7, and 15, Supplementary Table 2) for 30 min. After washing with PBS, the DC maturation status was detected by flow cytometry.

## Transcriptome sequencing and protein expression evaluation

The co-incubation system of B16F10 cells and mouse splenocytes was treated with Lip@AUR-aptPD-L1 or Lip@AUR-aptPD-L1 + 4 Gy IR, afterwards the melanoma cells were extracted and sent to Sangon Biotech (Shanghai) Co., LTD for detection. For the WB assay, the co-incubation system was treated with PBS, Lip, Lip-aptPD-L1, Lip-ACP-aptPD-L1, Lip@AUR-aptPD-L1, Lip@AUR-ACP-aptPD-L1. After 12 h incubation, the co-culture system was treated with 4 Gy IR. After 30 h incubation, the cells were lysed with RIPA lysate (containing 1% PMSF) for 30 min at 4 °C and centrifuged at $4000 \times g$ for 10 min. The collected supernatant was quantified by BCA method. Finally, the expression of corresponding proteins (Entry 20–25, Supplementary Table 2) was observed by 12% SDS-polyacrylamide gel electrophoresis.

## Evaluation on the impact of VEGF on anti-tumor immunity

B16F10 cells were inoculated into the 12-well plate at the concentration of $1 \times 10^5$ per well. When the cell confluence reached 80%, the cells were treated with PBS, Lip, Lip@AUR or Lip@AUR-aptPD-L1, the upper chamber is placed into 12-well plate. Splenocytes were added into the upper chamber with B16F10: splenocyte ratio of 1:10. After 12 h incubation, the IR groups were treated with 4 Gy IR. After 30 h incubation, the cells in the upper chamber were discarded and the bottom chamber supernatant was collected. After centrifugation at $666.7 \times g$ for 5 min, 200 μL PBS was added to each tube to resuspend the spleen immune cells. The corresponding antibodies (Entry 1, 6, 13, 14, 16, and 19, Supplementary Table 2) were added into each tube. Finally, the infiltration of Tregs or MDSCs in the bottom chamber was detected by flow cytometry.

Alternatively, the recovered cell samples in the bottom chamber were treated with the corresponding antibodies (Entry 1, 3, 4, 5, 7, 11, and 15, Supplementary Table 2). Finally, the infiltration of effector T cells or DCs was detected by flow cytometry.

The B16F10 tumor-bearing mouse model was constructed and treated with PBS, Lip, Lip@AUR or Lip@AUR-aptPD-L1 ($2\,mg \cdot kg^{-1}$) by intravenous injection and treated with 4 Gy IR after 12 h post intravenous injection. After 30 h post intravenous injection, the tumors were collected from each group after treatment and pulverized to collect various cell populations. 200 μL PBS was added to each tube to suspend tumor cells. The corresponding antibodies (Entry 1, 6, 13, 14, 16, and 19, Supplementary Table 2) were added into each tube. Finally, the infiltration of Tregs or MDSCs in tumor tissues was detected by flow cytometry.

## Evaluation of HIF-1α by confocal laser microscopy in vitro

B16F10 cells were inoculated into the confocal dish, and the initial cell density was $1 \times 10^5$ cells/dish. After 12 h treatment with PBS, Lip, Lip-aptPD-L1, Lip-ACP-aptPD-L1, Lip@AUR-aptPD-L1 or Lip@AUR-ACP-aptPD-L1, the IR groups were treated with 4 Gy IR. After further incubation for 30 h, cells in all groups were fixed with 4% paraformaldehyde for 30 min, followed by the addition of anti-HIF-1α antibody, and incubated at 4 °C overnight. Subsequently, Cy5-labeled fluorescent

secondary antibody was added, followed by incubation at room temperature for 2 h. The secondary antibody was removed and the cell nuclei were stained with DAPI for 10 min after washing with PBS. After cleaning, the immunofluorescence of HIF-1α (Entry 21, Supplementary Table 2) was detected by confocal laser microscopy.

### Evaluation of treatment-induced ICD of melanoma cells in vitro
The co-incubation system of B16F10 cells and mouse splenocytes were treated with PBS, Lip, Lip-aptPD-L1, Lip-ACP-aptPD-L1, Lip@AUR-aptPD-L1 or Lip@AUR-ACP-aptPD-L1 for 12 h. The IR groups were treated with 4 Gy IR. After 30 h incubation, the cells were washed with PBS multiple times to remove floating splenocytes, while the remaining B16F10 cells were fixed with 4% paraformaldehyde for 30 min. Subsequently, anti-HMGB1 antibody or anti-CRT antibody was added and incubated at 4 °C overnight, followed by Cy5-labeled or FITC-labeled fluorescent secondary antibody, afterwards the B16F10 cells were further incubated at room temperature for 2 h. The secondary antibody was removed and the cell nuclei were stained with DAPI for 10 min after washing with PBS. After cleaning, the immuno-fluorescence of HMGB1 or CRT (Entry 26-27, Supplementary Table 2) was detected by confocal laser microscopy.

### Evaluation of treatment-induced immunoactivation in vitro
Splenocytes of C57BL/6J mice were extracted and DCs were sorted out according to the above method. B16F10 cells were inoculated into 12-well plates with the initial cell density of $1 \times 10^5$ cells/well. When the cell confluence reached 80%, mouse DCs were added into 12-well plates and co-cultured with B16F10 cells at a ratio of B16F10:DC = 1:10. After 12 h treatment with PBS, Lip, Lip-aptPD-L1, Lip-ACP-aptPD-L1, Lip@AUR-aptPD-L1 or Lip@AUR-ACP-aptPD-L1, the IR groups were treated with 4 Gy IR. After 30 h incubation, DCs were collected via centrifugation, resuspended with 200 μL PBS, and then incubated with the corresponding antibodies (Entry 1, 4, 7, and 15, Supplementary Table 2) for 30 min. Finally, the treatment-induced stimulation effect on DCs maturation in each group was detected by flow cytometry.

After B16F10 cells were inoculated into the 12-well plate through the procedure described above, mouse splenocytes were added into the 12-well plate and co-cultured with B16F10 cells at the B16F10:splenocyte ratio of 1:10. After 12 h treatment with PBS, Lip, Lip-aptPD-L1, Lip-ACP-aptPD-L1, Lip@AUR-aptPD-L1 or Lip@AUR-ACP-aptPD-L1, the IR groups were treated with 4 Gy IR. After 30 h incubation, spleen cells and supernatants were collected for later use. Here the spleen cells were suspended with 200 μL PBS, then the corresponding antibodies (Entry 1, 3, 5, 8 and 11, Supplementary Table 2) were added to each tube. Finally, the activation status of T cells in each group was detected by flow cytometry. The secretion level of TNF-α, IL-2, IFN-γ, CXCL10, IL-4, IL-10, and TGF-β in the supernatant was detected by ELISA kit according the relevant specification.

### Detection of tumor cell apoptosis
The co-incubation system of B16F10 cells and mouse splenocytes were treated with PBS, Lip, Lip-aptPD-L1, Lip-ACP-aptPD-L1, Lip@AUR-aptPD-L1 or Lip@AUR-ACP-aptPD-L1 for 12 h. The IR groups were treated with 4 Gy IR. After 30 h incubation, all cells were collected and suspended with 200 μL FITC bonding solution at 37 °C for 30 min, then followed by PI dye solution for 10 min. After extensive staining, the corresponding antibodies (Entry 2, Supplementary Table 2) were added to each tube, then apoptosis of tumor cells under different treatments was detected by flow cytometry.

The co-incubation system of B16F10 cells and mouse splenocytes were treated with PBS, Lip, Lip-aptPD-L1, Lip-ACP-aptPD-L1, Lip@AUR-aptPD-L1 or Lip@AUR-ACP-aptPD-L1 for 12 h. The IR groups were treated with 4 Gy IR. After 30 h incubation, the cells were washed with PBS for 3 times and splenocytes were immediately drained. Then the cells were fixed with 4% paraformaldehyde for 30 min, blocked with 5% BSA for 30 min after cleaning, and permeabilized with 0.5% Triton X-100 solution for 5 min after cleaning with PBS. Then γ-H2AX antibody was added and incubated at 4 °C overnight. The primary antibody was removed, and Cy3-labeled fluorescent secondary antibody was added after purification, followed by the incubation at room temperature for another 2 h. The secondary antibody was removed and the cell nuclei were stained with DAPI for 10 min after washing with PBS. After cleaning, the cell samples were mounted on glass slides with glycerin and the immunofluorescence of γ-H2AX was detected by confocal laser microscopy.

### Blood circulation stability of different samples
C57BL/6J mice were randomly selected and intravenously injected with AUR, Lip@AUR, or Lip@AUR-aptPD-L1 (2 mg·kg$^{-1}$) with three mice per group. Then tail venous blood was collected according to the scheduled time point, and AUR content in samples of each group was detected by HPLC.

### ICP-dependent blood distribution analysis
Three groups of B16F10-bearing C57BL/6J mice were randomly selected and intravenously injected with AUR, Lip@AUR, or Lip@AUR-aptPD-L1 (2 mg·kg$^{-1}$) with three mice per group. The mice in each group were euthanized at predetermined time points to collect major organs and tumors were collected, and the supernatant was collected after grinding and cracking for 24 h. The samples were filled to 5 mL with deionized water, and the AUR concentration in each tissue was detected by ICP.

### Anti-tumor evaluation of the liposomes in vivo
The B16F10 tumor-bearing mice were randomly divided into 12 groups with 5 animals in each group, which were subjected to intravenous injection of PBS (100 μL) containing Lip, Lip-aptPD-L1, Lip-ACP-aptPD-L1, Lip@AUR-aptPD-L1 or Lip@AUR-ACP-aptPD-L1 (2 mg·kg$^{-1}$), and the same volume of fresh PBS was administered as the control group. After 12 h post intravenous injection, the IR groups were treated with 4 Gy IR. Treatment was performed once every 5 days for a total of 15 days. Bioluminescence imaging was performed every 5 days, and 20 μL (7.5 mg·mL$^{-1}$) luciferase was injected into the intraperitoneal cavity of mice. After anesthesia with isoflurane, tumor volume of each group was detected by IVIS imaging system. The tumor volume and body weight of mice were recorded by electronic balance and vernier caliper. The volume and size of the tumor were measured every 2 days, and the longitudinal and transverse diameters of the tumor were measured. The calculation formula was $V = 1/2*A*B^2$ (A was the longitudinal diameter, B was the transverse diameter). After 15 days of treatment, serums of all tumor mice were collected, and tumor tissues and major organs were collected for subsequent analysis. A parallel set of animal models were established, and the survival of mice in each group was observed until the 50th day after the 15-day treatment with six mice per group.

At the end of treatment, the tumors in each group were dissected, and the tumors were pulverized after freezing with liquid nitrogen, and then the cells were disintegrated by tip ultrasonication. The grinded tumors were treated with cell lysis solution on ice, and Western blot assay was carried out to detect the expression levels of related proteins in the tumor (Entry 20–25, Supplementary Table 2). Paraffin sections of tumor and heart, liver, spleen, lung, and kidney were created for optical imaging after H&E staining. The tumor was dissected and cleaned with PBS, and further cut into thin sections for TUNEL staining, CD4/IFN-γ immunofluorescence staining (Entry 8 and 11, Supplementary Table 2), CRT/HMGB1 immunofluorescence staining (Entry 26–27, Supplementary Table 2) and γ-H2AX immunofluorescence staining (Entry 24, Supplementary Table 2) which were observed by confocal laser microscopy.

The tumors were pulverized and treated with red cell lysate for 15 min, afterwards, T cells were sorted and isolated by the corresponding antibodies (Entry 3, Supplementary Table 2) and added with 10% CCK8 agent. After incubation at 37 °C for 1.5 h, the OD value was measured at 450 nm using the plate reader.

The tumor was pulverized and treated with red cell lysate for 15 min, followed by the incubation with the corresponding antibodies (Entry 1, 2, 3, 4, 5, 8, 7, 10, 11, and 15, Supplementary Table 2), then the infiltration of immune cells was detected by flow cytometry. The secretion level of IFN-γ, TNF-α, CXCL10, IL-2, IL-4, IL-10, and TGF-β in collected blood samples was detected using ELISA kits.

### Establishment and treatment of bilateral tumor model in C57BL/6J mice

$1 \times 10^6$ B16F10 cells were injected subcutaneously into the right flank of C57BL/6J mice to establish B16F10 tumor-bearing mice. They were cultured in the same way as above and divided into groups with three mice per group, and intravenously injected with 100 μL PBS, Lip, Lip-aptPD-L1, Lip-ACP-aptPD-L1, Lip@AUR-aptPD-L1 or Lip@AUR-ACP-aptPD-L1 (2 mg·mL$^{-1}$). After 15 days of treatment, secondary tumors were established by subcutaneous injection of $2 \times 10^6$ B16F10 cells on the left flank. The growth of distal tumor was monitored from the 15th day, and the treatment ended on the 25th day. Bilateral tumors were dissected for analysis. In addition, a batch of bilateral tumor models were established. After 15 days of treatment, the survival of mice in each group was observed for up to 50 days with six mice per group. The primary and distal tumors were dissected, cleaned with PBS, pulverized, and treated with erythrocyte lysate for detection. Cells in the primary tumors in each group were labeled with the corresponding antibodies (Entry 1, 3, 4, 5, 7, 9, 11, 15, and 17, Supplementary Table 2), then the infiltration of immune cells was detected by flow cytometry.

### Evaluation of IR doses or IR fractionations on the therapeutic effect in vivo

The B16F10 tumor-bearing C57BL/6J mice were randomly divided into 8 groups with 3 mice in each group, which were intravenously injected with 100 μL of Lip@AUR-ACP-aptPD-L1 (2 mg·kg$^{-1}$), while the same volume of fresh PBS was administered as the control group. At 12 h post intravenous injection, the IR groups were treated with 0 Gy, 2 Gy, 4 Gy, or 8 Gy IR, the IR treatment was performed three times with a 4-day interval for a total of 15 days. In addition, at 12 h post intravenous injection, mice in the IR groups were treated with 4 Gy IR for 0 time, 1 time (at the start of the treatment period), 3 times (with a 4-day interval), and 5 times (with a 2-day interval) through the 15-day treatment period. The tumor was pulverized and treated with red cell lysate for 15 min, followed by the incubation with the corresponding antibodies (Entry 1, 3, 4, 5, 6, 10, 11, 13, 14, 16, and 19, Supplementary Table 2). Subsequently, the infiltration of immune cells was detected by flow cytometry.

### Statistics and reproducibility

All measurements contained three or more independent replicates from separate experiments. The exact sample size and statistical test for each experiment are described in the relevant figure legends. All statistical analysis results are presented as mean ± standard error (S.E.M.). All statistical data were processed in GraphPad Prism (version 9.5 for Windows) by Student's one-way ANOVA. * indicates significance at $p < 0.05$, ** indicates significance at $p < 0.01$, *** indicates significance at $p < 0.001$, **** indicates significance at $p < 0.0001$. Sample size and the statistical method were showed in the figure legends.

### Reporting summary

Further information on research design is available in the Nature Portfolio Reporting Summary linked to this article.

## Data availability

The RNA-sequencing data for Lip@AUR-aptPD-L1 and Lip@AUR-aptPD-L1 + 4 Gy IR-treated melanoma cells in the present study were generated by Sangon (Shanghai, China) and deposited in NCBI under the accession code of PRJNA1100325, which can be viewed through the link. Source data are provided with this paper. All remaining data can be found in the Article, Supplementary and Source data Files. Source data are provided with this paper.

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

## Acknowledgements

We would like to thank the Analytical and Testing Centre of Chongqing University for their assistance during sample characterization. We would like to thank the Analytical and Testing Centre of Chongqing University for their assistance during sample characterization. This study is financially supported by National Natural Science Foundation of China (32122048 (Z.L.), 11832008 (Z.L.), 92059107 (Z.L.), 51825302 (Z.L.) and 82272755 (Y.L.)), Chongqing Science and Technology Commission (cstc2021ycjh-bgzxm0124 (Z.L.) and 2022NSCQ-MSX0706 (Y.L.)), Graduate Research and Innovation Foundation of Chongqing (CYB23016 (X.J.R.)), Fundamental Research Funds for the Central Universities (2022CDJYGRH-007 (Z.L.)) and Natural Science Foundation of Chongqing Municipal Government (cstb2022nscq-msx0488 (Z.L.)).

## Author contributions

F.D.W., M.H.L., and Z.L. conceptualized this study and supervised the experiments. F.D.W., X.J.R., and R.X. designed the experiments., X.J.R., R.X., and Y.L. performed experiments. X.J.R., R.X., Y.L., S.W., X.G., X.Y., L.Q.L., J.X.M. analyzed and interpreted the data. M.H.L., X.J.R., and R.X. wrote the paper. M.H.L., X.J.R., R.X., and Y.L. helped with the revision of the draft.

## Competing interests

The authors declare no competing interests.

## Additional information

[1]Key Laboratory of Biorheological Science and Technology, Ministry of Education, Chongqing University, Chongqing 400044, PR China. [2]School of Life Science, Chongqing University, Chongqing 400044, PR China. [3]Radiation Oncology Center, Chongqing University Cancer Hospital, Chongqing 400030, PR China. [4]Department of General Surgery, Xinqiao Hospital, Army Medical University, Chongqing 400037, PR China. [5]The Second Affiliated Hospital, The First Affiliated Hospital School of Public Health Institute of Translational Medicine State Key Laboratory of Experimental Hematology, Zhejiang University School of Medicine, Hangzhou 310058, PR China. [6]The First Affiliated Hospital Basic Medical Sciences, School of Public Health Hengyang Medical School University of South China, Hengyang 421001, PR China. [7]These authors contributed equally: Xijiao Ren, Rui Xue, Yan Luo. ✉e-mail: menghuanli@cqu.edu.cn; luozhong918@cqu.edu.cn; fwang@zju.edu.cn

