## [Peer Review File · Nature Communications]

REVIEWER COMMENTS

Reviewer #1 (Remarks to the Author):

Radioimmunotherapy has emerged as a gold standard for treating numerous metastatic and advanced solid tumors. However, many tumor-related biochemical and immunological features can seriously impair the efficacy of radioimmunotherapy in clinic, such as radiation dose deposition and immunosuppressive tumor microenvironments. This manuscript by Xue et al. cleverly developed a nanointegrated strategy with tumor-specific radiosensitization and anti-tumorigenic tumor microenvironment (TME) reprogramming capabilities to enhance the efficacy of radioimmunotherapy based on a rationally designed multifunctional fusogenic liposomal formulation (Lip@AUR-ACP-aptPD-L1). Targeted delivery of auranofin (AUR) enhances ionizing radiation (IR) dose deposition in melanoma cells to amplify radiation-mediated immunogenic death of tumor cells for triggering the AND-gate activation of ACP assembly while suppressing the tumor cell-mediated recruitment of immunosuppressor cells. These merits could improve the cancer immune cycle after IR therapy in a cascading manner. My specific comments are listed here below.

Major issues:

1. Hypoxia-inducible factor 1 α (HIF-1 α) is an essential upstream regulator in the AUR-mediated TME-remodeling effect, but the associated characterizations are insufficient. The authors should monitor HIF-1 α expression via more intuitive methods, such as immunofluorescence imaging.
2. Given the upregulated expression of programmed death ligand 1 (PD-L1) on B16F10 cell membrane after radiotherapy, the authors should provide relevant experiment data in addition to the reference support.
3. The authors only used flow cytometry to test the enhancement of DC and CD4⁺/CD8⁺ T cell infiltration after AUR-mediated inhibition of Tregs and myeloid-derived suppressor cells (MDSCs) in vivo. The reviewer suggests that the authors may consider employing other intuitive techniques to monitor the changes in the post-treatment immune composition of the tumor tissues, such as CD4⁺CD8⁺ T cell migration assay.
4. In Fig. 5, the secretion of IFN- γ , TNF- α , CXCL10, and IL-2 in the supernatant of the co-culture system was detected, but the secretion of immunosuppressive markers such as IL-10, TGF- β , and IL-4 in the supernatant is missing, which is needed to elucidate the capacity of the nanoformulation to reverse the "immunosuppressive state of solid tumors".
5. The authors have tested the immunofluorescence of typical DAMPs, such as HMGB1 and CRT in vivo, and they should also measure their release state in the in vitro co-culture system of tumor cells and splenic immune cells.
6. The recently published review or research articles should be discussed in the revision.

Minor issues

1. Fig. 6g is not entirely displayed. Please correct this by reorganizing the panels.
2. The label of treatment groups is inconsistent in several places, such as "PBS+IR" or "PBS+RT". Please unify the group labeling to avoid misunderstanding.
3. The text in Fig. 3f is unclear and needs adjustment.
4. The horizontal axis in Fig. 8f is not correctly displayed. Please adjust the figure layout.

Reviewer #2 (Remarks to the Author):

The authors reported multifunctional fusogenic liposomes for enhanced radio-immunotherapy with multivariate-gated aptamer assemblies. They demonstrated the programmable mode of action in vitro and observed combined therapeutic effects in vivo. However, experimental supports are not sufficient to verify their concept, particularly about "cascade-amplifying manner", and the "requirement of logic-gate". Due to complicated manufacturing process, potential for clinical translation of this therapeutic tool is also a big concern. On the one hand, overall readability is quite poor. More coherent and structured arrangement of figures and detailed descriptions about figures are required to present the key findings for readers. Based on the above-mentioned substantial issues, I cannot recommend publishing the manuscript in Nature Communications.

Specific comments for authors:

- 1) In introduction references for membrane fusion ability of liposomal nanostructure should be provided. More backgrounds that can support the advantage and necessity of multivariate logic-gated aptamer should be addressed. Abbreviation aptATP, aptPD-L1 in 4th paragraph of introduction first appears without explanation
- 2) The content of fig 2 seems disorganized and does not effectively convey the main message of the figure. Fig 2 appears to be overly focused on the characterization of engineered aptamers, compared to its title. The figures are inconsistently arranged and lack coherence.
 - In fig 2a, description about the components are not well matched with figure and some descriptions are missed. Those mismatches make it hard to figure out the scheme. (ex. Basic color doesn't match, cholesterol modification of ACP missed, aptATP, eCpG color change after responses, and etc.)
 - Fig 2b, c could be better suited for inclusion in the extended data section. Instead, it would be more beneficial to add additional data related to liposome characterization rather than focusing solely on aptamer modification results. Results in the extended data are not enough to provide physicochemical properties of liposome. Providing a summary table of the physicochemical properties of each liposomal formulation, including hydrodynamic size, polydispersity index (PDI), zeta potential, encapsulation efficiency (EE), and other relevant properties, would be valuable.

- Why there is same groups in figure 2g, h (all + groups)

- Figures from d-j should be reorganized, since figure 2f, h are data from liposome treated samples, but other figures are data from vehicle-free treated samples.

3) The content of fig 3 does not provide proper explanations about each figure for readers and some experimental supports are not sufficient and should be improved to verify their results.

- In fig 3a, details about FACS plot (marker for each fluorescence, or gating strategy) should be clearly described. Instead of gating plot, quantitative analysis of intensity of fluorescence should be provided as authors mentioned “~ 5-fold higher than splenocytes) Also authors mentioned “we developed a co-culture system comprising B16F10 cells and mouse splenocytes and ~” before starting description about fig 3b. But aren't these mentioned passage in the main text should appear before fig 3a to provide the explanation? Or are fig 3b from co-culture system?

- In fig 3b, why are they showing green signal? And also compared to group III, group II (fusogenic liposome) looks like their Dil signal is rather trapped in lysosome instead membrane. It would be better to show enlarged confocal image. Based on other studies about membrane fusogenic liposome, precise charge interaction between plasma membrane thought to be most crucial for fusion. Doesn't aptamer incorporation affect those charge interactions? Or PD-L1 binding induce ligand mediated endocytosis rather than fusion? Even though authors demonstrate the fusion ability of liposome by confocal image, more supportive results to prove that modification of liposomes does not change fusogenic property should be addressed.

- Based on the data of fig 3b, authors designed the time interval for in vivo administration, but it is not reasonable to set the in vivo administration condition based on those results, with same liposomal formulations, in vivo membrane retention kinetics analysis could be available.

- In fig 3g, description about group is not provided.

4) Fig 4, 5 provides overall immunostimulatory effects of developed liposome and mechanisms for those effects. However, there are some insufficient demonstrations about results since results are mainly based on co-culture system, and also more clear time-dependent studies should be addressed to support these liposomes will act similarly in vivo.

- In fig 4g, authors evaluate the effect of PmP binding on the stability of complexed eCpG. To confirm the role of PmP, authors should provide the increased MMP-2 expression after IR in prior.

- Also, those cascades suggested by authors will occur after (1) PD-L1 mediated interaction, and (2) IR irradiation. Then ATPs will be released as AUR&IR-mediated ICD occurs. In this situation, ATPs can actively bound to aptATP regardless of MMP-2 expression. I wonder if there are any reasons for the requirement of AND-gate.

- Also, as mentioned above, those cascades suggested by authors will be expected to occur step by step, and temporal control of those responses are most important issue for in vivo experiments. In vitro data from fig 4, 5 seems to be presenting only the results-oriented data from cascades. More supportive data about developed liposome works in temporal manner should be addressed.

- Authors summarized the figure 4 in text as “~and stimulate the adaptive antitumor immune response in IR-treated melanomas”, but based on results of figure 4, we cannot get any information about adaptive immune response.

- Authors summarized the figure 5 in text as “~through cascade-amplifiable manner” and also the title of manuscript emphasizes the cascade amplification. In what points, authors demonstrate the amplifiable manner? Based on the mechanism study, it seems to happen in programmable sequential manner, not the amplifiable manner. Clear demonstration or strong experimental supports should be provided to argue the cascade-amplification.

5) For results of in vivo experiments, figures with multiple groups make it hard to interpret which group should readers focus. Why groups without X-ray results are included in all the figures? Are there any reasons readers should go parallel with – X-ray groups?

Reviewer #3 (Remarks to the Author):

This manuscript reports a cascade-amplification strategy based on multivariate-aptamer assembly modified liposomes (Lip@AUR-ACP-aptPD-L1) to enhance the radio-immunotherapy efficiency of melanoma. In multivariate aptamer assembly, PD-L1 aptamers guide the targeted fusion with tumor cell and delivery AUR to induce immunogenic death of melanoma cells; the released ATP and upregulated MMP2 by damaged cells cause AND-gate activation of ACP, thus triggering the in-situ release of CpG-based immunoadjuvants for stimulating dendritic cell-mediated T cell priming; AUR also inhibit tumor-intrinsic ERK1/2-HIF-1 α -VEGF signaling to suppress infiltration of immunosuppressive cells for fostering an anti-tumorigenic TME. The cascade amplification and synergistic effects of multivariate liposome probe provided a promising tool for the clinical treatment of melanoma. Overall, it should be suitable for publication at Nat. Commun. after major revision.

The major concerns about the study were listed as following:

1. The stability of DNA assembly is a major obstacle for immunotherapy application. The Lip@AUR-ACP-aptPD-L1 probes were incubated in cells and within bodies for more than ten hours. How to guarantee the intactness and stability of DNA probes, under complicated enzyme conditions?
2. The characterization of multivariate aptamer assembly, such as the construction of Lip@AUR-ACP-aptPD-L1, the cascading sequence release and amplification, was based on PAGE assay, which is not clear and quantitative. We will suggest further characterization using fluorescent labeled DNA aptamer or linker sequence, to monitor the multivariate aptamer assembly and quantitatively analyze the release efficiency of each aptamer units.
3. The loading efficiency of AUR in liposomes was calculated to ~5%, which is extremely low. Have the authors attempted to optimize the concentration of AUR and increase the loading efficiency? How to achieve high radio-immunotherapy efficiency with such less AUR?

4. The uptake of liposomes into cells is commonly existed. How to demonstrate the targeted fusion guided by PD-L1 aptamer? Figure 3b is not sufficient. A quantitative and face-to-face comparison with (targeted fusion) and without (random fusion) PD-L1 aptamer should be conducted.

5. The caption of each figure is too simple. Some necessary description and explanation should be added to assist the reading of figures.

6. The manuscript should be double checked thoroughly, as there are lots of writing errors and typos, such as Line 486 "immunofluorescence", the format of number and unit such as 20nm.

Reviewer #4 (Remarks to the Author):

The authors presented a paper about "Cascade-amplification of melanoma-targeted radioimmunotherapy via fusogenic liposomes functionalized with multivariate-gated aptamer assemblies".

The topic is absolutely interesting and it totally deserves attention both a clinical and research point of view.

The background provided by the authors is that immunosuppressive microenvironment of solid tumors may limit the efficacy of this combination and therefore they propose a strategy to inhibit tumor-intrinsic signaling pathway to suppress immunosuppressive cells.

The idea itself sounds definitively interesting and the description of the multivariate-gated aptamer assembly-modified AUR-loaded fusogenic liposome as an adjuvant for melanoma-targeted radioimmunotherapy is clear and detailed.

Several molecular insights of the process are in fact provided and make the article scientifically sound.

However, I believe that since the ultimate goal of the authors was to provide some useful proposals to be evaluated in the clinical practice it would be an addition to introduce within the manuscript some up-to-date evidence about the use of combination therapy between immunotherapy and radiotherapy (see PMID: 33847208 for a comprehensive dissertation); in fact by adding more data coming from already published clinical studies would broaden the interest of the article to a wider audience. For example in the clinical practice radiotherapy can be delivered concomitantly to immunotherapy for melanoma either at the beginning of a treatment, Peri-Induction Radiotherapy (PIR) or after there has been an immunological escape with a clinical progression of the disease, Post-Escape Radiotherapy (PER). Also the role of doses and fractionations should be addressed more in detail.

Reviewer #5 (Remarks to the Author):

The study conducted by Xue and colleagues introduces a promising and innovative approach to combat melanoma, integrating targeted liposomal delivery, radiation therapy, and immune system activation. This multifaceted strategy holds significant potential for enhancing treatment effectiveness while minimizing patient risks. The research demonstrates meticulous planning and skillful execution of experiments, yielding compelling results. Nevertheless, to bolster the manuscript's comprehensibility and scientific rigor, it is imperative to furnish more comprehensive details, particularly within the methodology sections covering both in vitro and in vivo experiments. A more expansive methodological exposition would empower readers with a clearer grasp of the experimental protocols, facilitating replication and critical evaluation of the study's validity. Furthermore, improving the informativeness of figure legends is essential, as it would provide succinct yet vital explanations of the figures' content and their implications. Especially in the flow cytometry figures, as many of them are lacking either the marker or fluorochrome used. These refinements are crucial for enhancing the overall quality and impact of this research.

Figure 2a. Please provide the marker for each fluorochrome. It is not clear the gating strategy used in the figure.

Figure 2b. Please add information about what the green and red colors represent. It is hard to understand the findings in the figure.

How does the combined treatment of Lip@AUR-aptPD-L1 liposomes and 4Gy IR affect immune cells in extended data Figure 8?

Extended Figure 19. What is the concentration about?

The authors state that they found CD62L+CD44+ cells in the melanoma tissue samples. The memory T cells should be mentioned as CD62L-(low) and CD44+, figure 8f shows this cell population correctly but the text and figure legend are wrong.

Reviewer #1 (Remarks to the Author):

Radioimmunotherapy has emerged as a gold standard for treating numerous metastatic and advanced solid tumors. However, many tumor-related biochemical and immunological features can seriously impair the efficacy of radioimmunotherapy in clinic, such as radiation dose deposition and immunosuppressive tumor microenvironments. This manuscript by Xue et al. cleverly developed a nanointegrated strategy with tumor-specific radiosensitization and anti-tumorigenic tumor microenvironment (TME) reprogramming capabilities to enhance the efficacy of radioimmunotherapy based on a rationally designed multifunctional fusogenic liposomal formulation (Lip@AUR-ACP-aptPD-L1). Targeted delivery of auranofin (AUR) enhances ionizing radiation (IR) dose deposition in melanoma cells to amplify radiation-mediated immunogenic death of tumor cells for triggering the AND-gate activation of ACP assembly while suppressing the tumor cell-mediated recruitment of immunosuppressor cells. These merits could improve the cancer immune cycle after IR therapy in a cascading manner. My specific comments are listed here below.

A: Thank you for the appreciation of our work. Based on the comments and suggestions you kindly provided, we have carried out substantial amount of additional work to address your concerns. We ensure that all the issues you kindly pointed out have been properly addressed in the revised manuscript, and the detailed responses are shown here below for your review:

Major issues:

1. Hypoxia-inducible factor 1 α (HIF-1 α) is an essential upstream regulator in the AUR-mediated TME-remodeling effect, but the associated characterizations are insufficient. The authors should monitor HIF-1 α expression via more intuitive methods, such as immunofluorescence imaging.

A: Thank you for the suggestion. We agree with the reviewer that AUR predominantly exerts the immunoregulatory effects through modulating the HIF-1 α pathway as well as the importance of more thorough investigations on the expression status in melanoma cells after various treatments. Based on your advice, we have carried out Western blot and CLSM analysis as suggested on melanoma cells after various treatments to comprehensively elucidate the HIF-1 α therein. Western blot results in Supplementary Figure 16 showed that Lip@AUR-aptPD-L1 and Lip@AUR-ACP-aptPD-L1 treatments substantially reduced HIF-1 α abundance in the nucleus compared with the PBS, Lip, Lip-aptPDL1 and Lip-ACP-aptPDL1 groups, evidently suggesting the AUR-mediated suppression of HIF-1 α expression. In further support of the WB results, we have carried out immunofluorescence imaging on the HIF-1 α expression in B16F10 cells from different groups (Supplementary Figure 17), which revealed that the nucleus HIF-1 α expression in the Lip, Lip-aptPDL1 and Lip-ACP-aptPDL1 groups remained at a comparable level to the PBS only control group, but decreased dramatically in the Lip@AUR-aptPD-L1 and Lip@AUR-ACP-aptPD-L1 groups. These data are immediate evidence that the AUR component in the liposomes could inhibit HIF-1 α expression in melanoma cells, which is conducive for remodeling the TME into an

immunoactivated state.

The related data and discussions are shown here below for your review:

“The WB data on HIF-1 α expression after different treatments were further supported by immunofluorescence imaging, which showed that the Lip@AUR-aptPD-L1 and Lip@AUR-ACP-aptPD-L1 treatments induced evident reduction in the HIF-1 α -intrinsic red fluorescence compared with the other AUR-free modalities, again confirming that AUR component in the liposomes could inhibit HIF-1 α expression in melanoma cells (Supplementary Figure 17).”

“Evaluation of HIF-1 α by confocal laser microscopy *in vitro*

B16F10 cells were inoculated into the confocal dish, and the initial cell density was 1×10^5 cells/dish. After 12 h treatment with PBS, Lip, Lip-aptPD-L1, Lip-ACP-aptPD-L1, Lip@AUR-aptPD-L1 or Lip@AUR-ACP-aptPD-L1, the IR groups were treated with 4 Gy IR. After further incubation for 30 h, cells in all groups were fixed with 4% paraformaldehyde for 30 min, followed by the addition of anti-HIF-1 α antibody and incubated at 4°C overnight. Subsequently, Cy5-labeled fluorescent secondary antibody was added, followed by incubation at room temperature for 2 h. The secondary antibody was removed and the cell nuclei were stained with DAPI for 10 min after washing with PBS. After cleaning, the immunofluorescence of HIF-1 α was detected by confocal laser microscopy.”

Supplementary Figure 16. Western Blot analysis on the biochemical changes in IR-treated B16F10 cells *in vitro*. (a) Western Blot analysis on the expression levels of key proteins related to IR damage and ERK1/2-HIF-1 α -VEGF pathway. (b-c) Statistical analysis of panel a. I: PBS, II: Lip, III: Lip-aptPD-L1, IV: Lip-ACP-aptPD-L1, V: Lip@AUR-aptPD-L1, VI: Lip@AUR-ACP-aptPD-L1.

Supplementary Figure 17. Immunofluorescence analysis on AUR-mediated suppression of HIF-1 α expression. Expression of HIF-1 α in the B16F10 nucleus after different treatment. I: PBS, II: Lip, III: Lip-aptPD-L1, IV: Lip-ACP-aptPD-L1, V: Lip@AUR-aptPD-L1, VI: Lip@AUR-ACP-aptPD-L1.

2. Given the upregulated expression of programmed death ligand 1 (PD-L1) on B16F10 cell membrane after radiotherapy, the authors should provide relevant experiment data in addition to the reference support.

A: Thank you for the advice. We are grateful for your suggestions and carry out additional analyses to investigate the PD-L1 expression status of B16F10 cells after IR treatment at graded doses. Immunofluorescence imaging of PD-L1 with fluorescently-labeled antibodies revealed a general positive correlation between IR dose and PD-L1 expression, which was in accordance with the observations in previous reports^{1,2,3} and confirmed the promotional effect of IR on PD-L1 expression. The visual trends above were further analyzed quantitatively using flow cytometry, which revealed that the PD-L1 abundance in the AUR+8 Gy group was more than 9-fold higher than the AUR+0 Gy and PBS+0 Gy groups. These data collectively supported the capacity of IR to upregulate PD-L1 expression in a dose-dependent manner, necessitating the incorporation of additional immunostimulatory modalities to compensate the potential negative impact on post-IR immune responses.

The related data and discussions are shown here below for your review:

“Immunofluorescence imaging of PD-L1 with fluorescently-labeled antibodies

revealed a general positive correlation between IR dose and PD-L1 expression, which was in accordance with the observations in previous reports^{62, 63, 64} and confirmed the promotional effect of IR on PD-L1 expression (Supplementary Figure S9). The visual trends above were further analyzed quantitatively using flow cytometry, which revealed that the PD-L1 abundance in the AUR+8 Gy group was more than 9-fold higher than the AUR+0 Gy and PBS+0 Gy groups. These data collectively supported the capacity of IR to upregulate PD-L1 expression in a dose-dependent manner, necessitating the incorporation of additional immunostimulatory modalities to compensate the potential negative impact on post-IR immune responses.”

“Impact of IR treatment on B16F10-intrinsic PD-L1 expression

B16F10 cells were inoculated into the confocal dish at the initial cell density was 1×10^5 cells/dish. After 12 h treatment with 20 $\mu\text{g}/\text{mL}$ AUR, the cells were treated with different radiation doses including 0 Gy, 2 Gy, 4 Gy and 8 Gy. After 30 h incubation, the above samples were fixed with 4% paraformaldehyde for 30 min, followed by the addition of anti-PD-L1 antibody and incubated at 4°C overnight. Afterwards, FAM-labeled fluorescent secondary antibody was added and the cell samples were further incubated at room temperature for 2 h. The secondary antibody was removed and the cell nuclei were stained with DAPI for 10 min after washing with PBS. After cleaning, the immunofluorescence of PD-L1 was detected by confocal laser microscopy.

B16F10 cells were inoculated into the 24-well plate at a density of 5×10^4 cells per well. After 12 h treatment with 20 $\mu\text{g}/\text{mL}$ AUR, the cells were treated with different radiation doses including 0 Gy, 2 Gy, 4 Gy and 8 Gy. After 30 h incubation, the cells were

detached with trypsin and sealed with 5% BSA for 30 min. The cells were incubated with FITC-anti-PD-L1 antibody at 4°C for 30 min. After cleaning, the PD-L1 expression was detected by flow cytometry.”

Supplementary Figure 9. Dose-dependent impact of IR on B16F10-intrinsic PD-L1 expression. (a) Confocal microscopic analysis of PD-L1 expression in AUR-treated B16F10 cells with graded IR doses. (b) Flow cytometry analysis on the expression status of PD-L1 in B16F10 cells after different treatment. I: PBS+0 Gy, II: PBS+2 Gy, III: PBS+4 Gy, IV: PBS+8 Gy, V: AUR+0 Gy, VI: AUR+2 Gy, VII: AUR+4 Gy, VIII: AUR+8 Gy.

3. The authors only used flow cytometry to test the enhancement of DC and CD4⁺/CD8⁺ T cell infiltration after AUR-mediated inhibition of Tregs and myeloid-derived suppressor cells (MDSCs) in vivo. The reviewer suggests that the authors may consider employing other intuitive techniques to monitor the changes in the post-treatment immune composition of the tumor tissues, such as CD4⁺CD8⁺ T cell migration assay.

A: Thank you for the advice. We agree with the necessity to investigate the post-

treatment immunocomposition in the melanoma tissues and carried out the in vivo analysis as suggested, during which the extracted tumors were lysed to isolate the T cell population. CCK8 assay on the isolated T cell population immediately revealed that the combinational Lip@AUR-ACP-aptPD-L1+IR treatment induced a substantial increase in the total T cell infiltration into melanoma tissues, which was more than 50% higher than the PBS only control group. The quantitative CCK8 analysis confirmed that the combined treatment of the liposomes and radiotherapy could facilitate T cell infiltration into the melanoma tissues, which is conducive for boosting the eventual immunotherapeutic potency.

The related data and discussions are shown here below for your review:

“The flow cytometric results regarding the tumor infiltration status of various immune cell populations were also consistently supported by the immunofluorescence assay based on relevant markers and CCK8 assay of T cells (Figure 7d and Supplementary Figure 36-Supplementary Figure 37).”

“The tumors were pulverized and treated with red cell lysate for 15 min, afterwards T cells were sorted and isolated by APC-anti-CD3 antibody and added with 10% CCK8 agent. After incubation at 37°C for 1.5 h, the OD value was measured at 450 nm using the plate reader.”

Supplementary Figure 37. Detection of CD3+T cell infiltration in B16F10 tumors treated with different materials by CCK8 assay. The group set-ups include I: PBS, II: Lip, III: Lip-aptPD-L1, IV: Lip-ACP-aptPD-L1, V: Lip@AUR-aptPD-L1, VI: Lip@AUR-ACP-aptPD-L1.

4. In Figure 5, the secretion of IFN- γ , TNF- α , CXCL10, and IL-2 in the supernatant of the co-culture system was detected, but the secretion of immunosuppressive markers such as IL-10, TGF- β , and IL-4 in the supernatant is missing, which is needed to elucidate the capacity of the nanoformulation to reverse the "immunosuppressive state of solid tumors".

A: Thank you for the advice. Based on your suggestions we have systematically measured the secretion levels of the mentioned immunosuppressive markers after various treatment both in vitro and in vivo via ELISA test. Remarkably, the secretion levels of IL-4, IL-10 and TGF- β in the supernatant of the B16F10/splenic immune cell co-incubation system decreased by 72%, 77% and 75% after the combined treatment of Lip@AUR-ACP-aptPD-L1+IR compared with the control group (Supplementary Figure 24). Similar trends were also observed on the extracted murine serum samples,

where the serum levels of IL-4, IL-10 and TGF- β in the Lip@AUR-ACP-aptPD-L1+IR group were 25%, 24% and 24% lower than the control group, thus supporting the immunostimulatory effect of the liposomes.

The related data and discussions are shown here below for your review:

“In addition, the secretion of key immune-related molecular markers in the co-incubation system was analyzed by ELISA assay to indicate the alteration in the immune composition, and the results revealed that the secretion levels of pro-inflammatory cytokines and chemokines including IFN- γ (Figure 5c), TNF- α (Figure 5d), CXCL10 (Figure 5e) and IL-2 (Figure 5f) in the Lip@AUR-ACP-aptPD-L1+IR group were the highest among all groups, which have increased by 4.2-fold, 6.3-fold, 4.3-fold and 5.5-fold compared to the control group, respectively. In contrast, the secretion levels of anti-inflammatory cytokines including IL-4 (Supplementary Figure 24a), IL-10 (Supplementary Figure 24b) and TGF- β (Supplementary Figure 24c) in the Lip@AUR-ACP-aptPD-L1+IR group were the lowest among all groups, which have decreased by 72%, 77% and 75% compared with the control group, respectively.”

“Extending from the treatment-induced changes in the immunocomposition of the melanoma tissues, tumors in the Lip@AUR-ACP-aptPD-L1+IR group showed the highest enhancement in the secretion levels of pro-inflammatory cytokines and chemokines including IFN- γ (4.4-fold), TNF- α (6.2-fold), CXCL10 (4.5-fold) and IL-2 (5.6-fold) as well as the greatest reduction in the secretion levels of anti-inflammatory cytokines including IL-4 (25%), IL-10 (24%) and TGF- β (24%), indicating that the liposome-augmented radio-immunotherapy has significantly boosted the adaptive

immune responses for eliminating the melanoma cells in vivo (Figure 7e-h and Supplementary Figure 38).”

“After B16F10 cells were inoculated into the 12-well plate through the procedure described above, mouse splenocytes were added into the 12-well plate and co-cultured with B16F10 cells at the B16F10: splenocyte ratio of 1:10. After 12 h treatment with PBS, Lip, Lip-aptPD-L1, Lip-ACP-aptPD-L1, Lip@AUR-aptPD-L1 or Lip@AUR-ACP-aptPD-L1, the IR groups were treated with 4 Gy IR. After 30 h incubation, spleen cells and supernatants were collected for later use. Here the spleen cells were suspended with 200 μ L PBS, 1 μ L PC7-anti-CD45 antibodies, 1 μ L APC-anti-CD3 antibodies, 1 μ L FITC-anti-CD4 antibodies, 1 μ L PE-anti-CD8a antibodies and 1 μ L FITC-anti-IFN- γ antibodies were added to each tube. Finally, the activation status of T cells in each group was detected by flow cytometry. The secretion level of TNF- α , IL-2, IFN- γ , CXCL10, IL-4, IL-10 and TGF- β in the supernatant was detected by ELISA kit according the relevant specification.”

“The tumor was pulverized and treated with red cell lysate for 15 min, followed by the incubation with 1 μ L APC-anti-CD45 antibodies, 1 μ L PC7-anti-CD45 antibodies, 1 μ L APC-anti-CD3 antibodies, 1 μ L FITC-anti-CD4 antibodies, 1 μ L PE-anti-CD8a antibodies, 1 μ L FITC-anti-IFN- γ antibodies, 1 μ L APC-anti-CD11c antibodies, 1 μ L FITC-anti-CD80 antibodies, 1 μ L PE-anti-CD86 antibodies, 1 μ L APC-anti-CD11c and 1 μ L PE-anti-MHC-II antibodies, then the infiltration of immune cells were detected by flow cytometry. The secretion level of IFN- γ , TNF- α , CXCL10, IL-2, IL-4, IL-10 and TGF- β in collected blood samples was detected using ELISA kits.”

Supplementary Figure 24. Evaluation on the treatment-induced changes in the secretion of anti-inflammatory factors. The secretion levels of various anti-inflammatory factors including (a) IL-4, (b) IL-10 and (c) TGF- β after treatment with different materials in the co-culture system. I: PBS, II: Lip, III: Lip-aptPD-L1, IV: Lip-ACP-aptPD-L1, V: Lip@AUR-aptPD-L1, VI: Lip@AUR-ACP-aptPD-L1.

Supplementary Figure 38. Evaluation on the post-treatment secretion levels of anti-inflammatory factors in vivo. Secretion of various anti-inflammatory factors including (a) IL-4, (b) IL-10 and (c) TGF- β after different treatment in B16F10 tumor-bearing mice. I: PBS, II: Lip, III: Lip-aptPD-L1, IV: Lip-ACP-aptPD-L1, V: Lip@AUR-aptPD-L1, VI: Lip@AUR-ACP-aptPD-L1.

5. The authors have tested the immunofluorescence of typical DAMPs, such as HMGB1 and CRT in vivo, and they should also measure their release state in the in vitro co-culture system of tumor cells and splenic immune cells.

A: Thank you for the advice. Based on your suggestion we have carried out the *in vitro* experiment as requested, during which the B16F10/splenocyte co-incubation system was sequentially treated with different samples and IR at 4Gy. Immunofluorescence imaging of the melanoma cells revealed that the combined treatment of Lip@AUR-ACP-aptPD-L1+IR induced an obvious decrease of HMGB1 in the nucleus region while enhancing CRT expression at the cytoplasmic membrane, which were consistent with the *in vivo* observations and confirmed the successful ICD induction on melanoma cells.

The related data and discussions are shown here below for your review:

“The confocal microscope results showed that Lip@AUR-ACP-aptPD-L1+IR group significantly decreased the expression of HMGB1 in the nucleus as well as significantly increased the expression of CRT on the cell membrane (Supplementary Figure 22), indicating that Lip@AUR-ACP-aptPD-L1 combined with 4 Gy IR could successfully induce significant tumor immunogenic death.”

“Evaluation of treatment-induced ICD of melanoma cells *in vitro*

Splenocytes of C57BL/6J mice were extracted. B16F10 cells were inoculated into the confocal dish, and the initial cell density was 1×10^5 cells/dish. When the cell confluence reached 80%, murine splenocytes were added into confocal dish and co-cultured with B16F10 cells at a ratio of B16F10: splenocyte=1:10. After 12 h treatment with PBS, Lip, Lip-aptPD-L1, Lip-ACP-aptPD-L1, Lip@AUR-aptPD-L1 or Lip@AUR-ACP-aptPD-L1, the IR groups were treated with 4 Gy IR. After 30 h incubation, the cells were washed with PBS multiple times to remove floating splenocytes, while the

remaining B16F10 cells were fixed with 4% paraformaldehyde for 30 min. Subsequently, anti-HMGB1 antibody or anti-CRT antibody was added and incubated at 4°C overnight, followed by Cy5-labeled or FITC-labeled fluorescent secondary antibody, afterwards the B16F10 cells were further incubated at room temperature for 2 h. The secondary antibody was removed and the cell nuclei were stained with DAPI for 10 min after washing with PBS. After cleaning, the immunofluorescence of HMGB1 or CRT was detected by CLSM.”

Supplementary Figure 22. Treatment-induced ICD of B16F10 cells in the co-culture system. (a) The HMGB1 expression in the B16F10 cell nucleus (red fluorescence). (b) The CRT expression on cell membrane (green fluorescence). I: PBS, II: Lip, III: Lip-aptPD-L1, IV: Lip-ACP-aptPD-L1, V: Lip@AUR-aptPD-L1, VI: Lip@AUR-ACP-aptPD-L1.

6. The recently published review or research articles should be discussed in the revision.

A: Thank you for the advice. Based on your suggestion we have expanded the introduction section by incorporating recent relevant publications, which we believe have substantially improved the informativity of this study. The associated descriptions and discussions are shown here below for your review:

“Indeed, clinical insights collectively confirm that combining radiotherapy with immunotherapy could convey significant improvement on the overall survival benefit of melanoma patients without inducing obvious side effects. For instance, preconditioning tumors with IR (peri-induction radiotherapy) could activate the immune system and facilitate the recognition and elimination of tumor cells by the sequentially administered immunotherapeutic modalities. On the other hand, there are reports that the local IR treatment of tumors following immunotherapy (post-escape radiotherapy) could potentially remodel the TME to reverse immunoresistance and prevent tumor immune escape^{7, 8, 9}.”

“Indeed, there are already reports that aptamer-based logic-gated nanosystems could convey programmable diagnostic or therapeutic activities, which may substantially improve their controllability and precision in vitro or in vivo^{39, 55, 56, 57}.”

Minor issues

1. Figure 6g is not entirely displayed. Please correct this by reorganizing the panels.

A: Thank you for the suggestion. Based on your advice we have reorganized Figure 6 so that all panels could fit into an A4 page.

Figure 6. Anti-tumor effects of Lip@AUR-ACP-aptPD-L1-augmented radio-

immunotherapy in vivo. (a) Schematic representation of the treatment protocol for B16F10-luc tumor-bearing mice. (b) In vivo bioluminescence images of B16F10-Luc tumor-bearing mice during treatment (n=5). (c) Tumor volume analysis throughout the treatment period (n=5). I: PBS, II: Lip, III: Lip-aptPD-L1, IV: Lip-ACP-aptPD-L1, V: Lip@AUR-aptPD-L1, VI: Lip@AUR-ACP-aptPD-L1, VII: PBS+IR, VIII: Lip+IR, IX: Lip-aptPD-L1+IR, X: Lip-ACP-aptPD-L1+IR, XI: Lip@AUR-aptPD-L1+IR, XII: Lip@AUR-ACP-aptPD-L1+IR. (d) Tumor weight analysis at the end of the treatment period (n=5). I: PBS, II: Lip, III: Lip-aptPD-L1, IV: Lip-ACP-aptPD-L1, V: Lip@AUR-aptPD-L1, VI: Lip@AUR-ACP-aptPD-L1. (e) Survival analysis of mice after different treatments (n=6). I: PBS, II: Lip, III: Lip-aptPD-L1, IV: Lip-ACP-aptPD-L1, V: Lip@AUR-aptPD-L1, VI: Lip@AUR-ACP-aptPD-L1, VII: PBS+IR, VIII: Lip+IR, IX: Lip-aptPD-L1+IR, X: Lip-ACP-aptPD-L1+IR, XI: Lip@AUR-aptPD-L1+IR, XII: Lip@AUR-ACP-aptPD-L1+IR. (f) Western blotting on the expression levels of related proteins in the tumor tissues. (g) The release level of ATP and MMP-2 in B16F10 tumors after different treatment (n=3). I: PBS+IR, II: Lip+IR, III: Lip@AUR+IR, IV: Lip@AUR-aptPD-L1+IR. (h) TUNEL staining of tumor tissue samples after treatment. I: PBS, II: Lip, III: Lip-aptPD-L1, IV: Lip-ACP-aptPD-L1, V: Lip@AUR-aptPD-L1, VI: Lip@AUR-ACP-aptPD-L1.

2. The label of treatment groups is inconsistent in several places, such as "PBS+IR" or "PBS+RT". Please unify the group labeling to avoid misunderstanding.

A: Thank you for the correction. Based on your suggestion we have unified the group

names into “IR” in case that the sample treatment involves ionizing radiation. Some representative changes are shown here below for your review:

“Furthermore, the immunosuppressive TME will substantially impair the T cell-mediated antitumor immunity despite the IR-triggered immunostimulatory effects^{20, 21, 22.}”

“As a result of the efficient AND-gate eCpG release, DCs in the Lip@AUR-ACP-aptPD-L1+IR group showed the highest maturation ratio (CD11c+CD80+CD86+) after 30 h incubation in vitro (Figure 4k), indicating that 4 Gy IR successfully triggered the AND-gate eCpG release to promote DC maturation.”

“Sole IR treatment induced modest inhibition on melanoma growth with a final tumor volume of around 1550 mm³, which was slightly lower than the control group and suggested the innate radiotherapeutic resistance of melanomas (Figure 6b-c).”

3. **The text in Figure 3f is unclear and needs adjustment.**

A: Thank you for the suggestion. Based on your advice we have reedited Figure 3f in the original manuscript by optimizing the figure texts to improve its readability. Due to the increases in images and font size, the panel was moved to SI as Supplementary Figure 14.

Supplementary Figure 14. Bioinformatic analysis of the treatment-induced therapeutic impact. Transcriptome sequencing analysis regarding the impact of combined Lip@AUR-aptPD-L1+ 4 Gy IR treatment on the VEGF pathway in B16F10 cells.

4. The horizontal axis in Figure 8f is not correctly displayed. Please adjust the figure layout.

A: Thank you for the reminder. We are sorry for the careless mistakes and have remade the figure as requested. The corrected figure is shown here below for your review:

Figure 8. Systemic anti-tumor immunity of Lip@AUR-ACP-aptPD-L1 to suppress distal B16F10 tumors. (a) Schematic diagram of the treatment schedule for bilateral B16F10 tumors. (b) Statistical analysis of distal B16F10 tumor volume during treatment (n=3). I: PBS, II: Lip, III: Lip-aptPD-L1, IV: Lip-ACP-aptPD-L1, V: Lip@AUR-aptPD-L1, VI: Lip@AUR-ACP-aptPD-L1, VII: PBS+IR, VIII: Lip+IR, IX: Lip-aptPD-L1+IR, X: Lip-ACP-aptPD-L1+IR, XI: Lip@AUR-aptPD-L1+IR, XII:

Lip@AUR-ACP-aptPD-L1+IR. (c) Survival analysis of bilateral B16F10 tumor model-bearing mice (n=6). I: PBS, II: Lip, III: Lip-aptPD-L1, IV: Lip-ACP-aptPD-L1, V: Lip@AUR-aptPD-L1, VI: Lip@AUR-ACP-aptPD-L1, VII: PBS+IR, VIII: Lip+IR, IX: Lip-aptPD-L1+IR, X: Lip-ACP-aptPD-L1+IR, XI: Lip@AUR-aptPD-L1+IR, XII: Lip@AUR-ACP-aptPD-L1+IR. (d-f) Flow cytometry analysis on the infiltration levels of DCs (CD11c+CD80+CD86+), effector T cells (CD3+CD4+CD8+) and memory CD8+T cells (CD8+CD44+CD62L-) within the distal tumors after treatment with I: PBS, II: Lip, III: Lip-aptPD-L1, IV: Lip-ACP-aptPD-L1, V: Lip@AUR-aptPD-L1, VI: Lip@AUR-ACP-aptPD-L1.

Reviewer #2 (Remarks to the Author):

The authors reported multifunctional fusogenic liposomes for enhanced radio-immunotherapy with multivariate-gated aptamer assemblies. They demonstrated the programmable mode of action in vitro and observed combined therapeutic effects in vivo. However, experimental supports are not sufficient to verify their concept, particularly about “cascade-amplifying manner”, and the “requirement of logic-gate”. Due to complicated manufacturing process, potential for clinical translation of this therapeutic tool is also a big concern. On the one hand, overall readability is quite poor. More coherent and structured arrangement of figures and detailed descriptions about figures are required to present the key findings for readers. Based on the above-mentioned substantial issues, I cannot recommend publishing the manuscript in Nature Communications.

A: Thank you for the careful review of our study as well as the constructive criticisms. We are grateful for the insights you kindly provided and would like to first apologize for the insufficient data support for major points in this study, based on which we have carried out substantial amount of additional characterizations to support our concepts and claims. Specifically, we have investigated major aspects of the liposome-augmented radio-immunotherapy in a time-dependent manner to demonstrate the stage-wise activation of the self-assembled aptamer construct-bearing liposomes in response to the input of different triggering signals, thus elucidating the concept regarding the multistage enhancement of the eventual anti-melanoma efficacy in a cascade manner while also supporting its logic-gate-like therapeutic activities. We also would like to apologize for the lack of elaboration on the preparation of the aptamer-based constructs and liposomal formulations from a translational perspective and corrected the wording in relevant places to avoid misunderstanding. Specifically, we would like to note that aptamer synthesis and modification are well-established technologies and the aptamer components in the present study are commercially available from various biochemical suppliers. It is also important to mention that the preparation of the aptamer constructs is a completely autonomous process driving the self-regulated complementary binding of individual aptamer sequences, thus ensuring the quality of the obtained aptamer constructs with molecular precision. Meanwhile, AUR is a clinically approved therapeutic agent that has been used for treating rheumatoid arthritis for more than 30 years, and the repositioning of AUR could substantially reduce the safety concerns of the liposomal formulation. Furthermore, the self-assembled aptamer constructs and

AUR were facilely integrated into a single liposomal platform via simple procedures, which is a clinically tested technology with intrinsic biocompatibility and easy to synthesize with high cost-effectiveness and quality control. Overall, these traits may substantially reduce the complexity for the synthesis and implementation of these liposomes and facilitate their clinical translation.

Regarding the issues associated with the writing and figure preparation you kindly pointed out, we have carefully optimized the wording and figure organization throughout the manuscript to improve its readability, while additional experimental data were added to support the conclusions of this study. For instance, we have carried out comprehensive time-dependent analysis regarding liposome fusion with cytoplasmic membranes of melanoma cells and the subsequent membrane retention of ACP assemblies via fluorescence labeling. Furthermore, the release kinetics of DC-stimulatory eCpG from the membrane-bound ACP assemblies was systematically investigated to validate the therapeutic benefit of the multivariate AND-gate operation of the ACP assemblies both in vitro and in vivo. We ensure that all the concerns you kindly pointed out have been properly addressed in the revised manuscript, of which the detailed responses are outlined below point by point. We hope that these revisions could address your concerns and mobilize this manuscript closer to the high standard of Nature Communications.

Specific comments for authors:

- 1) In introduction references for membrane fusion ability of liposomal nanostructure should be provided. More backgrounds that can support the advantage and necessity

of multivariate logic-gated aptamer should be addressed. Abbreviation aptATP, aptPD-L1 in 4th paragraph of introduction first appears without explanation.

A: Thank you for the advice. Based on your suggestions we first added several references in support of the therapeutic functions of membrane fusogenic liposomes into the introduction section, which were selected from relevant recent reports. Meanwhile, we have expanded the discussions in the last paragraph of the introduction section to highlight the therapeutic contributions of the multivariate logic-gated aptamer constructs for the cooperative radio-immunotherapy against melanoma, focusing on its capacity for the spatiotemporal control of the immunostimulatory effects. Furthermore, we have added the complete definition for the two terms including aptATP and aptPD-L1 in the introduction section. The detailed changes are shown here below for your review:

References regarding the therapeutic application of fusogenic liposomes:

[Adv Mater. 31(35), e1902952 (2019)]; [Nat Commun. 9(1), 1969, (2018)]; [Adv Mater. 35(14), e2206989 (2023)]; [Angew Chem Int Ed Engl. 53(23), 5815-5820 (2014)]; [Angew Chem Int Ed Engl. 61(1), e202111647 (2022)].

Descriptions related to the therapeutic benefits of the multivariate logic-gated aptamer constructs:

“Indeed, there are already reports that aptamer-based logic-gated nanosystems could convey programmable diagnostic or therapeutic activities, which may substantially improve their controllability and precision in vitro or in vivo^{39,55, 56, 57.}”

“We modified the 5' end of commercially available CpG with a 10-nucleotide long

sequence that could complex with the 5' end region of the ATP-binding aptamer (aptATP) through complementary binding (engineered CpG, eCpG). Meanwhile, we also prepared synthetic MMP-2-degradable peptide nucleic acid (PNA) sequence with complementary binding affinity with the 3' end region of aptATP, which could combine with the aptATP-eCpG complex to form physiologically-stable duplex assemblies (ACP). The preparation of ACP assemblies is a completely autonomous process driving the self-regulated complementary binding of individual aptamer sequences, thus ensuring the quality of the obtained aptamer constructs with molecular precision via simple procedures. Notably, the 3' ends of aptATP and PD-L1-binding aptamer (aptPD-L1) were both modified with lipophilic cholesterol moieties, thus allowing their insertion into the lipid bilayers of DMPC-based fusogenic liposomes. Meanwhile, the hydrophobic AUR was loaded into the lipid contents through physical dissolution, eventually leading to the spontaneous formation of bioresponsive fusogenic liposomes (Lip@AUR-ACP-aptPD-L1) through a simple film-hydration method with good cost-effectiveness and quality control.”

Figure 1. Schematic illustration of Lip@AUR-ACP-aptPD-L1 construction and its radio-immunotherapeutic effect. (I) Schematic depiction of the assembly process of ACP and construction of Lip@AUR-ACP-aptPD-L1. As the primary ATP binding sequence in aptATP was simultaneously occupied by the “GGAGTATTGC” segments in the 5' end of eCpG and the “AGGAA-GG-TAAGA” segments located near the MMP-2-cleavable peptidic chain in PNA, the ACP assemblies have high stability in physiological environment with negligible eCpG leakage. **(II)** Schematic representation of the AND-gate release of eCpG from ACP assembly in Lip@AUR-ACP-aptPD-L1 in the context of IR treatment. **(III)** Lip@AUR-ACP-aptPD-L1 mediates sequential radiosensitization of melanoma cells and anti-tumorigenic remodeling of tumor immune microenvironment, potentiating enhanced radio-immunotherapeutic efficacy.

2) The content of fig 2 seems disorganized and does not effectively convey the main

message of the figure. Fig 2 appears to be overly focused on the characterization of engineered aptamers, compared to its title. The figures are inconsistently arranged and lack coherence.

- In fig 2a, description about the components are not well matched with figure and some descriptions are missed. Those mismatches make it hard to figure out the scheme. (ex. Basic color doesn't match, cholesterol modification of ACP missed, aptATP, eCpG color change after responses, and etc.)

- Fig 2b, c could be better suited for inclusion in the extended data section. Instead, it would be more beneficial to add additional data related to liposome characterization rather than focusing solely on aptamer modification results. Results in the extended data are not enough to provide physicochemical properties of liposome. Providing a summary table of the physicochemical properties of each liposomal formulation, including hydrodynamic size, polydispersity index (PDI), zeta potential, encapsulation efficiency (EE), and other relevant properties, would be valuable.

- Why there is same groups in figure 2g, h (all + groups)

- Figures from d-j should be reorganized, since figure 2f, h are data from liposome treated samples, but other figures are data from vehicle-free treated samples.

A: Thank you for the constructive comments and suggestions. Based on the advices we kindly provided, we have substantially revamped Figure 2 to address your concerns:

For the issue associated with Figure 2a, we have comprehensively optimized panel a in such a way that the elements now correctly reflect the biochemical properties of the corresponding components in the proposed liposomal formulation. Specifically, the

image of ACP constructs now correctly shows the cholesterol tail for insertion into the liposome membrane. Meanwhile, individual components are now represented using the same color for consistency sake. We also would like to note that the colors for aptATP and eCpG after AND-gate release were different from the ACP constructs. Specifically, the ACP assemblies were represented using green color as a whole, while aptATP and eCpG were only part of the ACP constructs and thus separately labeled by purple and blue colors. The choices of different colors for individual ACP components and total ACP assemblies may thus help to better illustrate the ACP structure and activation modes while avoiding misunderstanding. The detailed molecular composition of ACP was presented in Figure 1 for reference. Furthermore, the complete composition of the ACP construct is included with necessary descriptive texts.

For the issues associated with Figure 2b, we have moved the schematic illustrations for aptATP and aptPD-L1 to the SI as Supplementary Figure 1 while adding more data regarding the characterization for the physical and chemical properties of the liposomes (Figure 2i-j). We ensure that the physicochemical properties of the intermediates and final liposome products including Lip, Lip@AUR, Lip@AUR-ACP and Lip@AUR-ACP-aptPD-L1 have been comprehensively investigated in the revised manuscript, which covered their hydrodynamic size, PDI, zeta potential, drug loading amount, encapsulation efficiency and stability in biomimetic buffers (Supplementary Figure 2). Notably, the liposomes are highly homogenous and have good aqueous stability as well as efficient AUR loading, and their size and zeta potential are within the optimal range for in vivo drug delivery. These data have also been summarized into a table for the

intuitive demonstration of the physicochemical properties of the liposome products, which is shown in the main text as Figure 2k.

We would also like to apologize for the confusing group labeling in Figure 2g-h that caused the misunderstanding, which have been corrected and shown in the revised manuscript as Figure 2d and 2g, respectively.

For the disordered data arrangement in Figure 2d-j, we have categorized the data according to the samples and reorganized the panels accordingly. Specifically, characterizations based on vehicle-free ACP constructs were summarized and presented in Figure 2b-d, while characterizations based on Lip@AUR-ACP-aptPD-L1 were shown in Figure 2e-j, thus improving the logic progression of the descriptions while avoiding misunderstanding.

The related data and discussions are shown here below for your review:

“Multivariate-gated activation of aptamer assembly

The multivariate-gated activation mode of the ACP assembly is an essential prerequisite for enhancing the radio-immunotherapeutic efficacy of the liposomal nanoformulation, which is crucial for enabling optimal immunostimulation in post-IR melanomas with spatial-temporal precision while minimizing the potential side effects. To obtain the bioresponsive multi-component aptamer assemblies, we first synthesized eCpG, aptATP, PmP and aptPD-L1 via established procedures as the basic components, of which the complementary binding affinity between aptATP/eCpG and aptATP/PmP pairs provided the mechanistic basis for assembly formation (Figure 2a). It is important to note that the primary ATP binding sequence in aptATP was simultaneously occupied

by the “GGAGTATTGC” segments in the 5' end of eCpG and the “AGGAA-GG-TAAGA” segments located near the MMP-2-cleavable peptidic chain in PNA^{39,55}. Notably, to avoid the potential negative impact of cholesterol modification on the structural and biochemical features of aptATP and aptPD-L1 aptamers, multiple base T units were added at the 3' end of the aptamer sequences as a functional handle. NUPACK simulation of secondary structures of these engineered aptamers showed no changes in the structure and ΔG of the aptamers (Supplementary Figure 1), confirming successful aptamer modification without altering their designated biological functions. To ensure effective eCpG detachment from aptATP/eCpG complexes under ATP competition, we proactively constructed aptamer assemblies with different aptATP/eCpG ratios and tested their responsiveness to ATP treatment. Comparative PAGE analysis under graded ATP concentrations showed that aptamer assemblies at the aptATP/eCpG ratio of 2:1 presented enhanced sensitivity to ATP competition to trigger efficient eCpG release, which was used as the standard condition for subsequent experiment (Figure 2b). Quantitative fluorescence analysis showed that aptATP with eCpG has a complexation efficiency of 97.07% (Supplementary Figure 4c), which was attributed to the molecular specificity of the complementary sequences thereof. The ATP-responsiveness of aptATP/eCpG complex was further profiled by PAGE assay, which showed that treating aptATP/eCpG complexes with an ATP concentration of 0.05 μM was sufficient to induce significant eCpG release (Figure 2c), emphasizing the necessity for the implementation of the AND gate eCpG release function to avoid premature eCpG leakage at background concentrations. Next, the aptATP/eCpG

complexes were sequentially integrated with PNA at an aptATP: PNA ratio of 1:1.5, leading to the formation of duplex structures (ACP) with robust stability under physiological conditions. Similarly, quantitative fluorescence analysis showed that the assembly efficiency of ACP with aptATP, eCpG and PNA was 95.52% (Supplementary Figure 4c), indicating that the assembly process is highly modular with molecular precision. According to Figure 2d, the eCpG release from ACP assembly under sole 200 nM ATP treatment was almost negligible (orange frame); similarly, treating ACP with only 10 nM MMP-2 also failed to induce obvious eCpG release (blue frame). It is worth noting that eCpG was potently released from ACP under the combined treatment of 200 nM ATP and 10 nM MMP-2 (Figure 2d red frame). Comparative analysis on eCpG release profiles immediately suggested that PNA complexation inhibited the ATP recognition and binding capability of aptATP and validated the multivariate-gated eCpG release behavior, which is beneficial for minimizing accidental ACP activation in response to tissue-intrinsic ATP stimulation at background levels.

Construction and characterization of the fusogenic liposomes

Liposomes are a well-tested pharmaceutical technology with easy production and high cost-effectiveness, which have already been used to formulate a myriad of therapeutic substances in the clinics for enhanced delivery. Here the liposomal nanosubstrates were synthesized through the self-assembly of DMPC, DSPE-PEG2000, DOTAP and AUR, thus endowing cytoplasm membrane fusion and long-circulating stability while also achieving spontaneous AUR loading. Due to the proactive modification of cholesterol on the 3' position of aptATP and aptPD-L1, the multivariate-gated ACP assembly and

tumor-targeting aptPD-L1 could be facilely inserted into the lipid bilayers for non-invasive modification to form Lip@AUR-ACP-aptPD-L1 (Figure 2a). According to transmission electron microscopic imaging analysis, the bioresponsive Lip@AUR-ACP-aptPD-L1 showed uniform spherical morphology and high monodispersity (Figure 2e). The imaging results were consistently supported by the quantitative DLS analysis, revealing an average diameter of around 130 nm for the final liposome products (Figure 2k and Supplementary Figure 2a) as well as a polydispersity index of around 0.12 (Figure 2k and Supplementary Figure 2b), confirming the morphological homogeneity of the liposomes thereof. Zeta potential analysis showed that pristine Lip had an average surface charge of around 38.16 mV due to the positively charged status of DOTAP contents (Figure 2k and Supplementary Figure 2c), while the zeta potential of Lip@AUR-ACP-aptPD-L1 dropped significantly to -10.71 mV, supporting the successful immobilization of the negatively-charged aptamers. We also found that the Lip@AUR-ACP-aptPD-L1 nanoformulation presented good loading capacity for the therapeutic contents. Specifically, ICP and quantitative fluorescence analysis showed that the AUR could be efficiently loaded into the fusogenic liposomes at a high efficiency of around 88.29%, and the relative AUR ratio in the final Lip@AUR-ACP-aptPD-L1 was around 4.98% (Figure 2k, Supplementary Figure 2d-e and Supplementary Figure 3). Due to the presence of the cholesterol tails, the liposomal integration of ACP assembly and aptPD-L1 was highly efficient with a loading efficiency of 86.5% and 81.0%, respectively, while the average number of ACP assembly and aptPD-L1 on a single liposome was 109 and 51 based on fluorescence

spectroscopy (Supplementary Figure 5). DLS analysis also revealed that Lip@AUR-ACP-aptPD-L1 has favorable stability in aqueous solution with no noticeable size changes after incubation for 2 days (Figure 2f). Alternatively, quantitative fluorescence analysis showed that the degradation rate of ACP in Lip@AUR-ACP-aptPD-L1 was only 19.21% or 14.31% after incubation in 10% FBS or DNase 1 for 30 h (Figure 2g). These data confirmed the relative stability of Lip@AUR-ACP-aptPD-L1 in biomimetic buffers, which is beneficial for improving the therapeutic index of AUR and ACP after systemic administration.

Multivariate AND-gate operation of Lip@AUR-ACP-aptPD-L1

To test if the multivariate AND-gate operation of the ACP assembly was maintained after the integration into Lip@AUR-ACP-aptPD-L1, the liposomes were processed with different stimulus inputs and the corresponding eCpG release was monitored by DNA-PAGE analysis or fluorescence spectroscopic analysis. Consistent with the observations of vehicle-free ACP assemblies in Figure 2d, the combinational treatment of 200 nM ATP and 10 nM MMP-2 induced efficient eCpG release from Lip@AUR-ACP-aptPD-L1 according to the DNA-PAGE analysis (Figure 2h), which was also supported by the results of the quantitative fluorescence spectroscopic analysis that eCpG release rate from Lip@AUR-ACCy5P-aptPD-L1 reached around 92.48% after 2 h incubation with 200 nM ATP and 10 nM MMP-2 (Supplementary Figure 4d). Contrastingly, eCpG release remained at a low level when the ATP and MMP-2 inputs are not simultaneously available (Figure 2i). These observations collectively supported the multivariate AND-gate operation of ACP assemblies was well maintained after

integration into the fusogenic liposomal platform. Alternatively, Lip@AUR-ACCy5-aptPD-L1 showed high sensitivity to ATP input even in the absence of MMP-2, where the eCpG release rate reached around 63.51% under a low ATP concentration of 100 nM (Figure 2j), immediately suggesting the compromised release control due to the lack of the engineered PNA segments. Considering the universal ATP release and MMP-2 upregulation in IR-treated tumors in the clinics, it is anticipated that the multivariate AND-gate operation of the liposome-integrated ACP assemblies could remain stable in response to ATP in physiological environment at background concentrations while enabling superior spatiotemporal control over their eCpG-dependent immunomodulatory activity, supporting its potential utility for post-IR immunostimulation.”

“Characterization of liposome sample series

Lip, Lip@AUR, Lip@AUR-ACP and Lip@AUR-ACP-aptPD-L1 were synthesized according to the above experimental procedure. The hydrodynamic size, polydispersity index and zeta potential of these liposomes were detected by DLS. The concentration of AUR was measured by ICP and the relative encapsulation amount and encapsulation efficiency of AUR were measured via standard curve calibration.”

“DNA degradation assay of Lip@AUR-ACCy5P-aptPD-L1

According to the experimental procedure in previous reports^{54,59}, Lip@AUR-ACCy5P-aptPD-L1 was synthesized and placed in the PBS with 10% FBS or DNase 1. The supernatant was obtained at different time points using an ultrafiltration tube (MWCO: 5000), afterwards the fluorescence intensity of eCpG in the supernatant was detected

by a fluorescence spectrophotometer.”

“Evaluation of ACP assembly construction

The standard curves of Cy5 labeled eCpG and PNA were obtained by fluorescence spectroscopy. According to the previously experimental procedure, AC^{Cy5} and ACP^{Cy5} were assembled with 4 nM aptATP, 2 nM eCpG and 6 nM PNA and then filtered using an ultrafiltration tube (MWCO: 8000). Finally, the fluorescence intensity of Cy5 was detected by fluorescence spectroscopy.”

“eCpG release patterns from Lip@AUR-ACP-aptPD-L1

The standard curves of Cy5 labeled eCpG was obtained by fluorescence spectroscopy. According to the previously reported experimental procedure^{54, 59}, Lip@AUR-ACCy5P-aptPD-L1 was assembled with 4 nM aptATP, 2 nM eCpG and 6 nM PNA, followed by incubation with 200 nM ATP and 10 nM MMP-2. The supernatant was obtained at different time points using an ultrafiltration tube (MWCO: 8000), afterwards the fluorescence intensity of Cy5 in the supernatant was detected by fluorescence spectroscopy.”

“Loading analysis of eCpG and aptPD-L1

The synthesis of fluorescently labeled liposomes was generally the same with those fluorescence-free ones except that the original aptamers were replaced by Cy5-labeled eCpG or FAM-labeled aptPD-L1, leading to the formation of Lip@AUR-ACCy5P-aptPD-L1 or Lip@AUR-ACP-aptPD-L1^{FAM}. 56 μ L Lip@AUR-ACCy5P-aptPD-L1 ($5\text{mg}\cdot\text{mL}^{-1}$) or Lip@AUR-ACP-aptPD-L1^{FAM} ($5\text{mg}\cdot\text{mL}^{-1}$) aqueous solution was added to $1\times$ DNase 1 buffer solution and then treated with $20\text{U}\cdot\text{mL}^{-1}$ DNase 1, incubated at

37°C for 15 min and transferred to an ultrafiltration tube. After centrifugation at 10,000 rpm for 15 min, the supernatant was collected and fluorescence intensity of Cy5 or FAM was detected by fluorescence spectroscopy. eCpG^{Cy5} or aptPD-L1^{FAM} solution with different concentrations were configured to establish the standard curves via a fluorescence spectrophotometer. The aptamer concentrations in Lip@AUR-AC^{Cy5}P-aptPD-L1 or Lip@AUR-ACP-aptPD-L1^{FAM} was quantified according to the standard curve, and then the load efficiency of eCpG^{Cy5} or aptPD-L1^{FAM} on liposomes was calculated accordingly.”

Figure 2. Physicochemical characterization of the fusogenic liposomes. (a) Preparation process and lipid composition of Lip@AUR-ACP-aptPD-L1. (b) DNA-PAGE analysis regarding eCpG release from aptATP/eCpG complex in response to different ATP concentrations. (c) Impact of competitive ATP binding on aptATP/eCpG complex via DNA-PAGE analysis (aptATP: eCpG=2:1). (d) DNA-PAGE analysis regarding eCpG release from the ACP assembly (aptATP: eCpG: PNA=2:1:3) with 200

nM ATP and 5 nM or 10 nM MMP-2. (e) TEM results of Lip@AUR-ACP-aptPD-L1 stained with 4% phosphotungstic acid. (f) The stability of Lip@AUR-ACP-aptPD-L1 in pH7.4 PBS buffer at 12-48 h by DLS analysis. (g) Degradation assessment of Lip@AUR-ACP-aptPD-L1 in DNase 1 or 10% FBS through 50 h incubation. (h) DNA-PAGE analysis of eCpG release from Lip@AUR-ACP-aptPD-L1 with 200 nM ATP and 5 nM or 10 nM MMP-2. (i) Fluorescence analysis of eCpG^{Cy5} release from Lip@AUR-ACP-aptPD-L1 with or without ATP (200 nM) and MM) stimulus input. I: ATP-/MMP-2-, II: ATP-/MMP-2+, III: ATP+/MMP-2-, IV: ATP+/MMP-2+. (j) Fluorescence analysis of eCpG^{Cy5} release from different liposome formulations under different ATP concentrations. I: Lip@AUR-AC^{Cy5}-aptPD-L1, II: Lip@AUR-AC^{Cy5}P-aptPD-L1, III: Lip@AUR-AC^{Cy5}P-aptPD-L1+MMP-2 (5 nM), IV: Lip@AUR-AC^{Cy5}P-aptPD-L1+MMP-2 (10 nM). (k) Summarization of the physicochemical properties of the intermediates and final product of the liposomes.

Supplementary Figure 1. Molecular simulation results for key aptamer components. NUPACK analysis of (a) aptATP and (b) aptPD-L1 after molecular engineering.

Supplementary Figure 2. Physical and chemical properties of the liposome sample series. (a-c) The hydrodynamic size, polydispersity index and zeta potential of liposomes by DLS. (d) The standard curve with AUR. (e) ICP-dependent determination of AUR content in liposomes.

Supplementary Figure 3. Fluorescence analysis of AUR loading into the liposomes.

(a) The fluorescence values of AUR at different concentrations and (b) the corresponding standard curve.

Supplementary Figure 4. Evaluation of aptamer assembly and eCpG release. (a-b)

Standard curve of eCpG^{Cy5} and PNA^{Cy5} according to fluorescence analysis. (c)

Assembly efficiency of AC^{Cy5} and ACP^{Cy5}. (d) Release efficiency of eCpG^{Cy5} from Lip@AUR-AC^{Cy5}P-aptPD-L1 in a time-dependent manner.

Supplementary Figure 5. Integration efficiency of ACP and aptPD-L1 in Lip@AUR-ACP-aptPD-L1. (a-b) Fluorescence spectra of eCpG^{Cy5} and aptPD-L1^{FAM} under different concentrations. (c) Schematic demonstration for the calculation of average number of aptamers in the liposomes. Notable parameters are described in the figure text.

3) The content of fig 3 does not provide proper explanations about each figure for readers and some experimental supports are not sufficient and should be improved to verify their results.

- In fig 3a, details about FACS plot (marker for each fluorescence, or gating strategy)

should be clearly described. Instead of gating plot, quantitative analysis of intensity of fluorescence should be provided as authors mentioned “~ 5-fold higher than splenocytes) Also authors mentioned “we developed a co-culture system comprising B16F10 cells and mouse splenocytes and ~” before starting description about fig 3b. But aren't these mentioned passage in the main text should appear before fig 3a to provide the explanation? Or are fig 3b from co-culture system?

- In fig 3b, why are they showing green signal? And also compared to group III, group II (fusogenic liposome) looks like their DiI signal is rather trapped in lysosome instead membrane. It would be better to show enlarged confocal image. Based on other studies about membrane fusogenic liposome, precise charge interaction between plasma membrane thought to be most crucial for fusion. Doesn't aptamer incorporation affect those charge interactions? Or PD-L1 binding induce ligand mediated endocytosis rather than fusion? Even though authors demonstrate the fusion ability of liposome by confocal image, more supportive results to prove that modification of liposomes does not change fusogenic property should be addressed.

- Based on the data of fig 3b, authors designed the time interval for in vivo administration, but it is not reasonable to set the in vivo administration condition based on those results, with same liposomal formulations, in vivo membrane retention kinetics analysis could be available.

- In fig 3g, description about group is not provided.

A: Thank you for the correction. Based on your advice we have added in-depth analysis for the existing data, while additional experiment and characterizations were carried

out to support our claims. The detailed responses and revisions are shown here below:

Regarding the issues associated with Figure 3a, we have added detailed descriptions regarding the meaning for individual fluorescence color. Meanwhile, we have expanded the captions of the gating strategy for ease of reading. We would also like to apologize for the missing quantitative data regarding fluorescence distribution and added the histograms into Figure 3a for supporting our conclusions. In the co-incubation system of B16F10 cells and splenocytes, we observed that the amount of B16F10-bound aptPD-L1 was 410% higher than that bond to splenocytes, while the amount of DC-bound eCpG was 220% higher than that bond to B16F10 cells, supporting the binding specificity of aptPD-L1 and eCpG with their designated molecular targets. Furthermore, we would like to note that all the cellular experiments in Figure 3 were carried out using the co-incubation system of B16F10 cells and murine splenocytes. Therefore, the wording and figure arrangement were revamped accordingly to avoid misunderstanding. Specifically, the statement regarding the implementation of the B16F10/splenocyte co-incubation system has been moved to the start of the section, while the data of VEGF pathways was transferred to Supplementary Figure 14 and Supplementary Figure 16, which was replaced by the time-dependence analysis results of the liposome activation in vivo (Figure 3d-e and Figure 3g).

For the issues in Figure 3b, we first would like to explain that the green fluorescence in the images was due to the staining of CellTracker Green (CMFDA) probe, which was added to reveal the cell boundary for highlighting the Dil-characteristic fluorescence distribution in the cytoplasmic membrane. We also agree with the reviewer on the

inappropriate group set-up and lack of representativeness for the data in the original version of this manuscript and completely redo the kinetic study on the membrane fusion process of the liposomes, which has three groups including Lip@Dil, Lip@Dil-ACP and Lip@Dil-ACP-aptPD-L1. The liposomal samples were added into the B16F10/splenocyte co-incubation system and incubated for 3, 6, 12 and 18 h, afterwards the Dil fluorescence was monitored by CLSM imaging. As shown in Figure 3b and Supplementary Figure 6-Supplementary Figure 8, both Lip@Dil and Lip@Dil-ACP showed low fusogenic capacity according to the CLSM results. It is notable that the fusogenic capacity of Lip@Dil-ACP was even lower than Lip@Dil, which was attributed to the electrostatic repulsion between the negatively charged ACP and cytoplasmic membrane that hinders the interaction of liposomal and cytoplasmic membranes. Notably, Lip@Dil-ACP-aptPD-L1 showed evidently superior fusogenic capacity than both Lip@Dil and Lip@Dil-ACP in the co-culture system, ascribing to the integration of melanoma-targeting aptPD-L1 components. The aptPD-L1-boosted membrane fusion of liposomes is consistent with recent findings that the presence of cell-binding ligands could facilitate the fusion of the liposomes and cytoplasmic membrane through enhancing the direct interaction in between ^{4, 5, 6, 7}. Overall, the observations above collectively demonstrated that modifying the fusogenic liposomes with ACP and aptPD-L1 could offer melanoma-specific binding affinity to compensate the introduced negative charges and enhance their interaction with the cytoplasmic membranes, leading to accelerated membrane fusion.

For the issues associated with the establishment of the in vivo treatment schedule, we

agree with the reviewer's concerns and created B16F10 melanoma-bearing C57BL/6J mouse models (average tumor size: 100 mm³) to determine if the parameters were suitable for in vivo liposome administration. Specifically, Cy5-labeled liposomes (Cy5-Lip@AUR-ACP-aptPD-L1) were synthesized and administered through tail vein injection. The time of injection was set as the start of the treatment period (0 h) and IR (4 Gy) was applied at 12 h. Melanoma tissues were extracted at 0 h, 6 h, 12 h, 16 h and 18 h, which were pulverized, filtered and lysed with RBC lysis buffer for CLSM imaging. As the melanoma cells have entered a floating state due to IR-mediated cellular damage, we carried out single cell imaging to monitor the membrane fusion of the liposomes. As shown in the updated Figure 3g, substantial amount of Cy5-Lip@AUR-ACP-aptPD-L1 were already bound to the B16F10 cell membrane at 6h, immediately suggesting the melanoma-targeting effect of the liposomes in vivo. Indeed, apparent Cy5 fluorescence was detected in the cytoplasm of B16F10 cells at 16 h, and at 18 h almost all the Cy5 fluorescence was translocated in the cytoplasm while the membrane-bound Cy5 fluorescence decreased to a negligible level. The melanoma cell membrane fusion performance of Lip@AUR-ACP-aptPD-L1 in vivo was consistent with our in vitro observations and supported the reliability for the current in vivo experimental set-up, which is conducive for maximizing eCpG release into the post-IR TME.

As for the issue with Figure 3g in the original manuscript, we have added the group labels into the image for ease of readers. It is notable that the image was moved to Supplementary Materials as Supplementary Figure 16.

The related data and discussions are shown here below for your review:

“Cell-nano-interaction modes of Lip-ACP-aptPD-L1

To investigate the interaction behavior of Lip-ACP-aptPD-L1 in TME under clinically relevant conditions in vitro, we synthesized aptPD-L1 and eCpG with fluorescent FAM tags to construct fluorescently labeled liposomes, which were used for treating the B16F10/splenocyte co-incubation system. Flow cytometric results immediately suggested that the amount of aptPD-L1 bound to B16F10 cell surface was 410% higher than splenocytes, which was in line with the elevated PD-L1 expression status of melanoma cells compared with their normal counterparts or immune cells (Figure 3a). Alternatively, eCpG showed preferential binding to DCs that was 220% higher than other cells (Figure 3a). The fusion of Lip-ACP-aptPD-L1 with cytoplasmic membrane would transfer liposomal ligands onto tumor cell surface, which is crucial for enabling the AND-gate logic operation of ACP in IR-treated melanomas. To monitor the membrane retention kinetics of the fusogenic liposomes, we systematically monitored the fluorescence distribution patterns of different Dil-labeled liposomes after incubation for 3/6/12/18 h in the co-culture system of B16F10 cells and mouse splenocytes. As shown in Figure 3b and Supplementary Figure 6-Supplementary Figure 8, both Lip@Dil and Lip@Dil-ACP showed low fusogenic capacity according to the CLSM results. It is notable that the fusogenic capacity of Lip@Dil-ACP was even lower than Lip@Dil, which was attributed to the electrostatic repulsion between the negatively charged ACP and cytoplasmic membrane that hinders the interaction of liposomal and cytoplasmic membranes. Notably, Lip@Dil-ACP-aptPD-L1 showed evidently superior

fusogenic capacity than both Lip@Dil and Lip@Dil-ACP in the co-culture system, ascribing to the integration of melanoma-targeting aptPD-L1 components. The aptPD-L1-boosted membrane fusion of liposomes is consistent with recent findings that the presence of cell-binding ligands could facilitate the fusion of the liposomes and cytoplasmic membrane through enhancing the direct interaction in between^{58, 59, 60, 61}. Based on the data above, the time interval between liposome administration and IR treatment for the in vitro and in vivo experiments was set to 12 h to ensure that sufficient ACP assemblies were still anchored on tumor cell surface, thus maximizing the eCpG release into the post-IR TME. The tumor-targeted binding and uptake capability of the Lip-ACCy5P-aptPD-L1 liposomes was further validated using tumor spheroid model, evidenced by the strong Cy5 fluorescence in the Lip-ACCy5P-aptPD-L1 group (Figure 3c).

Liposome-mediated radiosensitization and the associated immunogenic effects

Immunofluorescence imaging of PD-L1 with fluorescently-labeled antibodies revealed a general positive correlation between IR dose and PD-L1 expression, which was in accordance with the observations in previous reports^{62, 63, 64} and confirmed the promotional effect of IR on PD-L1 expression (Supplementary Figure S9). The visual trends above were further analyzed quantitatively using flow cytometry, which revealed that the PD-L1 abundance in the AUR+8 Gy group was more than 9-fold higher than the AUR+0 Gy and PBS+0 Gy groups. These data collectively supported the capacity of IR to upregulate PD-L1 expression in a dose-dependent manner, necessitating the incorporation of additional immunostimulatory modalities to compensate the potential

negative impact on post-IR immune responses. Meanwhile, cytotoxicity assay on B16F10 cells or NIH3T3 cells revealed an optimal dosage of Lip@AUR-aptPD-L1 at 40 µg/mL for melanoma treatment, based on a balanced consideration of adverse toxicity and therapeutic potency (Supplementary Figure 10). To test if the liposome-delivered Au-containing AUR could enhance the IR susceptibility of melanoma cells, we incubated B16F10 cells under different conditions of liposomal nanosamples with or without IR treatment. B16F10 cells showed significant resistance to radiotherapy that their survival rate was still around 90% under the IR dose of 4 Gy (Supplementary Figure 11a). In contrast, the combined treatment of Lip@AUR-aptPD-L1 liposomes and 4 Gy IR caused significant melanoma inhibition effect, of which the survival rate dropped to only around 65% at 24 h post treatment, evidently supporting the radiosensitization effect of AUR-containing liposomes (Supplementary Figure 11a). It is also observed that the Lip@AUR-aptPD-L1 + 4 Gy IR treatment induced negligible negative impact on the immune cell populations with a relative viability decrease of less than 10% (Supplementary Figure 11b), while the combined treatment of 8 Gy IR and Lip@AUR-aptPD-L1 liposomes caused a 21% reduction in splenocyte survival and the changes were statistically significant, immediately suggesting the importance to lower the necessary IR doses for maximizing the post-IR immunostimulatory effect with optimal safety. It is also of interest to note that Lip@AUR-aptPD-L1 liposomes induced slight melanoma inhibition effects even without IR treatment, which was ascribed to the intrinsic anti-tumor activity of AUR and also consistent with the observations in recent reports (Supplementary Figure 10), although the changes were

not therapeutically appreciable due to the low loading amount of AUR^{65, 66, 67}. Based on a balanced consideration of AUR-enabled radiosensitization and potential risk of immunosuppression according to the above data, the final IR dose for in vitro and in vivo tests was set to 4 Gy. According to the optimized treatment schedule above, Lip@AUR-aptPD-L1 showed significant improvement in the co-culture system even under the low IR dose of 4Gy according to MTT assay (Supplementary Figure 12). Next, we measured the total ATP and MMP-2 release in vivo at 12/14/16/18/30/36 h incubation with Lip@AUR-ACP-aptPD-L1+4 Gy IR treatment, which eventually reached a plateau after 30 h incubation (Figure 3d-e). It is also observed that the B16F10-associated Dil fluorescence in the Lip@Dil-ACP-aptPD-L1 with 4 Gy IR group was gradually translocated to the cytoplasm after 16 h incubation (Figure 3f and Supplementary Figure 13), which was similar to the result in Figure 3b. To test if the parameters of the in vitro experiment could be used to design the experimental set-up for in vivo analysis, Cy5-labeled liposomes (Cy5-Lip@AUR-ACP-aptPD-L1) were synthesized and administered into B16F10 tumor-bearing mice through tail vein injection. The time of injection was set as the start of the treatment period (0 h) and IR (4 Gy) was applied at 12 h. Melanoma tissues were extracted at 0 h, 6 h, 12 h, 16 h and 18 h, which were pulverized, filtered and lysed with RBC lysis buffer for single-cell CLSM imaging. Images in Figure 3g showed that substantial amount of Cy5-Lip@AUR-ACP-aptPD-L1 were already bound to the B16F10 cell membrane at 6h, immediately suggesting the melanoma-targeting effect of the liposomes in vivo. Indeed, apparent Cy5 fluorescence was detected in the cytoplasm of B16F10 cells at 16 h, and

at 18 h almost all the Cy5 fluorescence was enriched in the cytoplasm while the membrane-bound Cy5 fluorescence decreased to a negligible level. The melanoma cell membrane fusion performance of Lip@AUR-ACP-aptPD-L1 in vivo was consistent with our in vitro observations and the time interval between liposome administration and IR exposure was determined at 12 h for maximizing eCpG release into the post-IR TME.

Based on the kinetic insights described above, the treatment schedule of Lip@AUR-aptPD-L1 in vitro was established and shown in Figure 3h to ensure balanced AUR-mediated IR sensitization/VEGF inhibition and logic operation of ACP. The crosstalk between tumor cells and immunosuppressive cells is a major driver of the immunosuppressive TME. There is already clinical evidence that VEGF secreted by melanoma cells could recruit MDSCs and Tregs to TME for suppressing the effector function of CTLs, thus contributing to their immune escape. Interestingly, recent reports reveal that AUR could demonstrate potent VEGF suppressing capability through inhibiting ERK1/2-HIF-1 α signaling activity in tumor cells^{68, 69, 70}. Indeed, we have carried out transcriptome sequencing on AUR-treated B16F10 cells to screen the treatment-induced impact on various immune-related signaling pathways, and the KEGG enrichment analysis results immediately suggested that AUR treatment pronouncedly inhibited the VEGF signaling pathways (Supplementary Figure 14-Supplementary Figure 15). The VEGF-inhibiting function of AUR-incorporated liposomes was investigated in greater detail via western blot assay. As shown in Supplementary Figure 16, sole IR treatment induced significant activation of the

ERK1/2-HIF-1 α -VEGF axis, which was attributed to the oxygen-consumption effect of IR and consistent with the clinical data in previous reports^{71, 72, 73, 74}. Similar trends in the activation status of ERK1/2-HIF-1 α -VEGF signaling pathway were also observed in those non-AUR-containing groups including Lip+IR, Lip-aptPD-L1+IR and Lip-ACP-aptPD-L1+IR, suggesting their inability to suppressive VEGF expression in melanoma cells. In contrast, Lip@AUR-aptPD-L1+IR and Lip@AUR-ACP-aptPD-L1+IR both induced obvious inhibition on ERK1/2, HIF-1 α and VEGF regardless of the IR treatment condition. The WB data on HIF-1 α expression after different treatments were further supported by immunofluorescence imaging, which showed that the Lip@AUR-aptPD-L1 and Lip@AUR-ACP-aptPD-L1 treatments induced evident reduction in the HIF-1 α -intrinsic red fluorescence compared with the other AUR-free modalities, again confirming that AUR component in the liposomes could inhibit HIF-1 α expression in melanoma cells (Supplementary Figure 17). The data above collectively confirmed that the AUR component in the Lip@AUR-ACP-aptPD-L1 liposomes could effectively inhibit VEGF expression in IR-treated melanoma cells through inhibiting ERK1/2-HIF-1 α axis, offering potential opportunities to impede the recruitment of immunosuppressive cells into TME for restoring antitumor immunity. The potential therapeutic benefit of liposome-induced VEGF suppression was evaluated using co-culture system of B16F10 cells and splenocytes. Flow cytometry analysis showed that fewer Tregs and MDSCs were recruited after Lip@AUR-aptPD-L1+IR treatment, which were 18.47% and 9.44% lower than the control group (Supplementary Figure 18), respectively, accompanied with increasing DC (14.17%,

Supplementary Figure 19b) and CD8⁺ T cell (12.23%, Supplementary Figure 19a) infiltration. The results showed that AUR-mediated VEGF inhibition could potentially establish an anti-tumorigenic microenvironment by reducing Treg and MDSC infiltration. We further investigated if the Lip@AUR-aptPD-L1-mediated radiosensitization of melanoma cells could enhance their immunogenic feature and contribute to immunostimulation. Here we first monitored the cellular status of key DAMPs including ATP (Supplementary Figure 20a), HMGB1 (Supplementary Figure 20b) and CRT (Supplementary Figure 20c) using the corresponding assay kits. Notably, untreated B16F10 cells showed negligible CRT expression as well as low levels of ATP and HMGB1 release, which is in accordance with their low immunogenic potential under common conditions. Low dose (4 Gy) IR treatment induced significant enhancement in CRT expression (140%) and ATP/HMGB1 release (170%/130%) (Supplementary Figure 20), which was attributed to the IR-induced ICD of melanoma cells. However, the relative increase for the abundance of typical DAMPs in IR-treated B16F10 cells were modest at most due to ineffective radiotherapeutic effect. Remarkably, melanoma cells in the Lip@AUR-aptPD-L1+IR group showed the greatest increase in CRT expression (370%) and ATP/HMGB1 secretion (570%/310%) compared with the control group (Supplementary Figure 20), which is in line with the pronounced radiosensitization effect of the Lip@AUR-aptPD-L1 liposomes. These observations evidently supported our hypothesis that the radiosensitization effect of the Lip@AUR-aptPD-L1 liposomes could induce pronounced ICD of melanoma cells and thus offer multifaceted therapeutic benefit. On one hand, the released DAMPs and

tumor-associated neoantigens could stimulate the adaptive immune system to initiate antitumor immune responses. On the other hand, the enhanced ATP secretion could cooperate with IR up-regulated MMP-2 to trigger the AND-gate activation of the ACP assembly and release eCpG into TME for programmable sequential DC stimulation.”

“Observation of membrane fusion behavior of the liposomes

Herein, orange-red probe Dil was loaded into the liposome instead of AUR for fluorescence tracking, of which the samples were denoted as Lip@Dil, Lip@Dil-ACP and Lip@Dil-ACP-aptPD-L1. B16F10 cells were inoculated into confocal dishes at a density of 1×10^5 cells/well. When the cell confluence reached 80%, the co-culture system was established with the B16F10: splenocyte ratio of 1:10. 40 $\mu\text{g/mL}$ Lip@Dil, Lip@Dil-ACP or Lip@Dil-ACP-aptPD-L1 was added respectively into the co-culture system. At 3, 6, 12 and 18 h of incubation, cell samples were washed with PBS to remove the floating spleen cells, while the remaining B16F10 cells were first stained with CellTracker Green CMFDA for 15 min and then by DAPI for 10 min. Finally, the membrane fusion status of the liposomes with B16F10 cells was detected by laser confocal microscopy.

In addition, B16F10 cells were inoculated into confocal dishes at a density of 1×10^5 cells/well. When the cell confluence reached 80%, the co-culture system was established with the B16F10: splenocyte ratio of 1:10. 40 $\mu\text{g/mL}$ Lip@Dil, Lip@Dil-ACP or Lip@Dil-ACP-aptPD-L1 was added separately into the co-culture system. After 12 h incubation, the above cells were treated with 4 Gy IR. At 16, 18 and 30 h of incubation, while the remaining B16F10 cells were first stained with CellTracker Green

CMFDA for 15 min and then by DAPI for 10 min. Finally, the membrane fusion status of the liposomes with B16F10 cells was detected by laser confocal microscopy.”

“Evaluation of IR-mediated impact on ATP and MMP-2 expression *in vivo*

1×10^6 B16F10 cells were injected subcutaneously into 6-week-old mice to establish B16F10-bearing C57BL/6J mouse model. The mice were cultured continuously until the tumor size reached 100 mm^3 , then treated with Lip@AUR-ACP-aptPD-L1 ($2 \text{ mg} \cdot \text{kg}^{-1}$) by intravenous injection. The administration time was defined as 0 h. The mice received 4 Gy IR treatment at 12 h post intravenous injection. The tumors were dissected at 12, 14, 16, 18, 30 and 36 h post injection, then ground to the powder using liquid nitrogen, and then lysed with RIPA lysate (containing 1% PMSF) for 30 min at 4°C and centrifuged at $4000 \times g$ for 10 min. The collected supernatant was used for detect the concentrations of ATP and MMP-2 in each tumor by the relevant kits.”

“Observation of membrane fusion behavior of the liposomes *in vivo*

1×10^6 B16F10 cells were injected subcutaneously into 6-week-old mice to establish B16F10-bearing C57BL/6J mouse model. The mice were cultured continuously until the tumor size reached 100 mm^3 , then treated with Cy5-Lip@AUR-ACP-aptPD-L1 ($2 \text{ mg} \cdot \text{kg}^{-1}$) by intravenous injection. The administration time was defined as 0 h. The mice received 4 Gy IR treatment at 12 h post intravenous injection. The tumors were dissected at 0, 6, 12, 16 and 18 h post injection, which were pulverized and filtered to extract the cells. The extracted cells were mixed with 5 mL red blood cell lysis buffer and stood for 10 min, and then centrifuged at $666.7 \times g$ for 5 min. Afterwards they were stained with CellTracker Green CMFDA for 15 min and then with DAPI for 10 min.

Finally, the membrane fusion status of liposomes with B16F10 cells was detected by laser confocal microscopy.”

Figure 3. Melanoma targeting and membrane fusion performance of Lip@AUR-ACP-aptPD-L1 in vitro based on B16F10/splenocyte co-incubation system. (a)

Targeting ability of eCpG and aptPD-L1 with the designated cell targets in the B16F10/splenocyte co-incubation system by flow cytometry. (b) Fusion status of different liposomes to B16F10 cell membranes in the co-culture system at 3, 6, 12 or 18 h of incubation by CLSM. I: Lip@Dil, II: Lip@Dil-ACP, III: Lip@Dil-ACP-aptPD-L1. Red: Dil. Green: CellTracker Green CMFDA. Blue: DAPI. (c) Tumor sphere assay on the targeting ability of different samples at 12 h incubation. I: AC^{Cy5}P, II: Lip-AC^{Cy5}P, III: Lip-AC^{Cy5}P-aptPD-L1. (d-e) Time-dependent changes in ATP and MMP-2 abundance in vivo with Lip@AUR-ACP-aptPD-L1 + 4 Gy IR treatment. (f) Fusion status of different liposomes to B16F10 cell membranes in the co-culture system at 16, 18 or 30 h of incubation by CLSM. 4 Gy IR treatment was applied at 12 h. I: Lip@Dil+IR, II: Lip@Dil-ACP+IR, III: Lip@Dil-ACP-aptPD-L1+IR. Red: Dil. Green: CellTracker Green CMFDA. Blue: DAPI. (g) Time-dependent melanoma-targeted membrane fusion performance of Cy5-Lip@Dil-ACP-aptPD-L1 in vivo. 4 Gy IR treatment was applied after 12 h post intravenous injection. (h) Schedule of the combinational Lip@Dil-ACP-aptPD-L1 + 4 Gy IR treatment set-up in vitro. Statistical analysis in panels d-e was carried out via one-way ANOVA method. * indicates significance at $p < 0.05$, *** indicates significance at $p < 0.001$, **** indicates significance at $p < 0.0001$.

Supplementary Figure 6. Fluorescence analysis on the membrane fusion of the fusogenic liposomes. Evaluation on the membrane fusion performance of Lip@Dil with B16F10 cells and spleen cells in the co-culture system.

Supplementary Figure 7. Fluorescence analysis on the membrane fusion of the ACP-integrated liposomes. Membrane fusion of Lip@Dil-ACP in the co-culture system with B16F10 cells and spleen cells.

Supplementary Figure 8. Fluorescence analysis on the membrane fusion of the ACP-integrated melanoma-targeted liposomes. Membrane fusion of Lip@Dil-ACP-aptPD-L1 in the co-culture system with B16F10 cells and spleen cells.

Supplementary Figure 9. Dose-dependent impact of IR on B16F10-intrinsic PD-L1 expression. (a) Confocal microscopic analysis of PD-L1 expression in AUR-treated B16F10 cells with graded IR doses. (b) Flow cytometry analysis on the expression status of PD-L1 in B16F10 cells after different treatment. I: PBS+0 Gy, II: PBS+2 Gy, III: PBS+4 Gy, IV: PBS+8 Gy, V: AUR+0 Gy, VI: AUR+2 Gy, VII: AUR+4 Gy, VIII: AUR+8 Gy.

Supplementary Figure 10. Cytotoxicity profiles of the liposome systems. MTT assay regarding the cytotoxicity of Lip@AUR-aptPD-L1 on NIH3T3 cells or B16F10 cells under different concentrations.

Supplementary Figure 11. Evaluation on the radiosensitizing effect of the liposomes. Impact of the combined treatment of Lip@AUR-aptPD-L1 and IR of different doses on B16F10 cells (a) and splenocytes (b).

Supplementary Figure 12. Evaluation on the radio-immunotherapeutic effect of the liposomes. Cytotoxic impact of Lip@AUR-aptPD-L1 + 4 Gy IR on B16F10 cells and splenocytes in the co-culture system by MTT assay.

Supplementary Figure 13. Time-dependent analysis of the membrane fusion of liposomes. Comparative analysis of the membrane fusion capacity of different liposomes with 4 Gy IR to B16F10 cells under co-culture condition.

Supplementary Figure 14. Bioinformatic analysis of the treatment-induced therapeutic impact. Transcriptome sequencing analysis regarding the impact of combined Lip@AUR-aptPD-L1+ 4 Gy IR treatment on the VEGF pathway in B16F10 cells.

Supplementary Figure 15. Transcriptome sequencing of Lip@AUR-aptPD-L1+4

Gy IR treated B16F10 cells. (a) KEGG pathway classification results of B16F10 cells.

(b) KEGG enrichment results of B16F10 cells.

Supplementary Figure 16. Western Blot analysis on the biochemical changes in IR-treated B16F10 cells in vitro. (a) Western Blot analysis on the expression levels of key proteins related to IR damage and ERK1/2-HIF-1 α -VEGF pathway. (b-c) Statistical analysis of panel a. I: PBS, II: Lip, III: Lip-aptPD-L1, IV: Lip-ACP-aptPD-L1, V: Lip@AUR-aptPD-L1, VI: Lip@AUR-ACP-aptPD-L1.

Supplementary Figure 17. Immunofluorescence analysis on AUR-mediated suppression of HIF-1 α expression. Expression of HIF-1 α in the B16F10 nucleus after different treatment. I: PBS, II: Lip, III: Lip-aptPD-L1, IV: Lip-ACP-aptPD-L1, V: Lip@AUR-aptPD-L1, VI: Lip@AUR-ACP-aptPD-L1.

Supplementary Figure 18. Evaluation on the treatment-induced inhibition of major immunosuppressor cell populations. Frequency of Tregs (CD25+CTLA-4+) and MDSCs+ (CD11b+GR1+) in the co-culture system after treatment with I: PBS, II: Lip, III: Lip@AUR, IV: Lip@AUR-aptPD-L1.

Supplementary Figure 19. Evaluation on the treatment-induced expansion of major cell populations related to the adaptive immunity. Frequency of mature DCs (CD11c+CD80+CD86+) and IFN- γ + CD8+T cells (CD8a+IFN- γ +) in the co-culture system after treatment with I: PBS, II: Lip, III: Lip@AUR, IV: Lip@AUR-aptPD-L1.

Supplementary Figure 20. Quantitative analysis of treatment-induced release of key DAMPs. The group set-ups include (a) ATP, (b) HMGB1 and (c) CRT. I: PBS, II: Lip, III: Lip-aptPD-L1, IV: Lip@AUR-aptPD-L1.

4) Fig 4, 5 provides overall immunostimulatory effects of developed liposome and mechanisms for those effects. However, there are some insufficient demonstrations about results since results are mainly based on co-culture system, and also more clear time-dependent studies should be addressed to support these liposomes will act similarly in vivo.

- In fig 4g, authors evaluate the effect of PmP binding on the stability of complexed eCpG. To confirm the role of PmP, authors should provide the increased MMP-2 expression after IR in prior.

- Also, those cascades suggested by authors will occur after (1) PD-L1 mediated interaction, and (2) IR irradiation. Then ATPs will be released as AUR&IR-mediated ICD occurs. In this situation, ATPs can actively bound to aptATP regardless of MMP-2 expression. I wonder if there are any reasons for the requirement of AND-gate.

- Also, as mentioned above, those cascades suggested by authors will be expected to occur step by step, and temporal control of those responses are most important issue for in vivo experiments. In vitro data from fig 4, 5 seems to be presenting only the results-

oriented data from cascades. More supportive data about developed liposome works in temporal manner should be addressed.

- Authors summarized the figure 4 in text as “~and stimulate the adaptive antitumor immune response in IR-treated melanomas”, but based on results of figure 4, we cannot get any information about adaptive immune response.

- Authors summarized the figure 5 in text as “~through cascade-amplifiable manner” and also the title of manuscript emphasizes the cascade amplification. In what points, authors demonstrate the amplifiable manner? Based on the mechanism study, it seems to happen in programmable sequential manner, not the amplifiable manner. Clear demonstration or strong experimental supports should be provided to argue the cascade-amplification.

A: Thank you for the insights. Based on the advice you kindly suggested, we have made substantial addition to the data presented in the two figures, which included explorations on the molecular basis of liposome activation using appropriate cell models and time-dependence of the liposome-augmented radio-immunotherapeutic activity. For ease of discussion, the time of drug administration both in vitro and in vivo was defined as 0 h throughout the manuscript.

For the issue associated with the impact of IR on MMP-2 expression, we have monitored the expression status of MMP-2 in vivo after the combined treatment with Lip@AUR-ACP-aptPD-L1 and 4 Gy IR. As shown in Figure 3d-e, at 2 h post IR treatment (14 h of incubation in total), the ATP and MMP-2 abundance in B16F10 tumors already showed evident increase. Notably, the ATP and MMP-2 levels within

the time period of 12 h-16 h were sufficient for triggering efficient AND-gate release of eCpG from ACP, which was in line with the data shown in Figure 4g-i.

As for the issues associated the necessity of the multivariate AND-gate eCpG release behaviors, we would like to first apologize for the lack of elaboration on the ACP structure and the associated therapeutic benefit. We would like to state that the ACP assembly was formed through the simple mixture of aptATP, eCpG and PNA at the ratio of 2:1:3. It is important to note that primary ATP binding sequence in aptATP was simultaneously occupied by the “GGAGTATTGC” segments in the 5' end of eCpG and the “AGGAA-GG-TAAGA” segments located near the MMP-2-cleavable peptidic chain in PNA^{8, 9}, of which the detailed binding pattern is illustrated in Figure 1. Therefore, the ACP assembly would remain in a complexed state when only ATP was present. Contrastingly, when both the ATP and MMP-2 inputs were present at adequate levels, the MMP-2 could decompose the PNA and liberate the ATP binding site in aptATP for ATP complexation, leading to the AND-gate release of eCpG. Notably, this multivariate AND-gate operation of the ACP assemblies may convey several therapeutic merits, all of which are highly relevant for boosting the radio-immunotherapeutic efficacy against melanoma. It is well-established that ATP are universally present in various healthy and malignant tissues at background concentrations^{8, 10, 11, 12}. Considering the high sensitivity of aptATP/eCpG complexes to competitive ATP binding, they are prone to rapid eCpG leakage after intravenous administration, which would substantially enhance the risk of adverse off-target immune reactions while impairing antitumor efficacy. Meanwhile, it is notable that the

immunosuppressive TME is characterized by the lack of tumor-associated antigens, which leads to critical defects in the antigen processing and presentation cascades in tumors. Here the multivariate AND-gate operation of the ACP assemblies ensures the spatiotemporally controlled release of eCpG into the IR-remodeled TME, which would stimulate the tumor-infiltrating DCs to recognize the IR-induced tumor-derived immunogenic materials and present them to T cells for activating robust adaptive antitumor immunity. Furthermore, the ACP nanoassemblies were anchored on melanoma surface through the membrane fusion of the liposomes, thus avoiding their endocytosis into tumor cells while facilitating eCpG release into the TME. To test if the multivariate AND-gate operation of the ACP assemblies could achieve the desired eCpG release profiles, we synthesized a series of fluorescently labeled liposome samples including the PNA-free Lip-AC^{Cy5}-aptPD-L1 and PNA complexed Lip-AC^{Cy5}P-aptPD-L1, which were separately treated with 0 nM ATP+5 nM MMP-2, 100 nM ATP+5 nM MMP-2, 0 nM ATP+10 nM MMP-2 and 200 nM ATP+10 nM MMP-2, resembling the triggering input in tumors before and after IR treatment by referring to Figure 3d-e. According to the CLSM imaging results in Figure 4g-h, the incubation condition of 100 nM ATP+5 nM MMP-2 induced partial eCpG release from Lip-AC^{Cy5}-aptPD-L1 but failed to trigger obvious eCpG release from Lip-AC^{Cy5}P-aptPD-L1. In comparison, both Lip-AC^{Cy5}-aptPD-L1 and Lip-AC^{Cy5}P-aptPD-L1 showed almost complete eCpG release under the incubation conditions of 200 nM ATP+10 nM MMP-2. By referring to the data in Figure 3 d-e, these observations immediately suggested that a large proportion of complexed eCpG in Lip-AC^{Cy5}-aptPD-L1 would be released

before the implementation of IR in a premature manner, while Lip-AC^{Cy5}P-aptPD-L1 could release the eCpG payload in IR-treated tumors in a spatiotemporally coordinated manner. The eCpG release profiles was further investigated in vivo. Here the B16F10 tumor-bearing mouse models were intravenously injected with Lip@AUR-AC^{Cy5}-aptPD-L1 and Lip@AUR-AC^{Cy5}P-aptPD-L1 and exposed to 4 Gy IR at 12 h post injection, after which the release of eCpG^{Cy5} was observed in single tumor cell by laser confocal microscopy. The results showed that most of eCpG^{Cy5} of Lip@AUR-AC^{Cy5}-aptPD-L1 was released regardless of the IR treatment conditions, while eCpG^{Cy5} of Lip@AUR-AC^{Cy5}P-aptPD-L1 was mostly retained on B16F10 tumors in the absence IR treatment but almost completely released after IR exposure (Figure 4i). The comparative analysis results on the eCpG release behavior were consistent with the immunostimulatory effects of different samples in vitro and in vivo, evidently confirming the necessity of the multivariate AND-gate operation for improving the radio-immunotherapeutic efficacy against melanomas.

For the issues regarding the time-dependent execution of the therapeutic cascade, we first synthesized fluorescently labeled Lip@AUR-AC^{Cy5}P-aptPD-L1 and used it to treat B16F10 tumor-bearing C57BL/6J mice via intravenous injection. The time of liposome injection was defined as 0 h, and 4 Gy IR was applied at 12 h post-injection. The B16F10 tumors were extracted at 0, 6, 12, 18, 24, 30, 36 h post-injection, which were pulverized, filtered and lysed with RBC lysis buffer to analyze the accumulation of fluorescently-labeled eCpG in B16F10 cells via flow cytometry. As shown in Figure 4j, the B16F10-specific eCpG accumulation reached the maximum at 12 h post-injection

but decreased to a negligible level at 18 h post-injection, suggesting the gradual release of eCpG from B16F10 cell surface during the 12-18 h period. In addition, to further investigate if the released eCpG could promote DC maturation in the post-IR TME, we also isolated DC populations from the extracted tumors described above for flow cytometric analysis. According to the flow cytometric data in Figure 4l-m, the relative frequency of mature DCs showed no obvious changes in the 0-12 h period post-injection but started to increase rapidly after the time point of 16 h, which eventually reached a plateau after 30 h. These observations confirmed that the released eCpG efficiently promoted DC maturation in the post-IR TME in a time-sequenced manner and contributed to the eventual antimelanoma efficacy.

For the issues associated with the statement “and stimulate the adaptive antitumor immune response in IR-treated melanomas”, we would like to apologize for the inappropriate statement here as the evaluation of treatment-induced T cell-mediated immune responses was covered in the next figure but not in Figure 4. Therefore, we have changed the wording to provide a more objective description regarding the DC stimulatory effect of the liposomes, which was changed to “which is conducive for promoting the adaptive antitumor immune response in IR-treated melanomas.”

For the issues associated with the statement “cascade-amplifiable manner”, we would like to first explain that our intention was to highlight the programmable therapeutic activities of the liposome-augmented radio-immunotherapy against melanomas, which would be activated in order after systemic administration and cooperate in a spatiotemporally controlled manner, thus enhancing the eventual antimelanoma

potency. From an overall perspective, the liposome-augmented radio-immunotherapy against melanomas could be divided into 3 major phases. In the initial phase, the Lip@AUR-ACP-aptPD-L1 liposomes bind to melanoma cells through aptPD-L1-mediated targeting effect and fuse with the cytoplasmic membrane, which will transfer the ACP assemblies onto melanoma cell surface while simultaneously delivering AUR. In the second phase, the AUR contents substantially sensitize melanoma cells to IR-mediated cellular damage, thus enhancing the direct melanoma inhibition effect as well as facilitating the release of tumor-derived immunogenic materials including ATP and upregulating MMP-2. In the final phase, the IR-triggered ATP release and MMP-2 activation trigger the AND-gate activation of membrane-bound ACP assemblies to release eCpG, thus promoting DC maturation in the post-IR TME. This may synergize with the AUR-dependent inhibition of the melanoma-intrinsic ERK1/2/HIF-1 α /VEGF pathway to alleviate immunosuppression in the TME, eventually leading to boosted immunotherapeutic efficacy for robust and persistent melanoma elimination at a systemic level. We totally understood the reviewer's concerns and have thus changed the wording throughout the manuscript to avoid misunderstanding. For instance, the title has been changed to "Programmable melanoma-targeted radio-immunotherapy via fusogenic liposomes functionalized with multivariate-gated aptamer assemblies".

The related data and discussions are shown here below for your review:

"AND-gate eCpG release and the immunostimulatory effects of liposomes

Extending from the IR-triggered liposome-augmented ICD of melanoma cells above, we further comprehensively investigated the immunostimulatory impact of liposome-

sensitized melanoma radiotherapy *in vitro*. To start with, we evaluated if the molecular engineering of 5' end of CpG ODN would alter its immunological activities via NUPACK analysis. As shown by the simulation results, the addition of the 10-base aptATP binding sequence caused no alterations in the structure of the stem-loop domain (Figure 4a, b). Subsequently, we employed 3D model-based molecular dock analysis to further profile the complexation of pristine CpG ODN and eCpG with TLR9 proteins. The binding sequence of CpG ODN to TLR9 is base 6-11 (GACGTT) that directly complexes to 337Arg and 338Lys on TLR9 while also presenting indirect interaction with 347Lys, 348Arg and 353His (Figure 4c, d), which was consistent with the structural analysis in previous reports^{75, 76, 77}. Interestingly, eCpG bond to TLR9 through the same GACGTT sequence with identical amino acid interaction, immediately suggesting that the addition of aptATP-binding sequence at the 5' end of CpG induced negligible impact on its TLR9 binding behavior. We further prepared Cy5 labeled eCpG and tested their binding with TLR9-positive DCs (Figure 4e). Notably, eCpG showed comparable TLR9-binding affinity to pristine CpG ODN and was capable of substantially promoting DC maturation (51.10%) (Figure 4f), while mutating the CG bases in the GACGTT sequence induced significant reduction in the DC-binding capacity of the aptamers and failed to induce significant changes in DC maturation ratio after co-incubation. Meanwhile, we detected that pretreating eCpG with the complementary sequence (CTGCAA) of the TLR9-binding domain also impaired their complexation with TLR9-positive DCs and abolished their pro-DC maturation function (20.80%) (Figure 4f). These results collectively supported that the molecularly

engineered eCpG successfully expanded its nanointegrative functionality without impairing its DC-stimulatory activity.

Next, we investigated if the Lip-ACP-aptPD-L1 liposomes could stimulate the DC populations through mediating AND-gate eCpG release in B16F10 cells. To monitor the cellular distribution of eCpG, the eCpG molecules were labeled by Cy5 for fluorescent tracking, of which the samples were denoted as Lip-ACCy5-aptPD-L1 (PNA-free) and Lip-ACCy5P-aptPD-L1 (PNA complexed). By referring to the ATP and MMP-2 abundance in B16F10 cells treated with 4 Gy IR (Figure 3d-e), Lip-ACCy5-aptPD-L1 and Lip-ACCy5P-aptPD-L1 were used to incubate B16F10 cells with 0 nM ATP+5 nM MMP-2, 100 nM ATP+5 nM MMP-2, 0 nM ATP+10 nM MMP-2 or 200 nM ATP+10 nM MMP-2 for 2 h (Figure 4g-h). Notably, Lip-ACCy5-aptPD-L1 only showed partial eCpG release under the incubation condition of 100 nM ATP+5 nM MMP-2, while Lip-ACCy5P-aptPD-L1 did not demonstrate obvious eCpG release under the same incubation condition. In comparison, both Lip-ACCy5-aptPD-L1 and Lip-ACCy5P-aptPD-L1 showed almost complete eCpG release under the incubation conditions of 200 nM ATP+10 nM MMP-2. The eCpG release was further investigated in vivo by treating B16F10 tumor-bearing mouse models with Lip@AUR-ACCy5-aptPD-L1 or Lip@AUR-ACCy5P-aptPD-L1 via intravenous injection and 4 Gy IR at 12 h post injection, and the release of eCpGCy5 from the B16F10 surfaces was eventually monitored by CLSM at 16 h post intravenous injection. Specifically, most of eCpGCy5 of Lip@AUR-ACCy5-aptPD-L1 was released regardless of the IR treatment conditions, while eCpGCy5 of Lip@AUR-ACCy5P-aptPD-L1 was mostly

retained on B16F10 tumors in the absence IR treatment but almost completely released after IR exposure (Figure 4i). These observations immediately suggested the capacity of Lip-ACCy5P-aptPD-L1/Lip@AUR-ACCy5P-aptPD-L1 to release the eCpG payload under conditions resembling IR-treated tumors in a spatiotemporally coordinated manner and confirmed the necessity of the multivariate AND-gate operation of the ACP assembly for maximizing the immunostimulatory benefit while reducing potential adverse immune reactions. The time-dependent execution of the eCpG release from membrane-bound ACP assemblies was further investigated in vivo using B16f10 tumor-bearing mice and liposomes constructed with Cy5-labeled eCpG sequences. As shown in Figure 4j, the B16F10-specific eCpG accumulation reached the maximum at 12 h post-injection but decreased to a negligible level at 18 h post-injection, suggesting the gradual release of eCpG from B16F10 cell surface during the 12-18 h period. As a result of the efficient AND-gate eCpG release, DCs in the Lip@AUR-ACP-aptPD-L1+IR group showed the highest maturation ratio (CD11c+CD80+CD86+) after 30 h incubation in vitro (Figure 4k), indicating that 4 Gy IR successfully triggered the AND-gate eCpG release to promote DC maturation. To further investigate if the released eCpG could promote DC maturation in the post-IR TME, we also isolated DC populations from the extracted tumors for flow cytometric analysis. According to the data in Figure 4l-m, the relative frequency of mature DCs showed no obvious changes in the 0-12 h period post-injection but started to increase rapidly after the time point of 16 h, which eventually reached a plateau after 30 h. These observations confirmed that eCpG released through the AND-gate operation of the membrane-bound ACP

assemblies efficiently promoted DC maturation in the post-IR TME in a time-sequenced manner, which is conducive for promoting the adaptive antitumor immune response in IR-treated melanomas.

We further studied whether the liposome-augmented IR-induced ICD of melanoma cells and the cooperative AND-gate eCpG release could evoke adaptive immunity to achieve effective radio-immunotherapy against melanomas. It is well-established that tumor cells undergoing ICD would release tumor-associated immunogenic materials for the processing and recognition by tumor-infiltrating antigen-presenting cells for mediating the downstream immune reactions. Firstly, to verify whether Lip@AUR-ACP-aptPD-L1 could cause ICD in co-culture system, we detected the expressions of HMGB1 and CRT in B16F10 cells. The confocal microscope results showed that Lip@AUR-ACP-aptPD-L1+IR group significantly decreased the expression of HMGB1 in the nucleus as well as significantly increased the expression of CRT on the cell membrane (Supplementary Figure 22), indicating that Lip@AUR-ACP-aptPD-L1 combined with 4 Gy IR could successfully induce significant tumor immunogenic death. Indeed, flow cytometric analysis on the extracted immune cell populations from the co-incubation system showed that the combined Lip@AUR-ACP-aptPD-L1+IR treatment substantially improved the maturation and antigen-presentation capacity of DC population, where the frequencies of CD11c+CD80+CD86+ (Figure 5a) and CD11c+MHC-II+ (Supplementary Figure 23a) DCs have increased by 31.76% and 34.44% compared with the control group and obviously higher than all other groups. In line with the enhanced activation status of DCs, the Lip@AUR-ACP-aptPD-L1+IR

group showed a substantial expansion of the CD4⁺CD8⁺T cell populations to 77.66% (Figure 5b), while the frequency of IFN- γ ⁺CD8⁺ (Supplementary Figure 23b) T cells had also increased to 45.81%, suggesting effective DC-mediated priming of antitumor T cells thereof. In addition, the secretion of key immune-related molecular markers in the co-incubation system was analyzed by ELISA assay to indicate the alteration in the immune composition, and the results revealed that the secretion levels of pro-inflammatory cytokines and chemokines including IFN- γ (Figure 5c), TNF- α (Figure 5d), CXCL10 (Figure 5e) and IL-2 (Figure 5f) in the Lip@AUR-ACP-aptPD-L1+IR group were the highest among all groups, which have increased by 4.2-fold, 6.3-fold, 4.3-fold and 5.5-fold compared to the control group, respectively. In contrast, the secretion levels of anti-inflammatory cytokines including IL-4 (Supplementary Figure 24a), IL-10 (Supplementary Figure 24b) and TGF- β (Supplementary Figure 24c) in the Lip@AUR-ACP-aptPD-L1+IR group were the lowest among all groups, which have decreased by 72%, 77% and 75% compared with the control group, respectively. Extending from the mechanistic evaluations above, we then systematically evaluated the antitumor efficacy of the liposome-augmented radio-immunotherapy using B16F10/mouse splenocyte co-incubation system. According to the flow cytometric data, the death rate of B16F10 cells in Lip@AUR-ACP-aptPD-L1+IR group reached around 76.78%, which was almost 9-fold higher than the PBS+IR group (Figure 5g). Consistently, MTT data showed that Lip@AUR-ACP-aptPD-L1+IR group presented the lowest B16F10 survival rate of only around 17.98% (Supplementary Figure 25). It is also of interest to note that B16F10 cells in the Lip@AUR-ACP-aptPD-L1+IR group

showed significantly elevated γ -H2AX levels, a typical marker of IR-induced DNA damage, according to immunochemical staining and western blotting analysis (Figure 5h and Supplementary Figure 26), again validating the therapeutic contribution of AUR-mediated radiosensitization. These observations are immediate evidence that the Lip@AUR-ACP-aptPD-L1 could enhance the radio-immunotherapeutic efficacy against melanoma cells in vitro through a programmable sequential manner.”

“As the combined result of these immunostimulatory traits, the Lip@AUR-ACP-aptPD-L1+IR treatment substantially enhanced the overall immune cell infiltration (CD45+) in the melanoma tissues by about 11.99% (Figure 7a). Specifically, the frequency of mature DCs (CD11c+CD80+CD86+/CD11c+MHC-II+) in the melanoma tissues in the Lip@AUR-ACP-aptPD-L1+IR group has increased by more than 30% compared with the control group (Figure 7b and Supplementary Figure 35a). Meanwhile, the Lip@AUR-aptPD-L1+IR group also showed drastically lower frequency of tumor-infiltrating immunosuppressive cells including Tregs (7.31%) (Supplementary Figure 34a) and MDSCs (1.78%) (Supplementary Figure 34b), which was in line with the VEGF-inhibiting function of AUR incorporated liposomes. Owing to the liposome mediated stimulation of DCs and inhibition of immunosuppressive cell populations, mice in the Lip@AUR-ACP-aptPD-L1+IR group showed enhanced tumor infiltration of CD4+/CD8+ T cells that was 37.78% higher than the control group (Figure 7c), accompanied with a significant expansion of IFN- γ +CD8+ T cells by 30.19% (Supplementary Figure 35b). The flow cytometric results regarding the tumor infiltration status of various immune cell populations were also consistently supported

by the immunofluorescence assay based on relevant markers and CCK8 assay of T cells (Figure 7d and Supplementary Figure 36-Supplementary Figure 37). Extending from the treatment-induced changes in the immunocomposition of the melanoma tissues, tumors in the Lip@AUR-ACP-aptPD-L1+IR group showed the highest enhancement in the secretion levels of pro-inflammatory cytokines and chemokines including IFN- γ (4.4-fold), TNF- α (6.2-fold), CXCL10 (4.5-fold) and IL-2 (5.6-fold) as well as the greatest reduction in the secretion levels of anti-inflammatory cytokines including IL-4 (25%), IL-10 (24%) and TGF- β (24%), indicating that the liposome-augmented radio-immunotherapy has significantly boosted the adaptive immune responses for eliminating the melanoma cells in vivo (Figure 7e-h and Supplementary Figure 38). In addition to the therapeutic evaluations above, we also comprehensively studied the biocompatibility of the liposomes in vivo from a translational perspective. Notably, mice receiving combinational liposome+IR treatment showed no significant weight loss compared to the PBS-only control group, which was attributed to the low toxicity of the liposomal formulations and the minimal IR dose (Supplementary Figure 39). Alternatively, histological inspections on the tissue slices of H&E-stained organs showed that Lip@AUR-ACP-aptPD-L1 did not induce obvious damage to major mouse organs regardless of the IR treatment conditions (Supplementary Figure 40). These results indicate that Lip@AUR-ACP-aptPD-L1 could be a safe and effective radio-immunotherapeutic option for melanomas.”

“Validation of AND-gate release of eCpG

B16F10 cells were inoculated into the confocal dish, and the initial cell density was 1×10^5 cells/dish. Subsequently, 40 $\mu\text{g}/\text{mL}$ Lip-AC^{Cy5}-aptPD-L1 or Lip-AC^{Cy5}P-aptPD-L1 was added into the above cells and mixed with 0 nM ATP+5 nM MMP-2, 100 nM ATP+5 nM MMP-2, 0 nM ATP+10 nM MMP-2 or 200 nM ATP+10 nM MMP-2 for 2 h. The B16F10 cells were washed with PBS and the cell nuclei were stained with DAPI for 10 min. Finally, the Cy5 fluorescence of eCpG was detected by CLSM.

For the flow cytometric analysis of eCpG release, B16F10 cells were inoculated into the 1.5 mL centrifuge tube with a density of 5×10^4 cells per tube. Then 40 $\mu\text{g}/\text{mL}$ Lip-AC^{Cy5}-aptPD-L1 or Lip-AC^{Cy5}P-aptPD-L1 was added into the above cells and treated with 0 nM ATP+5 nM MMP-2, 100 nM ATP+5 nM MMP-2, 0 nM ATP+10 nM MMP-2 or 200 nM ATP+10 nM MMP-2 for 2 h. The B16F10 cells were washed with PBS and then the Cy5 fluorescence of eCpG was quantified by flow cytometry.”

“Observation of eCpG release *in vivo*

B16F10 cells (1×10^6 cells) were injected subcutaneously into 6-week-old mice to establish B16F10-bearing C57BL/6J mouse model. The mice were cultured continuously until the tumor size reached 100 mm^3 , then treated with Lip@AUR-AC^{Cy5}-aptPD-L1 or Lip@AUR-AC^{Cy5}P-aptPD-L1 ($2 \text{ mg} \cdot \text{kg}^{-1}$) by intravenous injection. The time point of administration was defined as 0 h, while 4 Gy IR treatment was applied at 12 h post intravenous injection. The tumors were extracted after 16 h post intravenous injection, pulverized and filtered. The above cells were mixed with 5 mL red blood cell lysis buffer and stood for 10 min, and then centrifuged at $666.7 \times g$ for 5 min. Subsequently, the cells were first stained with CellTracker Green CMFDA for 15

min and then by DAPI for 10 min. After washing with PBS, single B16F10 cell was observed by laser confocal microscopy.

B16F10 cells (1×10^6 cells) were injected subcutaneously into 6-week-old mice to establish B16F10-bearing C57BL/6J mouse model. The mice were cultured continuously until the tumor size reached 100 mm^3 , then treated with Lip@AUR-AC^{Cy5}P-aptPD-L1 ($2 \text{ mg} \cdot \text{kg}^{-1}$) by intravenous injection. The time point of administration was defined as 0 h, while 4 Gy IR treatment was applied at 12 h post intravenous injection. The tumors were extracted at the time points of 0, 6, 12, 18, 24, 30 and 36 h, pulverized and filtered. The above cells were mixed with 5 mL red blood cell lysis buffer and stood for 10 min, and then centrifuged at $666.7 \times g$ for 5 min. Subsequently, the cells were stained with PC7-anti-CD45 antibodies for 30 min and washed with PBS. Finally, the Cy5 fluorescence of eCpG was quantified by flow cytometry.”

“Analysis of eCpG-mediated stimulation of DC maturation

B16F10 cells were inoculated into 12-well plates, and the initial cell density was 1×10^5 cells/well. When the cell confluence reached 80%, the co-culture system was established with the B16F10: splenocyte ratio of 1:10. Lip@AUR-ACP-aptPD-L1 was added into the co-incubation system and incubated for 12 h until 4 Gy IR was applied. At 12, 16, 20, 24, 30 or 36 h post liposome administration, the above cells were collected and added with PC7-anti-CD45 antibodies, APC-anti-CD11c antibodies, FITC-anti-CD80 antibodies and PE-anti-CD86 antibodies for 30 min. After washing with PBS, the DC maturation status was detected by flow cytometry.

B16F10 tumor cells (1×10^6 cells) were injected subcutaneously into 6-week-old mice to establish B16F10-bearing C57BL/6J mouse model. The mice were cultured continuously until the tumor size reached 100 mm^3 , afterwards they were treated with Lip@AUR-ACP-aptPD-L1 ($2 \text{ mg} \cdot \text{kg}^{-1}$) by intravenous injection. The time point of administration was defined as 0 h, while 4 Gy IR treatment was applied at 12 h post intravenous injection. The tumors were extracted at the time points of 0, 6, 12, 18, 24, 30 and 36 h, pulverized and filtered. The above cells were mixed with 5 mL red blood cell lysis buffer and stood for 10 min, and then centrifuged at $666.7 \times g$ for 5 min. Subsequently, the cells were stained with PC7-anti-CD45 antibodies, APC-anti-CD11c antibodies, FITC-anti-CD80 antibodies and PE-anti-CD86 antibodies for 30 min. After washing with PBS, the DC maturation status was detected by flow cytometry.”

Figure 4. Evaluation on AND-gate release of eCpG from Lip@AUR-ACP-aptPD-L1 for DC stimulation. (a) NUPACK analysis on eCpG secondary structure. (b)

Molecular docking analysis on the TLR9-binding behaviors of CpG ODN and eCpG.

(c-d) Molecular docking analysis on the specific binding sites of CpG ODN or eCpG to TLR9. (e) Flow cytometry analysis on the target binding of different DNA sequences to DCs in co-culture system. I: Control, II: CpG ODN, III: eCpG, IV: mutated CpG ODN, V: mutated eCpG, VI: closed eCpG. (f) Stimulatory impact of various DNA sequences to DC maturation by flow cytometry analysis in co-culture system. I: Control, II: CpG ODN, III: eCpG, IV: mutated CpG ODN, V: mutated eCpG, VI: closed eCpG.

(g) CLSM analysis regarding the effect of PNA complexation on eCpG^{Cy5} release from membrane-bound aptamer assemblies under different conditions. I: 0 nM ATP+5 nM MMP-2, II: 100 nM ATP+5 nM MMP-2, III: 0 nM ATP+10 nM MMP-2, IV: 200 nM ATP+10 nM MMP-2. (h) Quantitative flow cytometry analysis regarding eCpG^{Cy5} release from membrane-bound aptamer assemblies under different conditions. I: 0 nM ATP+5 nM MMP-2, II: 100 nM ATP+5 nM MMP-2, III: 0 nM ATP+10 nM MMP-2, IV: 200 nM ATP+10 nM MMP-2. (i) CLSM analysis regarding the effect of PNA complexation on eCpG^{Cy5} release from membrane-bound aptamer assemblies at 16 h incubation in vivo. (j) Time-dependent analysis on eCpG^{Cy5} release from Lip@AUR-AC^{Cy5}P-aptPD-L1 in vivo. 4 Gy IR treatment was applied at the time point of 12 h. (k) Time-dependent evaluation on DC maturation status (CD11c+CD80+CD86+) in the co-culture system after Lip@AUR-ACP-aptPD-L1 + 4 Gy IR treatment. (l-m) Time-dependent evaluation on DC maturation status (CD11c+CD80+CD86+) in the co-culture system after Lip@AUR-ACP-aptPD-L1 + 4 Gy IR treatment in vivo. (n) Treatment schedule for the B16F10-mouse splenocyte co-incubation system for the

evaluation of the immunostimulatory effects.

Figure 5. Immunostimulatory effect of combined Lip@AUR-ACP-aptPD-L1 and 4 Gy IR treatment in vitro. (a) Flow cytometry analysis on the maturation status (CD11c+CD80+CD86+) of DCs in the co-incubation system after different treatments. (b) Flow cytometry analysis on T cell activation status (CD3+CD4+CD8+) in the co-incubation system after different treatments. (c-f) Secretion levels of immunostimulatory markers including IFN- γ , TNF- α , CXCL10 and IL-2 in the supernatants of the co-culture system after different treatments. (g) Flow cytometry analysis on the apoptosis of B16F10 cells after different treatments in co-culture system. (h) Fluorescence microscopy images of γ -H2AX staining in B16F10 cells.

(h) γ -H2AX immunofluorescence of IR-treated B16F10 cells after different sample treatments. I: PBS, II: Lip, III: Lip-aptPD-L1, IV: Lip-ACP-aptPD-L1, V: Lip@AUR-aptPD-L1, VI: Lip@AUR-ACP-aptPD-L1.

Figure 7. Mechanism underlying Lip@AUR-ACP-aptPD-L1-augmented radio-immunotherapy in vivo. (a-c) Flow cytometry analysis on the infiltration of total immune cells (CD45+), DCs (CD11c+CD80+CD86+) and effector T cells (CD3+CD4+CD8+) at the tumor site after treatment with I: PBS, II: Lip, III: Lip-aptPD-L1, IV: Lip-ACP-aptPD-L1, V: Lip@AUR-aptPD-L1, VI: Lip@AUR-ACP-

aptPD-L1. (d) Immunofluorescence images of the extracted tumors showing infiltration of CD8⁺T cells after treatment with I: PBS, II: Lip, III: Lip-aptPD-L1, IV: Lip-ACP-aptPD-L1, V: Lip@AUR-aptPD-L1, VI: Lip@AUR-ACP-aptPD-L1. (e-h) The secretion levels of IFN- γ , TNF- α , CXCL10 and IL-2 in mouse serum after treatment with I: PBS, II: Lip, III: Lip-aptPD-L1, IV: Lip-ACP-aptPD-L1, V: Lip@AUR-aptPD-L1, VI: Lip@AUR-ACP-aptPD-L1.

Supplementary Figure 21. AND-gate ACP activation from Lip-AC^{Cy5}P-aptPD-L1+ 4 Gy IR treated B16F10 cells. (a) The release of eCpG^{Cy5} from B16F10 cell surface by CLSM imaging. (b) The release of eCpG^{Cy5} from B16F10 cell surface by fluorescence spectroscopy.

Supplementary Figure 22. Treatment-induced ICD of B16F10 cells in the co-culture system. (a) The HMGB1 expression in the B16F10 cell nucleus (red fluorescence). (b) The CRT expression on cell membrane (green fluorescence). I: PBS, II: Lip, III: Lip-aptPD-L1, IV: Lip-ACP-aptPD-L1, V: Lip@AUR-aptPD-L1, VI: Lip@AUR-ACP-aptPD-L1.

Supplementary Figure 23. Treatment-induced stimulatory effect on the adaptive immune system. (a) Frequencies of DCs (CD11c⁺MHC-II⁺) in the co-culture system after treatment with different groups. (b) Frequencies of cytotoxic CD8⁺T cells (CD8a⁺IFN- γ ⁺) in the co-culture system after treatment with different groups. I: PBS, II: Lip, III: Lip-aptPD-L1, IV: Lip-ACP-aptPD-L1, V: Lip@AUR-aptPD-L1, VI: Lip@AUR-ACP-aptPD-L1.

Supplementary Figure 24. Evaluation on the treatment-induced changes in the secretion of anti-inflammatory factors. The secretion levels of various anti-inflammatory factors including (a) IL-4, (b) IL-10 and (c) TGF-β after treatment with different materials in the co-culture system. I: PBS, II: Lip, III: Lip-aptPD-L1, IV: Lip-ACP-aptPD-L1, V: Lip@AUR-aptPD-L1, VI: Lip@AUR-ACP-aptPD-L1.

Supplementary Figure 25. MTT assay on the radio-immunotherapeutic effect of the liposomes in vitro. B16F10 inhibition effect of different groups (a) without or (b) with 4 Gy IR under the co-culture condition by MTT assay. I: Lip, II: Lip-aptPD-L1, III: Lip-ACP-aptPD-L1, IV: Lip@AUR-aptPD-L1, V: Lip@AUR-ACP-aptPD-L1.

Supplementary Figure 26. Radiosensitization effect of different samples on B16F10 cells in vitro by detecting γ -H2AX immunofluorescence. The group set-ups are I: PBS, II: Lip, III: Lip-aptPD-L1, IV: Lip-ACP-aptPD-L1, V: Lip@AUR-aptPD-L1, VI: Lip@AUR-ACP-aptPD-L1.

Supplementary Figure 34. In vivo evaluation on the tumor-residing immunosuppressor cell populations after various treatments. (a) Frequencies of Tregs (CD4+CD25+CTLA-4+) in B16F10 tumor tissues after different treatments. (b) Frequencies of MDSCs (CD11b+GR1+) in B16F10 tumor tissues after different treatments. I: PBS, II: Lip, III: Lip-@AUR, IV: Lip@AUR-aptPD-L1.

Supplementary Figure 35. Treatment-induced remodeling of B16F10 tumor immune microenvironment. (a) Frequencies of DCs (CD11c⁺MHC-II⁺) in B16F10 tumor tissues after different treatments. (b) Frequencies of effector CD8⁺T cells (CD8a⁺IFN- γ ⁺) in B16F10 tumor tissues after different treatments. I: PBS, II: Lip, III: Lip-aptPD-L1, IV: Lip-ACP-aptPD-L1, V: Lip@AUR-aptPD-L1, VI: Lip@AUR-ACP-aptPD-L1.

Supplementary Figure 36. T cell infiltration in B16F10 tumor tissues. (a) Immunofluorescence staining of CD4+T cells in B16F10 tumor tissues after different groups treatment. (b) Immunofluorescence staining of IFN- γ + CD8+T cells in B16F10 tumor tissues after different groups treatment. I: PBS, II: Lip, III: Lip-aptPD-L1, IV: Lip-ACP-aptPD-L1, V: Lip@AUR-aptPD-L1, VI: Lip@AUR-ACP-aptPD-L1.

Supplementary Figure 37. Detection of CD3+T cell infiltration in B16F10 tumors

treated with different materials by CCK8 assay. The group set-ups include I: PBS, II: Lip, III: Lip-aptPD-L1, IV: Lip-ACP-aptPD-L1, V: Lip@AUR-aptPD-L1, VI: Lip@AUR-ACP-aptPD-L1.

Supplementary Figure 38. Evaluation on the post-treatment secretion levels of anti-inflammatory factors in vivo. Secretion of various anti-inflammatory factors including (a) IL-4, (b) IL-10 and (c) TGF- β after different treatment in B16F10 tumor-bearing mice. I: PBS, II: Lip, III: Lip-aptPD-L1, IV: Lip-ACP-aptPD-L1, V: Lip@AUR-aptPD-L1, VI: Lip@AUR-ACP-aptPD-L1.

Supplementary Figure 39. Body weight analysis of the mouse models.

Biocompatibility analysis by monitoring the body weight changes of B16F10 tumor-bearing mouse after different treatments. I: PBS, II: Lip, III: Lip-aptPD-L1, IV: Lip-

ACP-aptPD-L1, V: Lip@AUR-aptPD-L1, VI: Lip@AUR-ACP-aptPD-L1, VII:
PBS+IR, VIII: Lip+IR, IX: Lip-aptPD-L1+IR, X: Lip-ACP-aptPD-L1+IR, XI:
Lip@AUR-aptPD-L1+IR, XII: Lip@AUR-ACP-aptPD-L1+IR.

Supplementary Figure 40. Histopathological analysis on the biocompatibility of

the liposome-augmented radio-immunotherapy in vivo. Histological analysis on the biocompatibility of the liposome-augmented radio-immunotherapy in vivo according to H&E staining of mouse major organs after treatment with I: PBS, II: Lip, III: Lip-aptPD-L1, IV: Lip-ACP-aptPD-L1, V: Lip@AUR-aptPD-L1, VI: Lip@AUR-ACP-aptPD-L1.

5) For results of in vivo experiments, figures with multiple groups make it hard to interpret which group should readers focus. Why groups without X-ray results are included in all the figures? Are there any reasons readers should go parallel with – X-ray groups?

A: Thank you for the insight. We agree with the reviewer's concern that including multiple groups under two IR treatment conditions would significantly increase the complexity of the figures. However, we would like to explain that this is a compromised approach between intuitiveness and scientific informativity. The 12 groups in the in vivo experiment are all selected with their own purpose. For instance, the comparison between Lip and PBS group showed the nontoxicity of the liposomal carriers, while the Lip-aptPD-L1 was added to highlight the importance of aptPD-L1-mediated melanoma targeting for robust drug delivery. The comparison of Lip@AUR-ACP-aptPD-L1 and Lip-ACP-aptPD-L1 showed the necessity of AUR for improving the radio-immunotherapy, while the superior therapeutic performance of Lip@AUR-ACP-aptPD-L1 over Lip@AUR-aptPD-L1 confirmed the importance of ACP-mediated DC stimulation for eliciting robust antitumor immunity. Meanwhile, we also would like to

explain that it is necessary to comparatively analysis the therapeutic responses under IR-present and IR-absent conditions as the IR-triggered biochemical changes (ATP release and MMP-2 upregulation) in melanoma cells are crucial for triggering the AND-gate operation of membrane-bound ACP to release eCpG. For instance, the data in Figure 3b-g and Figure 4i showed that the membrane-bound ACP would be eventually transferred to the cytoplasmic compartment in the absence the spatiotemporally coordinated IR exposure, thus failing to stimulate the maturation of DCs in the TME. To facilitate the extraction of information of interest from the figures, we have processed the data of the relevant in vivo experiment and compiled them into histograms, allowing for the direct comparison and analysis of the numerical results in an intuitive manner. We hope that the explanations above may address your concerns regarding the group set-up in our in vivo experiment.

Figure 6. Anti-tumor effects of Lip@AUR-ACP-aptPD-L1-augmented radio-

immunotherapy in vivo. (a) Schematic representation of the treatment protocol for B16F10-luc tumor-bearing mice. (b) In vivo bioluminescence images of B16F10-Luc tumor-bearing mice during treatment (n=5). (c) Tumor volume analysis throughout the treatment period (n=5). I: PBS, II: Lip, III: Lip-aptPD-L1, IV: Lip-ACP-aptPD-L1, V: Lip@AUR-aptPD-L1, VI: Lip@AUR-ACP-aptPD-L1, VII: PBS+IR, VIII: Lip+IR, IX: Lip-aptPD-L1+IR, X: Lip-ACP-aptPD-L1+IR, XI: Lip@AUR-aptPD-L1+IR, XII: Lip@AUR-ACP-aptPD-L1+IR. (d) Tumor weight analysis at the end of the treatment period (n=5). I: PBS, II: Lip, III: Lip-aptPD-L1, IV: Lip-ACP-aptPD-L1, V: Lip@AUR-aptPD-L1, VI: Lip@AUR-ACP-aptPD-L1. (e) Survival analysis of mice after different treatments (n=6). I: PBS, II: Lip, III: Lip-aptPD-L1, IV: Lip-ACP-aptPD-L1, V: Lip@AUR-aptPD-L1, VI: Lip@AUR-ACP-aptPD-L1, VII: PBS+IR, VIII: Lip+IR, IX: Lip-aptPD-L1+IR, X: Lip-ACP-aptPD-L1+IR, XI: Lip@AUR-aptPD-L1+IR, XII: Lip@AUR-ACP-aptPD-L1+IR. (f) Western blotting on the expression levels of related proteins in the tumor tissues. (g) The release level of ATP and MMP-2 in B16F10 tumors after different treatment (n=3). I: PBS+IR, II: Lip+IR, III: Lip@AUR+IR, IV: Lip@AUR-aptPD-L1+IR. (h) TUNEL staining of tumor tissue samples after treatment. I: PBS, II: Lip, III: Lip-aptPD-L1, IV: Lip-ACP-aptPD-L1, V: Lip@AUR-aptPD-L1, VI: Lip@AUR-ACP-aptPD-L1.

Reviewer #3 (Remarks to the Author):

This manuscript reports a cascade-amplification strategy based on multivariate-aptamer

assembly modified liposomes (Lip@AUR-ACP-aptPD-L1) to enhance the radio-immunotherapy efficiency of melanoma. In multivariate aptamer assembly, PD-L1 aptamers guide the targeted fusion with tumor cell and delivery AUR to induce immunogenic death of melanoma cells; the released ATP and upregulated MMP2 by damaged cells cause AND-gate activation of ACP, thus triggering the in-situ release of CpG-based immunoadjuvants for stimulating dendritic cell-mediated T cell priming; AUR also inhibit tumor-intrinsic ERK1/2-HIF-1 α -VEGF signaling to suppress infiltration of immunosuppressive cells for fostering an anti-tumorigenic TME. The cascade amplification and synergistic effects of multivariate liposome probe provided a promising tool for the clinical treatment of melanoma. Overall, it should be suitable for publication at Nat. Commun. after major revision.

A: Thank you for the appreciation of our work. We're grateful for your insightful and constructive comments and suggestions as they greatly enhance the scientific quality of this research. We ensure that all your concerns have been properly addressed in the revised manuscript, the detailed responses and discussions are shown here below point-by-point for your reference.

The major concerns about the study were listed as following:

1. The stability of DNA assembly is a major obstacle for immunotherapy application.

The Lip@AUR-ACP-aptPD-L1 probes were incubated in cells and within bodies for more than ten hours. How to guarantee the intactness and stability of DNA probes, under complicated enzyme conditions?

A: Thank you for the advice. We totally agree with the reviewer on the importance of aptamer stability for ensuring their therapeutic efficacy and consistence in clinically relevant conditions and monitored the degradation behavior of the liposome-integrated ACP in biomimetic buffer. Typically, the strong interaction among individual components in the ACP assemblies could significantly enhance the stability of the nucleic acid-based materials in physiological environment. As the PNA sequences are neutrally charged in aqueous environment, the integrity of the ACP assemblies could not be measured directly using PAGE assay. Therefore, we have proactively labeled the eCpG species with Cy5 for constructing the liposomes (denoted as Lip@AUR-AC^{Cy5}P-aptPD-L1), where changes in the fluorescent intensity could thus be used to indicate the degradation status of the liposome under different incubation conditions. Here the Lip@AUR-AC^{Cy5}P-aptPD-L1 was incubated in 10% FBS or DNase1 solutions that mimic the complex enzyme conditions, where the degradation rates of the ACP assemblies were calculated by measuring the Cy5 fluorescence in the supernatant. As shown in Figure 2g, after 30 h of incubation, the degradation rate of the ACP assemblies in 10% FBS or DNase1 solutions was only around 19.21% or 14.31%, immediately suggesting their stability under clinically relevant conditions that is beneficial for maintaining their therapeutic activity in the complex in vivo environment.

The related data and discussions are shown here below for your review:

“Alternatively, quantitative fluorescence analysis showed that the degradation rate of ACP in Lip@AUR-ACP-aptPD-L1 was only 19.21% or 14.31% after incubation in 10% FBS or DNase 1 for 30 h (Figure 2g). These data confirmed the relative stability of

Lip@AUR-ACP-aptPD-L1 in biomimetic buffers, which is beneficial for improving the therapeutic index of AUR and ACP after systemic administration.”

“DNA degradation assay of Lip@AUR-ACCy5P-aptPD-L1

According to the experimental procedure in previous reports^{54, 59}, Lip@AUR-ACCy5P-aptPD-L1 was synthesized and placed in the PBS with 10% FBS or DNase 1. The supernatant was obtained at different time points using an ultrafiltration tube (MWCO: 5000), afterwards the fluorescence intensity of eCpG in the supernatant was detected by a fluorescence spectrophotometer.”

Figure 2. Physicochemical characterization of the fusogenic liposomes. (a) Preparation process and lipid composition of Lip@AUR-ACP-aptPD-L1. (b) DNA-PAGE analysis regarding eCpG release from aptATP/eCpG complex in response to different ATP concentrations. (c) Impact of competitive ATP binding on aptATP/eCpG complex via DNA-PAGE analysis (aptATP: eCpG=2:1). (d) DNA-PAGE analysis regarding eCpG release from the ACP assembly (aptATP: eCpG: PNA=2:1:3) with 200 nM ATP and 5 nM or 10 nM MMP-2. (e) TEM results of Lip@AUR-ACP-aptPD-L1 stained with 4% phosphotungstic acid. (f) The stability of Lip@AUR-ACP-aptPD-L1 in pH7.4 PBS buffer at 12-48 h by DLS analysis. (g) Degradation assessment of Lip@AUR-ACP-aptPD-L1 in DNase 1 or 10% FBS through 50 h incubation. (h) DNA-PAGE analysis of eCpG release from Lip@AUR-ACP-aptPD-L1 with 200 nM ATP and 5 nM or 10 nM MMP-2. (i) Fluorescence analysis of eCpG^{Cy5} release from Lip@AUR-ACP-aptPD-L1 with or without ATP (200 nM) and MM) stimulus input. I: ATP-/MMP-2-, II: ATP-/MMP-2+, III: ATP+/MMP-2-, IV: ATP+/MMP-2+. (j) Fluorescence analysis of eCpG^{Cy5} release from different liposome formulations under different ATP concentrations. I: Lip@AUR-AC^{Cy5}-aptPD-L1, II: Lip@AUR-AC^{Cy5}P-aptPD-L1, III: Lip@AUR-AC^{Cy5}P-aptPD-L1+MMP-2 (5 nM), IV: Lip@AUR-AC^{Cy5}P-aptPD-L1+MMP-2 (10 nM). (k) Summarization of the physicochemical properties of the intermediates and final product of the liposomes.

2. The characterization of multivariate aptamer assembly, such as the construction of Lip@AUR-ACP-aptPD-L1, the cascading sequence release and amplification, was

based on PAGE assay, which is not clear and quantitative. We will suggest further characterization using fluorescent labeled DNA aptamer or linker sequence, to monitor the multivariate aptamer assembly and quantitatively analyze the release efficiency of each aptamer units.

A: Thank you for the advice. Based on the insights you kindly provided, we have proactively labeled eCpG and PNA with Cy5 for constructing the aptamer assemblies. Notably, the molecular complex of aptATP and Cy5-labeled eCpG was denoted as AC^{Cy5}, while the molecular complex of aptATP, eCpG and Cy5-labeled PNA was denoted ACP^{Cy5}. Therefore, the fluorescent intensity of the samples could be used to quantitatively determine the assembly efficiency via the standard calibration method. As shown in Supplementary Figure 4c, the assembly processes for AC^{Cy5} and ACP^{Cy5} were highly efficient, which were around 97.07% and 95.52%, respectively. Results of the quantitative fluorescence analysis were consistent with the visual trends in PAGE assay that collectively validate the efficient ACP assembly process through self-regulated binding of complementary sequences. Meanwhile, we used ACP^{Cy5} and FAM-labeled aptPD-L1 to synthesize Lip@AUR-AC^{Cy5}P-aptPD-L1 and Lip@AUR-ACP-aptPD-L1^{FAM}, based on which the loading efficiency of ACP and aptPD-L1 could also be obtained via standard curve calibration. As shown in Supplementary Figure 5, the loading efficiency of ACP and aptPD-L1 into the fusogenic liposomes was around 86.5% and 81.0%, respectively.

To investigate the AND-gate eCpG release from the liposomes, we incubated Lip@AUR-AC^{Cy5}P-aptPD-L1 with 200 nM ATP and 10 nM MMP-2 to mimic the

exposure to IR-treated melanoma TME, afterwards the Cy5 fluorescence intensity in the supernatant was used to determine the eCpG release profiles via standard curve calibration. As shown in Supplementary Figure 4d, the accumulative eCpG release ratio increased gradually with time and reached around 92.48% after 2 h of incubation, confirming the rapid AND-gate activation of the ACP assemblies to the combinational ATP/MMP-2 inputs.

The related data and discussions are shown here below for your review:

“Quantitative fluorescence analysis showed that aptATP with eCpG has a complexation efficiency of 97.07% (Supplementary Figure 4c), which was attributed to the molecular specificity of the complementary sequences thereof. The ATP-responsiveness of aptATP/eCpG complex was further profiled by PAGE assay, which showed that treating aptATP/eCpG complexes with an ATP concentration of 0.05 μ M was sufficient to induce significant eCpG release (Figure 2c), emphasizing the necessity for the implementation of the AND gate eCpG release function to avoid premature eCpG leakage at background concentrations. Next, the aptATP/eCpG complexes were sequentially integrated with PNA at an aptATP: PNA ratio of 1:1.5, leading to the formation of duplex structures (ACP) with robust stability under physiological conditions. Similarly, quantitative fluorescence analysis showed that the assembly efficiency of ACP with aptATP, eCpG and PNA was 95.52% (Supplementary Figure 4c), indicating that the assembly process is highly modular with molecular precision.”

“Specifically, ICP and quantitative fluorescence analysis showed that the AUR could be efficiently loaded into the fusogenic liposomes at a high efficiency of around 88.29%,

and the relative AUR ratio in the final Lip@AUR-ACP-aptPD-L1 was around 4.98% (Figure 2k, Supplementary Figure 2d-e and Supplementary Figure 3). Due to the presence of the cholesterol tails, the liposomal integration of ACP assembly and aptPD-L1 was highly efficient with a loading efficiency of 86.5% and 81.0%, respectively, while the average number of ACP assembly and aptPD-L1 on a single liposome was 109 and 51 based on fluorescence spectroscopy (Supplementary Figure 5).”

“To test if the multivariate AND-gate operation of the ACP assembly was maintained after the integration into Lip@AUR-ACP-aptPD-L1, the liposomes were processed with different stimulus inputs and the corresponding eCpG release was monitored by DNA-PAGE analysis or fluorescence spectroscopic analysis. Consistent with the observations of vehicle-free ACP assemblies in Figure 2d, the combinational treatment of 200 nM ATP and 10 nM MMP-2 induced efficient eCpG release from Lip@AUR-ACP-aptPD-L1 according to the DNA-PAGE analysis (Figure 2h), which was also supported by the results of the quantitative fluorescence spectroscopic analysis that eCpG release rate from Lip@AUR-AC^{Cy5}P-aptPD-L1 reached around 92.48% after 2 h incubation with 200 nM ATP and 10 nM MMP-2 (Supplementary Figure 4d).”

“Evaluation of ACP assembly construction

The standard curves of Cy5 labeled eCpG and PNA were obtained by fluorescence spectroscopy. According to the previously experimental procedure, AC^{Cy5} and ACP^{Cy5} were assembled with 4 nM aptATP, 2 nM eCpG and 6 nM PNA and then filtered using an ultrafiltration tube (MWCO: 8000). Finally, the fluorescence intensity of Cy5 was detected by fluorescence spectroscopy.”

“eCpG release patterns from Lip@AUR-ACP-aptPD-L1

The standard curves of Cy5 labeled eCpG was obtained by fluorescence spectroscopy. According to the previously reported experimental procedure^{54, 59}, Lip@AUR-ACCy5P-aptPD-L1 was assembled with 4 nM aptATP, 2 nM eCpG and 6 nM PNA, followed by incubation with 200 nM ATP and 10 nM MMP-2. The supernatant was obtained at different time points using an ultrafiltration tube (MWCO: 8000), afterwards the fluorescence intensity of Cy5 in the supernatant was detected by fluorescence spectroscopy.”

“Loading analysis of eCpG and aptPD-L1

The synthesis of fluorescently labeled liposomes was generally the same with those fluorescence-free ones except that the original aptamers were replaced by Cy5-labeled eCpG or FAM-labeled aptPD-L1, leading to the formation of Lip@AUR-ACCy5P-aptPD-L1 or Lip@AUR-ACP-aptPD-L1^{FAM}. 56 μL Lip@AUR-ACCy5P-aptPD-L1 ($5\text{mg}\cdot\text{mL}^{-1}$) or Lip@AUR-ACP-aptPD-L1^{FAM} ($5\text{mg}\cdot\text{mL}^{-1}$) aqueous solution was added to $1\times$ DNase 1 buffer solution and then treated with $20\text{ U}\cdot\text{mL}^{-1}$ DNase 1, incubated at 37°C for 15 min and transferred to an ultrafiltration tube. After centrifugation at 10,000 rpm for 15 min, the supernatant was collected and fluorescence intensity of Cy5 or FAM was detected by fluorescence spectroscopy. eCpG^{Cy5} or aptPD-L1^{FAM} solution with different concentrations were configured to establish the standard curves via a fluorescence spectrophotometer. The aptamer concentrations in Lip@AUR-ACCy5P-aptPD-L1 or Lip@AUR-ACP-aptPD-L1^{FAM} was quantified according to the standard curve, and then the load efficiency of eCpG^{Cy5} or aptPD-L1^{FAM} on liposomes was

calculated accordingly.”

Supplementary Figure 4. Evaluation of aptamer assembly and eCpG release. (a-b) Standard curve of eCpG^{Cy5} and PNA^{Cy5} according to fluorescence analysis. (c) Assembly efficiency of AC^{Cy5} and ACP^{Cy5}. (d) Release efficiency of eCpG^{Cy5} from Lip@AUR-AC^{Cy5}P-aptPD-L1 in a time-dependent manner.

Supplementary Figure 5. Integration efficiency of ACP and aptPD-L1 in

Lip@AUR-ACP-aptPD-L1. (a-b) Fluorescence spectra of eCpG^{Cy5} and aptPD-L1^{FAM} under different concentrations. (c) Schematic demonstration for the calculation of average number of aptamers in the liposomes. Notable parameters are described in the figure text.

3. The loading efficiency of AUR in liposomes was calculated to ~5%, which is extremely low. Have the authors attempted to optimize the concentration of AUR and increase the loading efficiency? How to achieve high radio-immunotherapy efficiency with such less AUR?

A: Thank you for the correction. We would like to first apologize for the miswording here as it was the relative AUR loading amount that was around 4.98%, while the actual loading efficiency was around 88.29%. The AUR loading in the fusogenic liposomes were validated using both ICP and fluorescence spectroscopy and the results were highly consistent. We also would like to note that the AUR loading amount in the liposomes was at a comparable level to the values in recent reports on AUR-based antitumor therapies^{13, 14, 15, 16}. Furthermore, therapeutic evaluations also confirmed that current AUR loading condition was sufficient to confer effective radiosensitization of melanoma cells as well as reducing the recruitment of immunosuppressor cell populations through inhibiting the ERK1/2-HIF-1 α -VEGF pathway. We are really sorry for the careless mistakes and the corrected descriptions and data are shown here below for your review:

The related data is shown here below for your review:

Supplementary Figure 2. Physical and chemical properties of the liposome sample series. (a-c) The hydrodynamic size, polydispersity index and zeta potential of liposomes by DLS. (d) The standard curve with AUR. (e) ICP-dependent determination of AUR content in liposomes.

Supplementary Figure 3. Fluorescence analysis of AUR loading into the liposomes.

(a) The fluorescence values of AUR at different concentrations and (b) the corresponding standard curve.

4. The uptake of liposomes into cells is commonly existed. How to demonstrate the targeted fusion guided by PD-L1 aptamer? Figure 3b is not sufficient. A quantitative and face-to-face comparison with (targeted fusion) and without (random fusion) PD-L1 aptamer should be conducted.

A: Thank you for the insights. We totally agree with your concerns regarding the liposome uptake experiment due to the lack of crucial control groups and have redone the tests using a new group set-up as requested, which include Lip@Dil, Lip@Dil-ACP and Lip@Dil-ACP-aptPD-L1. As shown in Figure 3b and Supplementary Figure 6-Supplementary Figure 8, both Lip@Dil and Lip@Dil-ACP showed only modest uptake by B16F10 cells according to the CLSM results in the co-culture system. It is notable that the uptake of Lip@Dil-ACP was even lower than Lip@Dil, which was attributed to the electrostatic repulsion between the negatively charged ACP and cytoplasmic membrane. Notably, Lip@Dil-ACP-aptPD-L1 showed evidently superior uptake by B16F10 cells than both Lip@Dil and Lip@Dil-ACP in the co-culture system, evidently

confirming the contribution of the melanoma-targeting aptPD-L1 components on liposome uptake. This is also consistently supported by the findings in recent reports that the presence of cell-binding ligands could overcome the negative impact of the electrostatic repulsion and facilitate the fusion of the liposomes and cytoplasmic membrane through enhancing the direct interaction in between^{4, 5, 6, 7}. Overall, the observations above collectively demonstrated that modifying the fusogenic liposomes with aptPD-L1 could offer melanoma-specific binding affinity to accelerate membrane fusion between liposomes and the targeted melanoma cells.

The related data and discussions are shown here below for your review:

“As shown in Figure 3b and Supplementary Figure 6-Supplementary Figure 8, both Lip@Dil and Lip@Dil-ACP showed low fusogenic capacity according to the CLSM results. It is notable that the fusogenic capacity of Lip@Dil-ACP was even lower than Lip@Dil, which was attributed to the electrostatic repulsion between the negatively charged ACP and cytoplasmic membrane that hinders the interaction of liposomal and cytoplasmic membranes. Notably, Lip@Dil-ACP-aptPD-L1 showed evidently superior fusogenic capacity than both Lip@Dil and Lip@Dil-ACP in the co-culture system, ascribing to the integration of melanoma-targeting aptPD-L1 components. The aptPD-L1-boosted membrane fusion of liposomes is consistent with recent findings that the presence of cell-binding ligands could facilitate the fusion of the liposomes and cytoplasmic membrane through enhancing the direct interaction in between^{58, 59, 60, 61}.”

“Observation of membrane fusion behavior of the liposomes

Herein, orange-red probe Dil was loaded into the liposome instead of AUR for

fluorescence tracking, of which the samples were denoted as Lip@Dil, Lip@Dil-ACP and Lip@Dil-ACP-aptPD-L1. B16F10 cells were inoculated into confocal dishes at a density of 1×10^5 cells/well. When the cell confluence reached 80%, the co-culture system was established with the B16F10: splenocyte ratio of 1:10. 40 $\mu\text{g}/\text{mL}$ Lip@Dil, Lip@Dil-ACP or Lip@Dil-ACP-aptPD-L1 was added respectively into the co-culture system. At 3, 6, 12 and 18 h of incubation, cell samples were washed with PBS to remove the floating spleen cells, while the remaining B16F10 cells were first stained with CellTracker Green CMFDA for 15 min and then by DAPI for 10 min. Finally, the membrane fusion status of the liposomes with B16F10 cells was detected by laser confocal microscopy.

In addition, B16F10 cells were inoculated into confocal dishes at a density of 1×10^5 cells/well. When the cell confluence reached 80%, the co-culture system was established with the B16F10: splenocyte ratio of 1:10. 40 $\mu\text{g}/\text{mL}$ Lip@Dil, Lip@Dil-ACP or Lip@Dil-ACP-aptPD-L1 was added separately into the co-culture system. After 12 h incubation, the above cells were treated with 4 Gy IR. At 16, 18 and 30 h of incubation, while the remaining B16F10 cells were first stained with CellTracker Green CMFDA for 15 min and then by DAPI for 10 min. Finally, the membrane fusion status of the liposomes with B16F10 cells was detected by laser confocal microscopy.”

Figure 3. Melanoma targeting and membrane fusion performance of Lip@AUR-ACP-aptPD-L1 in vitro based on B16F10/splenocyte co-incubation system. (a) Targeting ability of eCpG and aptPD-L1 with the designated cell targets in the B16F10/splenocyte co-incubation system by flow cytometry. **(b)** Fusion status of

different liposomes to B16F10 cell membranes in the co-culture system at 3, 6, 12 or 18 h of incubation by CLSM. I: Lip@Dil, II: Lip@Dil-ACP, III: Lip@Dil-ACP-aptPD-L1. Red: Dil. Green: CellTracker Green CMFDA. Blue: DAPI. (c) Tumor sphere assay on the targeting ability of different samples at 12 h incubation. I: AC^{Cy5}P, II: Lip-AC^{Cy5}P, III: Lip-AC^{Cy5}P-aptPD-L1. (d-e) Time-dependent changes in ATP and MMP-2 abundance in vivo with Lip@AUR-ACP-aptPD-L1 + 4 Gy IR treatment. (f) Fusion status of different liposomes to B16F10 cell membranes in the co-culture system at 16, 18 or 30 h of incubation by CLSM. 4 Gy IR treatment was applied at 12 h. I: Lip@Dil+IR, II: Lip@Dil-ACP+IR, III: Lip@Dil-ACP-aptPD-L1+IR. Red: Dil. Green: CellTracker Green CMFDA. Blue: DAPI. (g) Time-dependent melanoma-targeted membrane fusion performance of Cy5-Lip@Dil-ACP-aptPD-L1 in vivo. 4 Gy IR treatment was applied after 12 h post intravenous injection. (h) Schedule of the combinational Lip@Dil-ACP-aptPD-L1 + 4 Gy IR treatment set-up in vitro. Statistical analysis in panels d-e was carried out via one-way ANOVA method. * indicates significance at $p < 0.05$, *** indicates significance at $p < 0.001$, **** indicates significance at $p < 0.0001$.

Supplementary Figure 6. Fluorescence analysis on the membrane fusion of the fusogenic liposomes. Evaluation on the membrane fusion performance of Lip@Dil with B16F10 cells and spleen cells in the co-culture system.

Supplementary Figure 7. Fluorescence analysis on the membrane fusion of the ACP-integrated liposomes. Membrane fusion of Lip@Dil-ACP in the co-culture system with B16F10 cells and spleen cells.

Supplementary Figure 8. Fluorescence analysis on the membrane fusion of the ACP-integrated melanoma-targeted liposomes. Membrane fusion of Lip@Dil-ACP-aptPD-L1 in the co-culture system with B16F10 cells and spleen cells.

5. The caption of each figure is too simple. Some necessary description and explanation should be added to assist the reading of figures.

A: Thank you for the correction. Based on your advice we have added the necessary details into the figure captions to enhance its informativity by covering the important experimental set-ups and explanations. Some of the representative examples are shown here below for your review:

The related legends are shown here below for your review:

“Figure 2. Physicochemical characterization of the fusogenic liposomes. (a)

Preparation process and lipid composition of Lip@AUR-ACP-aptPD-L1. (b) DNA-PAGE analysis regarding eCpG release from aptATP/eCpG complex in response to different ATP concentrations. (c) Impact of competitive ATP binding on aptATP/eCpG complex via DNA-PAGE analysis (aptATP: eCpG=2:1). (d) DNA-PAGE analysis regarding eCpG release from the ACP assembly (aptATP: eCpG: PNA=2:1:3) with 200 nM ATP and 5 nM or 10 nM MMP-2. (e) TEM results of Lip@AUR-ACP-aptPD-L1 stained with 4% phosphotungstic acid. (f) The stability of Lip@AUR-ACP-aptPD-L1 in pH7.4 PBS buffer at 12-48 h by DLS analysis. (g) Degradation assessment of Lip@AUR-ACP-aptPD-L1 in DNase 1 or 10% FBS through 50 h incubation. (h) DNA-PAGE analysis of eCpG release from Lip@AUR-ACP-aptPD-L1 with 200 nM ATP and 5 nM or 10 nM MMP-2. (i) Fluorescence analysis of eCpG^{Cy5} release from Lip@AUR-ACP-aptPD-L1 with or without ATP (200 nM) and MM) stimulus input. I: ATP-/MMP-2-, II: ATP-/MMP-2+, III: ATP+/MMP-2-, IV: ATP+/MMP-2+. (j) Fluorescence analysis of eCpG^{Cy5} release from different liposome formulations under different ATP concentrations. I: Lip@AUR-AC^{Cy5}-aptPD-L1, II: Lip@AUR-AC^{Cy5}P-aptPD-L1, III: Lip@AUR-AC^{Cy5}P-aptPD-L1+MMP-2 (5 nM), IV: Lip@AUR-AC^{Cy5}P-aptPD-L1+MMP-2 (10 nM). (k) Summarization of the physicochemical properties of the intermediates and final product of the liposomes.

Figure 3. Melanoma targeting and membrane fusion performance of Lip@AUR-ACP-aptPD-L1 in vitro based on B16F10/splenocyte co-incubation system. (a) Targeting ability of eCpG and aptPD-L1 with the designated cell targets in the B16F10/splenocyte co-incubation system by flow cytometry. (b) Fusion status of

different liposomes to B16F10 cell membranes in the co-culture system at 3, 6, 12 or 18 h of incubation by CLSM. I: Lip@Dil, II: Lip@Dil-ACP, III: Lip@Dil-ACP-aptPD-L1. Red: Dil. Green: CellTracker Green CMFDA. Blue: DAPI. (c) Tumor sphere assay on the targeting ability of different samples at 12 h incubation. I: AC^{Cy5}P, II: Lip-AC^{Cy5}P, III: Lip-AC^{Cy5}P-aptPD-L1. (d-e) Time-dependent changes in ATP and MMP-2 abundance in vivo with Lip@AUR-ACP-aptPD-L1 + 4 Gy IR treatment. (f) Fusion status of different liposomes to B16F10 cell membranes in the co-culture system at 16, 18 or 30 h of incubation by CLSM. 4 Gy IR treatment was applied at 12 h. I: Lip@Dil+IR, II: Lip@Dil-ACP+IR, III: Lip@Dil-ACP-aptPD-L1+IR. Red: Dil. Green: CellTracker Green CMFDA. Blue: DAPI. (g) Time-dependent melanoma-targeted membrane fusion performance of Cy5-Lip@Dil-ACP-aptPD-L1 in vivo. 4 Gy IR treatment was applied after 12 h post intravenous injection. (h) Schedule of the combinational Lip@Dil-ACP-aptPD-L1 + 4 Gy IR treatment set-up in vitro. Statistical analysis in panels d-e was carried out via one-way ANOVA method. * indicates significance at $p < 0.05$, *** indicates significance at $p < 0.001$, **** indicates significance at $p < 0.0001$.

Figure 4. Evaluation on AND-gate release of eCpG from Lip@AUR-ACP-aptPD-L1 for DC stimulation. (a) NUPACK analysis on eCpG secondary structure. (b) Molecular docking analysis on the TLR9-binding behaviors of CpG ODN and eCpG. (c-d) Molecular docking analysis on the specific binding sites of CpG ODN or eCpG to TLR9. (e) Flow cytometry analysis on the target binding of different DNA sequences to DCs in co-culture system. I: Control, II: CpG ODN, III: eCpG, IV: mutated CpG

ODN, V: mutated eCpG, VI: closed eCpG. (f) Stimulatory impact of various DNA sequences to DC maturation by flow cytometry analysis in co-culture system. I: Control, II: CpG ODN, III: eCpG, IV: mutated CpG ODN, V: mutated eCpG, VI: closed eCpG. (g) CLSM analysis regarding the effect of PNA complexation on eCpG^{Cy5} release from membrane-bound aptamer assemblies under different conditions. I: 0 nM ATP+5 nM MMP-2, II: 100 nM ATP+5 nM MMP-2, III: 0 nM ATP+10 nM MMP-2, IV: 200 nM ATP+10 nM MMP-2. (h) Quantitative flow cytometry analysis regarding eCpG^{Cy5} release from membrane-bound aptamer assemblies under different conditions. I: 0 nM ATP+5 nM MMP-2, II: 100 nM ATP+5 nM MMP-2, III: 0 nM ATP+10 nM MMP-2, IV: 200 nM ATP+10 nM MMP-2. (i) CLSM analysis regarding the effect of PNA complexation on eCpG^{Cy5} release from membrane-bound aptamer assemblies at 16 h incubation in vivo. (j) Time-dependent analysis on eCpG^{Cy5} release from Lip@AUR-AC^{Cy5}P-aptPD-L1 in vivo. 4 Gy IR treatment was applied at the time point of 12 h. (k) Time-dependent evaluation on DC maturation status (CD11c+CD80+CD86+) in the co-culture system after Lip@AUR-ACP-aptPD-L1 + 4 Gy IR treatment. (l-m) Time-dependent evaluation on DC maturation status (CD11c+CD80+CD86+) in the co-culture system after Lip@AUR-ACP-aptPD-L1 + 4 Gy IR treatment in vivo. (n) Treatment schedule for the B16F10-mouse splenocyte co-incubation system for the evaluation of the immunostimulatory effects.

Figure 5. Immunostimulatory effect of combined Lip@AUR-ACP-aptPD-L1 and 4 Gy IR treatment in vitro. (a) Flow cytometry analysis on the maturation status (CD11c+CD80+CD86+) of DCs in the co-incubation system after different treatments.

(b) Flow cytometry analysis on T cell activation status (CD3+CD4+CD8+) in the co-incubation system after different treatments. (c-f) Secretion levels of immunostimulatory markers including IFN- γ , TNF- α , CXCL10 and IL-2 in the supernatants of the co-culture system after different treatments. (g) Flow cytometry analysis on the apoptosis of B16F10 cells after different treatments in co-culture system. (h) γ -H2AX immunofluorescence of IR-treated B16F10 cells after different sample treatments. I: PBS, II: Lip, III: Lip-aptPD-L1, IV: Lip-ACP-aptPD-L1, V: Lip@AUR-aptPD-L1, VI: Lip@AUR-ACP-aptPD-L1.

Figure 6. Anti-tumor effects of Lip@AUR-ACP-aptPD-L1-augmented radio-immunotherapy in vivo. (a) Schematic representation of the treatment protocol for B16F10-luc tumor-bearing mice. (b) In vivo bioluminescence images of B16F10-Luc tumor-bearing mice during treatment (n=5). (c) Tumor volume analysis throughout the treatment period (n=5). I: PBS, II: Lip, III: Lip-aptPD-L1, IV: Lip-ACP-aptPD-L1, V: Lip@AUR-aptPD-L1, VI: Lip@AUR-ACP-aptPD-L1, VII: PBS+IR, VIII: Lip+IR, IX: Lip-aptPD-L1+IR, X: Lip-ACP-aptPD-L1+IR, XI: Lip@AUR-aptPD-L1+IR, XII: Lip@AUR-ACP-aptPD-L1+IR. (d) Tumor weight analysis at the end of the treatment period (n=5). I: PBS, II: Lip, III: Lip-aptPD-L1, IV: Lip-ACP-aptPD-L1, V: Lip@AUR-aptPD-L1, VI: Lip@AUR-ACP-aptPD-L1. (e) Survival analysis of mice after different treatments (n=6). I: PBS, II: Lip, III: Lip-aptPD-L1, IV: Lip-ACP-aptPD-L1, V: Lip@AUR-aptPD-L1, VI: Lip@AUR-ACP-aptPD-L1, VII: PBS+IR, VIII: Lip+IR, IX: Lip-aptPD-L1+IR, X: Lip-ACP-aptPD-L1+IR, XI: Lip@AUR-aptPD-L1+IR, XII: Lip@AUR-ACP-aptPD-L1+IR. (f) Western blotting on the expression

levels of related proteins in the tumor tissues. (g) The release level of ATP and MMP-2 in B16F10 tumors after different treatment (n=3). I: PBS+IR, II: Lip+IR, III:Lip@AUR+IR, IV: Lip@AUR-aptPD-L1+IR. (h) TUNEL staining of tumor tissue samples after treatment. I: PBS, II: Lip, III: Lip-aptPD-L1, IV: Lip-ACP-aptPD-L1, V: Lip@AUR-aptPD-L1, VI: Lip@AUR-ACP-aptPD-L1.

Figure 7. Mechanism underlying Lip@AUR-ACP-aptPD-L1-augmented radio-immunotherapy in vivo. (a-c) Flow cytometry analysis on the infiltration of total immune cells (CD45+), DCs (CD11c+CD80+CD86+) and effector T cells (CD3+CD4+CD8+) at the tumor site after treatment with I: PBS, II: Lip, III: Lip-aptPD-L1, IV: Lip-ACP-aptPD-L1, V: Lip@AUR-aptPD-L1, VI: Lip@AUR-ACP-aptPD-L1. (d) Immunofluorescence images of the extracted tumors showing infiltration of CD8+T cells after treatment with I: PBS, II: Lip, III: Lip-aptPD-L1, IV: Lip-ACP-aptPD-L1, V: Lip@AUR-aptPD-L1, VI: Lip@AUR-ACP-aptPD-L1. (e-h) The secretion levels of IFN- γ , TNF- α , CXCL10 and IL-2 in mouse serum after treatment with I: PBS, II: Lip, III: Lip-aptPD-L1, IV: Lip-ACP-aptPD-L1, V: Lip@AUR-aptPD-L1, VI: Lip@AUR-ACP-aptPD-L1.

Figure 8. Systemic anti-tumor immunity of Lip@AUR-ACP-aptPD-L1 to suppress distal B16F10 tumors. (a) Schematic diagram of the treatment schedule for bilateral B16F10 tumor model. (b) Statistical analysis of distal B16F10 tumor volume during treatment (n=3). I: PBS, II: Lip, III: Lip-aptPD-L1, IV: Lip-ACP-aptPD-L1, V: Lip@AUR-aptPD-L1, VI: Lip@AUR-ACP-aptPD-L1, VII: PBS+IR, VIII: Lip+IR, IX: Lip@AUR-aptPD-L1+IR, X: Lip-ACP-aptPD-L1+IR, XI: Lip@AUR-aptPD-L1+IR, XII:

Lip@AUR-ACP-aptPD-L1+IR. (c) Survival analysis of bilateral B16F10 tumor model-bearing mice (n=6). I: PBS, II: Lip, III: Lip-aptPD-L1, IV: Lip-ACP-aptPD-L1, V: Lip@AUR-aptPD-L1, VI: Lip@AUR-ACP-aptPD-L1, VII: PBS+IR, VIII: Lip+IR, IX: Lip-aptPD-L1+IR, X: Lip-ACP-aptPD-L1+IR, XI: Lip@AUR-aptPD-L1+IR, XII: Lip@AUR-ACP-aptPD-L1+IR. (d-f) Flow cytometry analysis on the infiltration levels of DCs (CD11c+CD80+CD86+), effector T cells (CD3+CD4+CD8+) and memory CD8+T cells (CD8+CD44+CD62L-) within the distal tumors after treatment with I: PBS, II: Lip, III: Lip-aptPD-L1, IV: Lip-ACP-aptPD-L1, V: Lip@AUR-aptPD-L1, VI: Lip@AUR-ACP-aptPD-L1.”

6. The manuscript should be double checked thoroughly, as there are lots of writing errors and typos, such as Line 486 “immunofluorescence”, the format of number and unit such as 20nm.

A: Thank you for the correction. Based on your advice we have thoroughly proofread the whole manuscript to eliminate those linguistic and formatting errors. For instance, the first letter for the word “immunofluorescence” in Line 486 of the original manuscript has been capitalized, while space was added between the numeric values and the units throughout the manuscript. Some of the examples are shown here below for your review:

“Immunofluorescence imaging of PD-L1 with fluorescently-labeled antibodies revealed a general positive correlation between IR dose and PD-L1 expression, which was in accordance with the observations in previous reports and confirmed the promotional effect of IR on PD-L1 expression (Supplementary Figure S9).”

“Sole IR treatment induced modest inhibition on melanoma growth with a final tumor volume of around 1550 mm³, which was slightly lower than the control group and suggested the innate radiotherapeutic resistance of melanomas (Figure 6b-c).”

“Mouse-derived melanoma cell line B16F10 was cultured in RPMI 1640 medium containing 10% fetal bovine serum (Gibco), penicillin (100 µg·mL⁻¹), and streptomycin (100 µg·mL⁻¹). Mouse embryonic fibroblasts NIH3T3 and B16F10-luc cell lines were cultured in high-glucose DMEM medium containing 10% fetal bovine serum (Gibco), penicillin (100 µg·mL⁻¹), and streptomycin (100 µg·mL⁻¹). The cells were cultured in a 37°C constant temperature incubator containing 5% carbon dioxide.”

Reviewer #4 (Remarks to the Author):

The authors presented a paper about "Cascade-amplification of melanoma-targeted radioimmunotherapy via fusogenic liposomes functionalized with multivariate-gated aptamer assemblies".

The topic is absolutely interesting and it totally deserves attention both a clinical and research point of view.

The background provided by the authors is that immunosuppressive microenvironment of solid tumors may limit the efficacy of this combination and therefore they propose a strategy to inhibit tumor-intrinsic signaling pathway to suppress immunosuppressive cells.

The idea itself sounds definitively interesting and the description of the multivariate-

gated aptamer assembly-modified AUR-loaded fusogenic liposome as an adjuvant for melanoma-targeted radioimmunotherapy is clear and detailed.

Several molecular insights of the process are in fact provided and make the article scientifically sound.

However, I believe that since the ultimate goal of the authors was to provide some useful proposals to be evaluated in the clinical practice it would be an addition to introduce within the manuscript some up-to-date evidence about the use of combination therapy between immunotherapy and radiotherapy (see PMID: 33847208 for a comprehensive dissertation); in fact by adding more data coming from already published clinical studies would broaden the interest of the article to a wider audience. For example in the clinical practice radiotherapy can be delivered concomitantly to immunotherapy for melanoma either at the beginning of a treatment, Peri-Induction Radiotherapy (PIR) or after there has been an immunological escape with a clinical progression of the disease, Post-Escape Radiotherapy (PER). Also the role of doses and fractionations should be addressed more in detail.

A: Thank you for the appreciation of our work. We are grateful for your insightful comments and carry out substantial revision as requested, which we believe have significantly improved the quality of this study. Specifically, regarding the issue related to the reference support for the application potential of radio-immunotherapy, we have added multiple recent representative publications in this area and expanded the related discussions in the introduction section, of which the corresponding reference numbers are [7], [8], [9]. The added discussions are also shown here below:

“Indeed, clinical insights collectively confirm that combining radiotherapy with immunotherapy could convey significant improvement on the overall survival benefit of melanoma patients without inducing obvious side effects. For instance, preconditioning tumors with IR (peri-induction radiotherapy) could activate the immune system and facilitate the recognition and elimination of tumor cells by the sequentially administered immunotherapeutic modalities. On the other hand, there are reports that the local IR treatment of tumors following immunotherapy (post-escape radiotherapy) could potentially remodel the TME to reverse immunoresistance and prevent tumor immune escape^{7,8,9}.”

Meanwhile, we would like to mention that the parameters for IR treatment have been investigated in a series of preliminary experiments to determine the optimal IR treatment conditions but were not included in the original version of this manuscript. Based on your advice, we have carried out additional analysis to investigate the impact of IR dose and fractionation on the post-IR immunocomposition and the eventual antimelanoma efficacy, which may improve the informativity of this study from a clinical perspective. For this purpose, we have established B16F10 tumor-bearing C57BL/6J mouse models, which were subjected to the treatment with PBS and Lip@AUR-ACP-aptPD-L1 as well as different IR treatment schedules. The treatment period lasted 15 days and the eventual therapeutic impact was comprehensively characterized, including the tumor size, survival curve, tumor cell death, immune cell infiltration status, etc. To start with, we first monitored the impact of sole IR treatment on melanoma progression in mice, where the melanoma sites were treated with 0 Gy, 2

Gy, 4 Gy and 8 Gy IR with a 5-day interval through the 15-day treatment period (3 times in total). As shown in Supplementary Figure 27, sole IR treatment did not induce obvious melanoma inhibitory effect in vivo, for which the relative tumor size reduction rates were all below 7%. Contrastingly, the coordinated treatment of Lip@AUR-ACP-aptPD-L1 and IR significantly improved the melanoma inhibition efficacy, where the average tumor weight in the Lip@AUR-ACP-aptPD-L1 + 4 Gy and Lip@AUR-ACP-aptPD-L1 + 8 Gy groups was only 0.25 g and 0.22 g with no statistical difference in between. The observations regarding the tumor progression status were consistent with the TUNEL staining results, which revealed that the Lip@AUR-ACP-aptPD-L1 + 4 Gy and Lip@AUR-ACP-aptPD-L1 + 8 Gy groups had the largest dead melanoma cell populations (Supplementary Figure 27b-e). The tumors were further extracted for flow cytometric analysis to investigate the immunocomposition therein. As shown in Supplementary Figure 27f-i, the frequencies of DCs, CD4⁺ T cells, CD8⁺ T cells, Tregs and MSDCs in the PBS + IR groups all showed no significant changes, indicating that there was no obvious alleviation of the immunosuppressive TME. However, the Lip@AUR-ACP-aptPD-L1 + 4 Gy and Lip@AUR-ACP-aptPD-L1 + 8 Gy treatments both induced significant expansion of the DC, CD4⁺ T cell and CD8⁺ T cell populations, while the tumor-residing Treg and MSDC populations have markedly decreased, suggesting the successful remodeling of the TME into an immunoactivated state. These data also confirmed the superior suitability of the Lip@AUR-ACP-aptPD-L1 + 4 Gy for melanoma treatment due to its potent radio-immunotherapeutic efficacy at a relatively lower IR dose.

On the other hand, we also investigated the impact of IR fractionation on the melanoma inhibition and immunostimulation effects. Herein, the B16F10 tumor-bearing mice were treated with 4 Gy IR for 0 time, 1 time (at the start of the treatment period), 3 times (with a 4-day interval) and 5 times (with a 2-day interval) through the 15-day treatment period. Notably, treating the melanoma-bearing mice with IR for 0 time and 1 time induced no obvious alleviation of the tumor burden even with Lip@AUR-ACP-aptPD-L1 pretreatment. In comparison, the combinational treatment of Lip@AUR-ACP-aptPD-L1 + 4 Gy × 3 and Lip@AUR-ACP-aptPD-L1 + 4 Gy × 5 induced potent melanoma inhibition effect, evidenced by the drastically reduced tumor weight (0.26 g and 0.24 g) and pronounced melanoma cell death under TUNEL staining. Flow cytometric analysis on the extracted tumors revealed that the TME remodeling effect of the combinational Lip@AUR-ACP-aptPD-L1 + IR treatment showed a general positive correlation with the times of IR exposure, where increasing the rounds of IR treatment at 4 Gy could elevate the frequency of tumor-residing DCs and CD4⁺/CD8⁺ T cells while reducing immunosuppressive Treg and MSDC populations. Nevertheless, it is notable that there is no significant difference between the antimelanoma and immunostimulatory effects of Lip@AUR-ACP-aptPD-L1 + 4 Gy × 3 and Lip@AUR-ACP-aptPD-L1 + 4 Gy × 5 groups. These observations again validated the feasibility of Lip@AUR-ACP-aptPD-L1 + 4 Gy × 3 treatment as the standard condition for melanoma treatment in vivo, which may elicit optimal antimelanoma efficacy with reduced IR exposure. Overall, these characterizations justified the current IR treatment set-up for the liposome-augmented radio-immunotherapy, presenting a favorable

antimelanoma option with balanced antitumor efficacy and safety.

The related data and discussions are shown here below for your review:

“Investigations on IR treatment parameters on the radio-immunotherapeutic efficacy in vivo

It is well-established in clinical practice that IR doses and fractionations have significant impact on their tumoricidal and immunostimulatory effects. Therefore, we comprehensively studied the effect of these IR-related parameters on the eventual radio-immunotherapeutic efficacy. Herein, the B16F10 tumor-bearing C57BL/6J mice were subjected to the treatment of PBS and Lip@AUR-ACP-aptPD-L1 as well as different IR treatment schedules for 15 days. For the evaluation of the dose-dependent impact of IR treatment on the radio-therapeutic efficacy, the melanoma sites were treated with 0 Gy, 2 Gy, 4 Gy and 8 Gy IR with a 5-day interval through the 15-day treatment period (3 times in total) (Supplementary Figure 27a). Notably, sole IR treatment under all dose conditions did not induce obvious tumor inhibition effect, while the coordinated treatment of Lip@AUR-ACP-aptPD-L1 and IR significantly improved the melanoma inhibition efficacy, where the average tumor weight in the Lip@AUR-ACP-aptPD-L1 + 4 Gy and Lip@AUR-ACP-aptPD-L1 + 8 Gy groups was only 0.25 g and 0.22 g with no statistical difference in between. These observations were consistent with the TUNEL staining results, which revealed that the Lip@AUR-ACP-aptPD-L1 + 4 Gy and Lip@AUR-ACP-aptPD-L1 + 8 Gy groups had the largest dead melanoma cell populations (Supplementary Figure 27b-e). Flow cytometric analysis on the immunocomposition in the extracted tumors showed that the frequencies of DCs, CD4+

T cells, CD8⁺ T cells, Tregs and MSDCs in the PBS + IR groups all showed no significant changes, indicating that there was no obvious alleviation of the immunosuppressive TME. However, the Lip@AUR-ACP-aptPD-L1 + 4 Gy and Lip@AUR-ACP-aptPD-L1 + 8 Gy treatments both induced significant expansion of the DC, CD4⁺ T cell and CD8⁺ T cell populations, while the tumor-residing Treg and MSDC populations have markedly decreased, suggesting the successful remodeling of the TME into an immunoactivated state (Supplementary Figure 27f-i). These data also confirmed the superior suitability of the Lip@AUR-ACP-aptPD-L1 + 4 Gy for melanoma treatment due to its potent radio-immunotherapeutic efficacy at a relatively lower IR dose.

The impact of IR fractionation on the melanoma inhibition and immunostimulation effects was also studied *in vivo*, where the B16F10 tumor-bearing mice were treated with 4 Gy IR for 0 time, 1 time (at the start of the treatment period), 3 times (with a 4-day interval) and 5 times (with a 2-day interval) through the 15-day treatment period. Notably, treating the melanoma-bearing mice with IR for 0 time and 1 time induced no obvious alleviation of the tumor burden even with Lip@AUR-ACP-aptPD-L1 pretreatment. In comparison, the combinational treatment of Lip@AUR-ACP-aptPD-L1 + 4 Gy × 3 and Lip@AUR-ACP-aptPD-L1 + 4 Gy × 5 drastically inhibited tumor growth with a low average final tumor weight of 0.26 g and 0.24 g and pronounced melanoma cell death under TUNEL staining. Flow cytometric analysis on the extracted tumors revealed that the TME remodeling effect of the combinational Lip@AUR-ACP-aptPD-L1 + IR treatment showed a general positive correlation with the times of IR

exposure, where increasing the rounds of IR treatment at 4 Gy could elevate the frequency of tumor-residing DCs and CD4⁺/CD8⁺ T cells while reducing immunosuppressive Treg and MSDC populations. Nevertheless, it is notable that there is no significant difference between the antimelanoma and immunostimulatory effects of Lip@AUR-ACP-aptPD-L1 + 4 Gy × 3 and Lip@AUR-ACP-aptPD-L1 + 4 Gy × 5 groups. These observations again validated the feasibility of Lip@AUR-ACP-aptPD-L1 + 4 Gy × 3 treatment as the standard condition for melanoma treatment *in vivo*, which may elicit optimal antimelanoma efficacy with reduced IR exposure.”

“Evaluation of IR doses on the therapeutic effect *in vivo*”

B16F10-luc tumor cells (1×10^6 cells) were injected subcutaneously into 6-week-old mice to establish B16F10-luc tumor C57BL/6J mouse model. The mice were cultured continuously until the tumor size reached 100 mm³. They were randomly divided into 8 groups with 3 mice in each group, which were intravenously injected with 100 μL of Lip@AUR-ACP-aptPD-L1 (2 mg·kg⁻¹), while the same volume of fresh PBS was administered as the control group. At 12 h post intravenous injection, the IR groups were treated with 0 Gy, 2 Gy, 4 Gy or 8 Gy IR. IR treatment was performed three times with a 4-day interval for a total of 15 days. Bioluminescence imaging was performed every 5 days, and 20 μL (7.5 mg·mL⁻¹) luciferase was injected into the intraperitoneal cavity of mice. After anesthesia with isoflurane, tumor volume of each group was detected by IVIS imaging system. A parallel set of mouse models were established, and the survival of mice in each group was observed until the 50th day after the 15-day treatment (n=3). At the end of treatment, the tumors in each group were dissected and

cleaned with PBS, and further cut into thin sections for TUNEL staining using related assay kits and observed by CLSM.

The tumor was pulverized and treated with red cell lysate for 15 min, followed by the incubation with 1 μ L PC7-anti-CD45 antibodies, 1 μ L APC-anti-CD11c antibodies, 1 μ L PE-anti-MHC-II antibodies, 1 μ L APC-anti-CD3 antibodies, 1 μ L FITC-anti-CD4 antibodies, 1 μ L PE-anti-CD8a antibodies, 1 μ L PE-anti-CD4 antibodies, 1 μ L APC-anti-CD25 antibodies, 1 μ L FITC-anti-CTLA-4 antibodies, 1 μ L FITC-anti-CD11b antibodies or 1 μ L PE-anti-GR1 antibodies. Subsequently, the infiltration of immune cells was detected by flow cytometry.”

“Evaluation of IR fractionations on the therapeutic effects *in vivo*

B16F10-luc tumor cells (1×10^6 cells) were injected subcutaneously into 6-week-old mice to establish B16F10-luc tumor C57BL/6J mouse model. The mice were cultured continuously until the tumor size reached 100 mm^3 . They were randomly divided into 8 groups with 3 animals in each group, which were subjected to the intravenous injection of 100 μ L Lip@AUR-ACP-aptPD-L1 ($2 \text{ mg} \cdot \text{kg}^{-1}$), while the same volume of fresh PBS was administered as the control group. At 12 h post intravenous injection, mice in the IR groups were treated with 4 Gy IR for 0 time, 1 time (at the start of the treatment period), 3 times (with a 4-day interval) and 5 times (with a 2-day interval) through the 15-day treatment period. Bioluminescence imaging was performed every 5 days, and 20 μ L ($7.5 \text{ mg} \cdot \text{mL}^{-1}$) luciferase was injected into the intraperitoneal cavity of mice. After anesthesia with isoflurane, tumor volume of each group was detected by IVIS imaging system. A parallel set of animal models were established, and the survival

of mice in each group was observed until the 50th day after the 15-day treatment (n=3). At the end of treatment, the tumors in each group were dissected and cleaned with PBS, and further cut into thin sections for TUNEL staining using related assay kits and observed by CLSM.

The tumor was pulverized and treated with red cell lysate for 15 min, followed by the incubation with 1 μ L PC7-anti-CD45 antibodies, 1 μ L APC-anti-CD11c antibodies, 1 μ L PE-anti-MHC-II antibodies, 1 μ L APC-anti-CD3 antibodies, 1 μ L FITC-anti-CD4 antibodies, 1 μ L PE-anti-CD8a antibodies, 1 μ L PE-anti-CD4 antibodies, 1 μ L APC-anti-CD25 antibodies, 1 μ L FITC-anti-CTLA-4 antibodies, 1 μ L FITC-anti-CD11b antibodies or 1 μ L PE-anti-GR1 antibodies, afterwards the infiltration of immune cells was detected by flow cytometry.”

Supplementary Figure 27. Evaluation on the effects of IR doses on the therapeutic potency in vivo. (a) Schematic representation of the treatment protocol for B16F10-luc tumor-bearing mice with graded IR doses. (b) In vivo bioluminescence images of B16F10-luc tumor-bearing mice with different IR doses (n=3). (c) Final tumor weight analysis after the 15-day treatment (n=3). (d) Survival curve of B16F10 tumor-bearing

mice after different treatments (n=3). I: PBS+0 Gy, II: PBS+2 Gy, III: PBS+4 Gy, IV: PBS+8 Gy, V: Lip@AUR-ACP-aptPD-L1+0 Gy, VI: Lip@AUR-ACP-aptPD-L1+2 Gy, VII: Lip@AUR-ACP-aptPD-L1+4 Gy, VIII: Lip@AUR-ACP-aptPD-L1+8 Gy. (e) TUNEL staining of B16F10 tumor tissue samples after different treatments. (f-i) Flow cytometry analysis on the infiltration of DCs (CD11c+MHC-II+), CD4+/CD8+ T cells (CD3+CD4+ and CD3+CD8+), Tregs (CD4+CD25+CTLA-4+) and MDSCs (CD11b+GR1+) in the B16F10 tumors after different treatments.

Supplementary Figure 28. Evaluation on the effect of IR fractionations on the therapeutic potency in vivo. (a) Schematic representation of the treatment protocol for B16F10-luc tumor-bearing mice with 4 Gy IR under different fractionation set-ups. (b) In vivo bioluminescence images of B16F10-luc tumor-bearing mice under different

fractionation set-ups (n=3). (c) Final tumor weight at the end of the incubation period (n=3). (d) Survival curve of B16F10 tumor-bearing mice after different treatments (n=3). I: PBS-IR, II: PBS+Once IR, III: PBS+3 times IR, IV: PBS+5 times IR, V: Lip@AUR-ACP-aptPD-L1-IR, VI: Lip@AUR-ACP-aptPD-L1+Once IR, VII: Lip@AUR-ACP-aptPD-L1+3 times IR, VIII: Lip@AUR-ACP-aptPD-L1+5 times IR. (e) TUNEL staining of B16F10 tumor tissue samples after different treatments. (f-i) Flow cytometry analysis on the infiltration of DCs (CD11c+MHC-II+), CD4+CD8+T cells (CD3+CD4+CD8+), Tregs (CD4+CD25+CTLA-4+) and MDSCs (CD11b+GR1+) in the B16F10 tumor tissues after different treatments.

Reviewer #5 (Remarks to the Author):

The study conducted by Xue and colleagues introduces a promising and innovative approach to combat melanoma, integrating targeted liposomal delivery, radiation therapy, and immune system activation. This multifaceted strategy holds significant potential for enhancing treatment effectiveness while minimizing patient risks. The research demonstrates meticulous planning and skillful execution of experiments, yielding compelling results. Nevertheless, to bolster the manuscript's comprehensibility and scientific rigor, it is imperative to furnish more comprehensive details, particularly within the methodology sections covering both in vitro and in vivo experiments. A more expansive methodological exposition would empower readers with a clearer grasp of the experimental protocols, facilitating replication and critical evaluation of the study's validity. Furthermore, improving the informativeness of figure legends is essential, as

it would provide succinct yet vital explanations of the figures' content and their implications. Especially in the flow cytometry figures, as many of them are lacking either the marker or fluorochrome used. These refinements are crucial for enhancing the overall quality and impact of this research.

A: Thank you for the careful review of manuscript and the insightful suggestion. Based on the information you kindly provided, we have made thorough corrections to the descriptions regarding the experimental methods in the whole manuscript. Details were also added into the revised manuscript at appropriate places including gating strategy and the choices of markers and fluorochromes. We believe that these revisions and additions have substantially improved the informativity and comprehensibility of the present study and hope that they may address your concerns.

Figure 2a. Please provide the marker for each fluorochrome. It is not clear the gating strategy used in the figure.

A: Thank you for the advice. Based on your suggestions we have listed the marker for individual fluorochromes in Figure 3a (Figure 2a in the original manuscript) as requested. Meanwhile, the gating strategy for this panel was provided in Supplementary Figure 42 and Supplementary Figure 43.

The related Figures are shown here below for your review:

Figure 3. Melanoma targeting and membrane fusion performance of Lip@AUR-ACP-aptPD-L1 in vitro based on B16F10/splenocyte co-incubation system. (a) Targeting ability of eCpG and aptPD-L1 with the designated cell targets in the B16F10/splenocyte co-incubation system by flow cytometry. **(b)** Fusion status of

different liposomes to B16F10 cell membranes in the co-culture system at 3, 6, 12 or 18 h of incubation by CLSM. I: Lip@Dil, II: Lip@Dil-ACP, III: Lip@Dil-ACP-aptPD-L1. Red: Dil. Green: CellTracker Green CMFDA. Blue: DAPI. (c) Tumor sphere assay on the targeting ability of different samples at 12 h incubation. I: AC^{Cy5}P, II: Lip-AC^{Cy5}P, III: Lip-AC^{Cy5}P-aptPD-L1. (d-e) Time-dependent changes in ATP and MMP-2 abundance in vivo with Lip@AUR-ACP-aptPD-L1 + 4 Gy IR treatment. (f) Fusion status of different liposomes to B16F10 cell membranes in the co-culture system at 16, 18 or 30 h of incubation by CLSM. 4 Gy IR treatment was applied at 12 h. I: Lip@Dil+IR, II: Lip@Dil-ACP+IR, III: Lip@Dil-ACP-aptPD-L1+IR. Red: Dil. Green: CellTracker Green CMFDA. Blue: DAPI. (g) Time-dependent melanoma-targeted membrane fusion performance of Cy5-Lip@Dil-ACP-aptPD-L1 in vivo. 4 Gy IR treatment was applied after 12 h post intravenous injection. (h) Schedule of the combinational Lip@Dil-ACP-aptPD-L1 + 4 Gy IR treatment set-up in vitro. Statistical analysis in panels d-e was carried out via one-way ANOVA method. * indicates significance at $p < 0.05$, *** indicates significance at $p < 0.001$, **** indicates significance at $p < 0.0001$.

Supplementary Figure 42. Gating strategy for the FACS tests. a was applied for Supplementary Figure 9b; b was applied for Figure 4e; c was applied for Figure 4j; d was applied for Supplementary Figure 18b, Supplementary Figure 27i, Supplementary Figure 28i and Supplementary Figure 34b; e was applied for Supplementary Figure 23b and Supplementary Figure 35b; f was applied for Figure 5g.

Supplementary Figure 43. Gating strategy for the FACS tests. a was applied for Figure 7a; b was applied for Supplementary Figure 18a, Supplementary Figure 27h, Supplementary Figure 28h and Supplementary Figure 34a; c was applied for Figure 5b, Figure 7c, Figure 8e, Supplementary Figure 19a, Supplementary Figure 27g, Supplementary Figure 28g and Supplementary Figure 41b; d was applied for Figure 4f, Figure 4k, Figure 4l, Figure 5a, Figure 7b, Figure 8d, Supplementary Figure 19b and Supplementary Figure 41a; e was applied for Figure 4m; f was applied for Figure 8f; g was applied for Supplementary Figure 23a, Supplementary Figure 27f, Supplementary Figure 28f and Supplementary Figure 35a.

Figure 2b. Please add information about what the green and red colors represent. It is hard to understand the findings in the figure.

A: Thank you for the suggestion. The definition and description for the green and red fluorescence were added into the figure legend as requested.

The related data and discussions are shown here below for your review:

Figure 3. Melanoma targeting and membrane fusion performance of Lip@AUR-ACP-aptPD-L1 in vitro based on B16F10/splenocyte co-incubation system. (a) Targeting ability of eCpG and aptPD-L1 with the designated cell targets in the B16F10/splenocyte co-incubation system by flow cytometry. **(b)** Fusion status of

different liposomes to B16F10 cell membranes in the co-culture system at 3, 6, 12 or 18 h of incubation by CLSM. I: Lip@Dil, II: Lip@Dil-ACP, III: Lip@Dil-ACP-aptPD-L1. Red: Dil. Green: CellTracker Green CMFDA. Blue: DAPI. (c) Tumor sphere assay on the targeting ability of different samples at 12 h incubation. I: AC^{Cy5}P, II: Lip-AC^{Cy5}P, III: Lip-AC^{Cy5}P-aptPD-L1. (d-e) Time-dependent changes in ATP and MMP-2 abundance in vivo with Lip@AUR-ACP-aptPD-L1 + 4 Gy IR treatment. (f) Fusion status of different liposomes to B16F10 cell membranes in the co-culture system at 16, 18 or 30 h of incubation by CLSM. 4 Gy IR treatment was applied at 12 h. I: Lip@Dil+IR, II: Lip@Dil-ACP+IR, III: Lip@Dil-ACP-aptPD-L1+IR. Red: Dil. Green: CellTracker Green CMFDA. Blue: DAPI. (g) Time-dependent melanoma-targeted membrane fusion performance of Cy5-Lip@Dil-ACP-aptPD-L1 in vivo. 4 Gy IR treatment was applied after 12 h post intravenous injection. (h) Schedule of the combinational Lip@Dil-ACP-aptPD-L1 + 4 Gy IR treatment set-up in vitro. Statistical analysis in panels d-e was carried out via one-way ANOVA method. * indicates significance at $p < 0.05$, *** indicates significance at $p < 0.001$, **** indicates significance at $p < 0.0001$.

How does the combined treatment of Lip@AUR-aptPD-L1 liposomes and 4Gy IR affect immune cells in extended data Figure 8?

A: Thank you for the reminder. We would like to apologize for the lack of elaboration for the data in Supplementary Figure 8 and expanded the descriptions as suggested. As shown in Supplementary Figure 11b, under the combined treatment of Lip@AUR-

aptPD-L1 and 4 Gy IR, the immune cells were not obviously inhibited, for which the relative decrease in immune cell viability was less than 10%. Notably, when increasing the IR dose to 8 Gy, the viability of immune cell populations markedly decreased by around 22%. These data justified the choice of 4 Gy IR as the standard IR treatment condition in the present study, presenting a balanced treatment option with optimal immunostimulatory potential and safety.

Related discussion:

“It is also observed that the Lip@AUR-aptPD-L1 + 4 Gy IR treatment induced negligible negative impact on the immune cell populations with a relative viability decrease of less than 10% (Supplementary Figure 11b), while the combined treatment of 8 Gy IR and Lip@AUR-aptPD-L1 liposomes caused a 21% reduction in splenocyte survival and the changes were statistically significant, immediately suggesting the importance to lower the necessary IR doses for maximizing the post-IR immunostimulatory effect with optimal safety.”

Supplementary Figure 11. Evaluation on the radiosensitizing effect of the liposomes. Impact of the combined treatment of Lip@AUR-aptPD-L1 and IR of different doses on B16F10 cells (a) and splenocytes (b).

Extended Figure 19. What is the concentration about?

A: Thank you for the reminder. We would like to state that Supplementary Figure 19 showed the inhibition effect of Lip, Lip-aptPD-L1, Lip-ACP-aptPD-L1, Lip@AUR-aptPD-L1 and Lip@AUR-ACP-aptPD-L1 with 4 Gy IR on B16F10 cells in the B16F10/splenocyte co-incubation system, while the axis label “concentration” represents the dosage of the liposomal samples. The axis labels were corrected according to your suggestions and the revised figure is shown here below for your review:

Supplementary Figure 25. MTT assay on the radio-immunotherapeutic effect of the liposomes in vitro. B16F10 inhibition effect of different groups (a) without or (b) with 4 Gy IR under the co-culture condition by MTT assay. I: Lip, II: Lip-aptPD-L1, III: Lip-ACP-aptPD-L1, IV: Lip@AUR-aptPD-L1, V: Lip@AUR-ACP-aptPD-L1.

The authors state that they found CD62L+CD44+ cells in the melanoma tissue samples. The memory T cells should be mentioned as CD62L-(low) and CD44+, figure 8f shows this cell population correctly but the text and figure legend are wrong.

A: Thank you for the reminder. Based on your advice we have corrected the text and figure legends as requested. The revised contents are shown here below for your review:

The related instructions are shown here below for your review:

“In addition, we have detected a significant expansion of CD62L-CD44+ memory CD8+T cells in the melanoma tissue samples according to flow cytometric analysis (Figure 8f).”

Figure 8. Systemic anti-tumor immunity of Lip@AUR-ACP-aptPD-L1 to suppress distal B16F10 tumors. (a) Schematic diagram of the treatment schedule for bilateral

B16F10 tumor model. (b) Statistical analysis of distal B16F10 tumor volume during treatment (n=3). I: PBS, II: Lip, III: Lip-aptPD-L1, IV: Lip-ACP-aptPD-L1, V: Lip@AUR-aptPD-L1, VI: Lip@AUR-ACP-aptPD-L1, VII: PBS+IR, VIII: Lip+IR, IX: Lip-aptPD-L1+IR, X: Lip-ACP-aptPD-L1+IR, XI: Lip@AUR-aptPD-L1+IR, XII: Lip@AUR-ACP-aptPD-L1+IR. (c) Survival analysis of bilateral B16F10 tumor model-bearing mice (n=6). I: PBS, II: Lip, III: Lip-aptPD-L1, IV: Lip-ACP-aptPD-L1, V: Lip@AUR-aptPD-L1, VI: Lip@AUR-ACP-aptPD-L1, VII: PBS+IR, VIII: Lip+IR, IX: Lip-aptPD-L1+IR, X: Lip-ACP-aptPD-L1+IR, XI: Lip@AUR-aptPD-L1+IR, XII: Lip@AUR-ACP-aptPD-L1+IR. (d-f) Flow cytometry analysis on the infiltration levels of DCs (CD11c+CD80+CD86+), effector T cells (CD3+CD4+CD8+) and memory CD8+T cells (CD8+CD44+CD62L-) within the distal tumors after treatment with I: PBS, II: Lip, III: Lip-aptPD-L1, IV: Lip-ACP-aptPD-L1, V: Lip@AUR-aptPD-L1, VI: Lip@AUR-ACP-aptPD-L1.

References

1. Andrews LP, Yano H, Vignali DAA. Inhibitory receptors and ligands beyond PD-1, PD-L1 and CTLA-4: breakthroughs or backups. *Nat Immunol* **20**, 1425-1434 (2019).
2. Deng L, *et al.* Irradiation and anti-PD-L1 treatment synergistically promote antitumor immunity in mice. *J Clin Invest* **124**, 687-695 (2014).
3. Gao Y, *et al.* Ataxia telangiectasia mutated kinase inhibition promotes irradiation-induced PD-L1 expression in tumour-associated macrophages through IFN-I/JAK signalling pathway. *Immunology* **168**, 346-361 (2023).
4. Kim B, Sun S, Varner JA, Howell SB, Ruoslahti E, Sailor MJ. Securing the Payload, Finding the Cell, and Avoiding the Endosome: Peptide-Targeted, Fusogenic Porous Silicon Nanoparticles for Delivery of siRNA. *Adv Mater* **31**,

e1902952 (2019).

5. Kim B, Pang HB, Kang J, Park JH, Ruoslahti E, Sailor MJ. Immunogene therapy with fusogenic nanoparticles modulates macrophage response to *Staphylococcus aureus*. *Nat Commun* **9**, 1969 (2018).
6. Zheng C, *et al.* Membrane-Fusion-Mediated Multiplex Engineering of Tumor Cell Surface Glycans for Enhanced NK Cell Therapy. *Adv Mater* **35**, e2206989 (2023).
7. Mo R, Jiang T, Gu Z. Enhanced anticancer efficacy by ATP-mediated liposomal drug delivery. *Angew Chem Int Ed Engl* **53**, 5815-5820 (2014).
8. Xiang Z, Zhao J, Qu J, Song J, Li L. A Multivariate-Gated DNA Nanodevice for Spatioselective Imaging of Pro-metastatic Targets in Extracellular Microenvironment. *Angew Chem Int Ed Engl* **61**, e202111836 (2022).
9. Widen JC, *et al.* AND-gate contrast agents for enhanced fluorescence-guided surgery. *Nat Biomed Eng* **5**, 264-277 (2021).
10. Xiang Z, Zhao J, Yi D, Di Z, Li L. Peptide Nucleic Acid (PNA)-Guided Peptide Engineering of an Aptamer Sensor for Protease-Triggered Molecular Imaging. *Angew Chem Int Ed Engl* **60**, 22659-22663 (2021).
11. Xie S, *et al.* Engineering Aptamers with Selectively Enhanced Biostability in the Tumor Microenvironment. *Angew Chem Int Ed Engl* **61**, e202201220 (2022).
12. Di Virgilio F, Sarti AC, Falzoni S, De Marchi E, Adinolfi E. Extracellular ATP and P2 purinergic signalling in the tumour microenvironment. *Nat Rev Cancer* **18**, 601-618 (2018).
13. Ding B, *et al.* ZIF-8 Nanoparticles Evoke Pyroptosis for High-Efficiency Cancer Immunotherapy. *Angew Chem Int Ed Engl* **62**, e202215307 (2023).
14. Gao W, *et al.* Alloying of Cu with Ru Enabling the Relay Catalysis for Reduction of Nitrate to Ammonia. *Adv Mater* **35**, e2202952 (2023).
15. Li K, *et al.* Oxygen Self-Generating Nanoreactor Mediated Ferroptosis Activation and Immunotherapy in Triple-Negative Breast Cancer. *ACS Nano* **17**, 4667-4687 (2023).
16. Cheng J, *et al.* Formulation of functionalized PLGA-PEG nanoparticles for in vivo targeted drug delivery. *Biomaterials* **28**, 869-876 (2007).

REVIEWERS' COMMENTS

Reviewer #1 (Remarks to the Author):

The revision is ready for publication.

Reviewer #2 (Remarks to the Author):

This manuscript has been significantly improved after this round of revision and I also have satisfied with authors' responses on my concerns. Thus, I would like to recommend it for publication in Nature Communications without further revision.

Reviewer #3 (Remarks to the Author):

The authors have added large amount of supplementary data to fully adress my concerns. I have not any other comments on the manuscript.

Reviewer #4 (Remarks to the Author):

The authors have satisfactorily addressed all my comments.

Reviewer #5 (Remarks to the Author):

The study presented by Ren X. et. al, investigates a novel approach for melanoma treatment using melanoma-targeted fusogenic liposomal nanoformulations integrated with Auranofin (AUR) and multivariate-gated aptamer assemblies. The complex nanoformulation aims to sensitize melanoma cells to ionizing radiation (IR) and stimulate melanoma immunogenicity. The study employs a diverse set of experiments, including in vitro assays and in vivo evaluations, to assess the targeting effect, membrane fusion behavior, immunogenicity, and therapeutic efficacy of the liposomes. While promising, the study would benefit from improved clarity in experimental design, detailed rationale for the nanoformulation components, discussion of limitations, considerations for clinical translation, and graphical representation to enhance comprehension.

Experimental Design and Clarity: The study involves diverse experiments, from in vitro assays to in vivo evaluations. However, the text needs more clarity in presenting the overall rationale behind the experimental design and the logic of choosing specific assays.

Nanoformulation Complexity: The complex nanoformulation (Lip@AUR-ACP-aptPD-L1) is described, but the text could benefit from a more detailed explanation of the rationale for selecting each component and the potential synergistic effects.

Characterization Techniques: While the study utilizes various characterization techniques such as DLS, ICP, and fluorescence spectroscopy, it would be beneficial to discuss the limitations and potential sources of error associated with these techniques for a more rigorous interpretation.

In Vitro and In Vivo Experiments: The text briefly mentions in vitro and in vivo experiments, but it needs more in-depth discussion on the relevance of the chosen cell lines and animal models to human melanoma. More details on the rationale behind these choices would strengthen the study.

Toxicity Evaluation: The text mentions toxicity evaluation but does not provide an in-depth discussion of the potential clinical implications and safety considerations associated with the nanoformulation. This is crucial for translational research.

Limitations and Challenges: Include a thorough discussion of potential limitations and challenges associated with the experimental approach. Addressing these aspects will contribute to a more nuanced interpretation of the results.

Clinical Translation: Discuss the potential challenges and considerations for translating the findings into clinical applications. Address scalability, reproducibility, and safety aspects of the proposed strategy.

REVIEWERS' COMMENTS

Reviewer #1 (Remarks to the Author):

The revision is ready for publication.

A: Thank you for the approval of our revisions.

Reviewer #2 (Remarks to the Author):

This manuscript has been significantly improved after this round of revision and I also have satisfied with authors' responses on my concerns. Thus, I would like to recommend it for publication in Nature Communications without further revision.

A: Thank you for the approval of our revisions.

Reviewer #3 (Remarks to the Author):

The authors have added large amount of supplementary data to fully adress my concerns. I have not any other comments on the manuscript.

A: Thank you for the approval of our revisions.

Reviewer #4 (Remarks to the Author):

The authors have satisfactorily addressed all my comments.

A: Thank you for the approval of our revisions.

Reviewer #5 (Remarks to the Author):

The study presented by Ren X. et. al, investigates a novel approach for melanoma treatment using melanoma-targeted fusogenic liposomal nanoformulations integrated with Auranofin (AUR) and multivariate-gated aptamer assemblies. The complex nanoformulation aims to sensitize melanoma cells to ionizing radiation (IR) and stimulate melanoma immunogenicity. The study employs a diverse set of experiments, including in vitro assays and in vivo evaluations, to assess the targeting effect, membrane fusion behavior, immunogenicity, and therapeutic efficacy of the liposomes. While promising, the study would benefit from improved clarity in experimental design, detailed rationale for the nanoformulation components, discussion of limitations, considerations for clinical translation, and graphical representation to enhance comprehension.

A: Thank you for the appreciation of our work and the approval of our revisions. Based on your advice we have further improved the wording of this manuscript, which may enhance its clarity and informativity for the ease of readers. Meanwhile, the figure quality and organization have both been improved as requested. The detailed responses are shown here below point-by-point for your review.

Experimental Design and Clarity: The study involves diverse experiments, from in vitro assays to in vivo evaluations. However, the text needs more clarity in presenting the overall rationale behind the experimental design and the logic of choosing specific assays.

A: Thank you for your advice. We agree with the reviewer that expanding the discussions regarding the overarching experimental design and choice of specific assays could help the readers better understand the concepts of this study as well as enhance its reproducibility. Based on the suggestions you kindly provided, we have added substantial amount of additional descriptions and discussions to highlight the scientific relevance of the experimental designs and assays. Some representative examples are listed below:

“To provide an intuitive demonstration of the nanoscale morphology of the liposomal products, the Lip@AUR-ACP-aptPD-L1 samples were observed by transmission electron microscopic (TEM) imaging analysis and the results indicated that the liposomes have uniform spherical morphology and high monodispersity (Figure 2e). However, TEM imaging only showed the liposome morphology under dried conditions. To characterize the size distribution of the liposomes under biomimetic solution environment, the liposomes were further dispersed in biomimetic buffers for DLS analysis.”

“Here the AUR contents in the liposomal systems were measured using both ICP and fluorescence spectroscopic analysis. From a technical perspective, ICP has higher limit of detection (LOD) but superior interference control, while fluorescence spectroscopy has lower LOD but is more susceptible to background noises in complex samples. Therefore, the two techniques were combined to accurately profile the AUR loading in the liposomal system.”

“To test if the multivariate AND-gate operation of the ACP assembly was maintained after the integration into Lip@AUR-ACP-aptPD-L1, the liposomes were processed with different stimulus inputs and then the corresponding release efficiency of eCpG was monitored by DNA-PAGE and fluorescence spectroscopic analysis, which allowed the accurate profiling of the eCpG release behavior under different conditions.”

“To investigate the targeting ability of aptPD-L1 or eCpG in TME under clinically relevant conditions in vitro, we synthesized aptPD-L1 or eCpG with fluorescent 5'-FAM tags, thus allowing the accurate profiling of the cellular distribution of the aptamer species in the B16F10/splenocyte co-incubation system.”

“The fusion of Lip-ACP-aptPD-L1 with cytoplasmic membrane would trigger the anchoring of the ACP assemblies on tumor cell surface, which is crucial for enabling the AND-gate logic operation of ACP in IR-treated melanomas. To monitor the kinetic features of the membrane retention of the fusogenic liposomes, we systematically monitored the fluorescence distribution patterns of different Dil-labeled liposomes after incubation for 3/6/12/18 h in the co-culture system of B16F10 cells and mouse splenocytes.”

“To monitor the pharmacokinetic properties of the liposomal systems in vivo, AUR, Lip@AUR or Lip@AUR-aptPD-L1 were injected into C57BL/6J mice through the tail vein with three mice per group, then AUR content in tail venous blood of mice was detected by HPLC at scheduled time points, which could quantitatively determine the AUR levels with molecular precision and thus indicate the blood retention time of the injected samples.”

“Meanwhile, we also profiled the systemic distribution of the liposomes by measuring the AUR abundance in specific organs and tissues via ICP test in vivo using B16F10 tumor-bearing C57BL/6J mouse model, on account of the wide applicability of ICP analysis for tracking trace elements in complex samples.”

“Alternatively, various samples were injected into the B16F10 tumor-bearing C57BL/6J mice through intravenous route for 15 days of treatment, afterwards the slices of major organs were stained by hematoxylin and eosin (H&E) for histological inspections, which revealed that the liposomal samples did not induce obvious damage to major mouse organs regardless of the IR treatment conditions (Supplementary Figure 40).”

Nanoformulation Complexity: The complex nanoformulation (Lip@AUR-ACP-aptPD-L1) is described, but the text could benefit from a more detailed explanation of the rationale for selecting each component and the potential synergistic effects.

A: Thank you for the suggestion. Based on the advice we have significantly expanded the descriptions regarding the assembly of the ACP constructs and liposomal systems as well as the synergy therein, which may better illustrate the structural and functional features of the as-developed liposomal systems. The associated changes are shown here below for your review:

“In this work, we report a multivariate-gated aptamer assembly-modified AUR-loaded fusogenic liposome as an adjuvant for melanoma-targeted radio-immunotherapy. Based on a balanced consideration of clinical potency and translational impact, the commercially available CpG molecules are selected as the basic aptamer substrates, afterwards a 10-nucleotide long sequence with complementary binding capacity with the 5' end region of ATP-

binding aptamer (aptATP) was conjugated to the 5' end of CpG for molecular engineering (eCpG), thus endowing aptATP binding capability without compromising their TLR9 stimulating functions. Meanwhile, a peptide nucleic acid (PNA) sequence is synthesized with both MMP-2-mediated degradability and complementary binding affinity with the 3' end region of aptATP. Consequently, the PNA segment could bind to the aptATP-eCpG complex to form physiologically-stable duplex assemblies (ACP), which will also occupy the ATP binding domain of the aptATP and prevent premature eCpG detachment even in the presence of ATP competition. The preparation of ACP assemblies is a completely autonomous process driven by the self-regulated complementary binding of individual aptamer sequences, thus ensuring the quality of the obtained aptamer constructs with molecular precision via simple procedures. Notably, the 3' ends of aptATP and PD-L1-binding aptamer (aptPD-L1) are both modified with lipophilic cholesterol moieties, thus allowing their insertion into the lipid bilayers of DMPC-based fusogenic liposomes as well as their anchoring in cytoplasmic membranes. Meanwhile, the hydrophobic AUR molecules are dissolved into the lipid precursors for liposome preparation, eventually leading to the spontaneous formation of ACP-integrated fusogenic liposomes (Lip@AUR-ACP-aptPD-L1) through a simple film-hydration method with good cost-effectiveness and quality control. Taking advantage of the specific binding capability of aptPD-L1 to PD-L1, Lip@AUR-ACP-aptPD-L1 can bind with PD-L1-overexpressing melanoma cells and fuse with the cytoplasmic membrane, which not only anchors the ACP assemblies onto melanoma cell surface but also enables targeted AUR delivery. Owing to the PNA-mediated preoccupation of the ATP binding domains in aptATP, the ACP constructs operate as an AND-gate in the biological environment, which show negligible responsiveness to separate ATP or MMP2 stimulation and can only be activated when both ATP and MMP-2 are at a high level for triggering eCpG release. Interestingly, the liposome-mediated tumor-targeted AUR delivery substantially enhances the IR dose accumulation in melanoma cells during radiotherapy sessions and induces efficient immunogenic cell death (ICD), releasing abundant tumor-derived antigens and damage associated molecular patterns (DAMPs) such as ATP into TME while also inducing MMP-2 upregulation, thus creating the input signals required for triggering eCpG release from ACP constructs. Notably, MMP-2 can remove the PNA chain from the ACP assembly through biocatalytic degradation, while tumor-derived ATP can further trigger the detachment of eCpG through competitive binding with the liberated aptATP, leading to AND-gate eCpG release into TME to promote DC maturation through

binding to TLR9, which can substantially enhance DC-mediated cross-priming of antitumor T cells. In addition, AUR can also inhibit the ERK1/2-HIF-1 α -VEGF axis in tumor cells and impair the immunosuppression orchestrated by tumor-infiltrating immunosuppressive cells such as MDSCs and Tregs for boosting the antitumor function of activated T cells. These effects can act in a cooperative manner to substantially abolish melanoma growth and establish robust antitumor immune memory to prevent melanoma metastasis or recurrence (Figure 1).”

Characterization Techniques: While the study utilizes various characterization techniques such as DLS, ICP, and fluorescence spectroscopy, it would be beneficial to discuss the limitations and potential sources of error associated with these techniques for a more rigorous interpretation.

A: Thank you for your suggestion. Based on your advice we have expanded the discussions for key characterization techniques regarding their advantages and disadvantages. The changes are shown here below for your review:

The related discussions are shown here below for your review:

“Although DNA-PAGE analysis could intuitively illustrate the changes in aptATP/eCpG binding state, it is unable to provide quantitative data for objective analysis. Hence, we also carried out quantitative fluorescence analysis for the samples, revealing that aptATP has a complexation efficiency of 97.07% with eCpG (Supplementary Figure 4c), ascribing to the molecular specificity of the complementary sequences thereof.”

“To provide an intuitive demonstration of the nanoscale morphology of the liposomal products, the Lip@AUR-ACP-aptPD-L1 samples were observed by transmission electron microscopic (TEM) imaging analysis and the results indicated that the liposomes have uniform spherical morphology and high monodispersity (Figure 2e). However, TEM imaging only showed the liposome morphology under dried conditions. To characterize the size distribution of the liposomes under biomimetic solution environment, the liposomes were further dispersed in biomimetic buffers for DLS analysis.”

“Here the AUR contents in the liposomal systems were measured using both ICP and fluorescence spectroscopic analysis. From a technical perspective, ICP has higher limit of detection (LOD) but superior interference control, while fluorescence spectroscopy has lower LOD but is more susceptible to background noises in complex samples. Therefore, the two techniques were combined to accurately profile the AUR loading in the liposomal system.”

“To monitor the pharmacokinetic properties of the liposomal systems in vivo, AUR, Lip@AUR or Lip@AUR-aptPD-L1 were injected into C57BL/6J mice through the tail vein with three mice per group, then AUR content in tail venous blood of mice was detected by HPLC at scheduled time points, which could quantitatively determine the AUR levels with molecular precision and thus indicate the blood retention time of the injected samples.”

In Vitro and In Vivo Experiments: The text briefly mentions in vitro and in vivo experiments, but it needs more in-depth discussion on the relevance of the chosen cell lines and animal models to human melanoma. More details on the rationale behind these choices would strengthen the study.

A: Thank you for your concern. We agree that elucidating the clinical relevance of the chosen cell lines and animal models could strengthen the translational value of this study. Specifically, there is abundant evidence that B16F10 cell, a murine melanoma cell line, presents strong similarities with human melanoma cells in terms of upregulated markers including PD-L1, VEGF and HIF-1 α (Cancer Cell. 2022, 40(11), 1324-1340.e8.), hyperglycolysis metabolism trait (Nat Commun. 2023, 14(1), 5333.) and pathological traits including high invasiveness and metastasis risk (Nat Commun. 2017, 8(1), 1319.; Cancer Cell. 2011, 20(1), 104-118.). Therefore, B16F10 cells and B16F10-based murine melanoma models are commonly used in preclinical research regarding melanoma pathology and treatment, on account of their biological resemblance with human melanomas in a clinical setting. The insights above justify our choices of cell lines and animal models in the present study. The associated discussions are also incorporated into the manuscript for your review:

“Concrete evidence confirms that B16F10 cells have strong similarities with human melanoma cells in terms of upregulated PD-L1, VEGF and HIF-1 α expression⁵⁸, hyperglycolysis metabolism trait⁵⁹ and pathological traits including high invasiveness and metastasis risk^{60, 61}, which is a widely used model system in melanoma research.”

“The in vivo therapeutic evaluation of the liposomal systems was further carried out on B16F10 melanoma-bearing C57BL/6J mice, which have marked resemblance with melanomas on real life patients in terms of pathologic, metabolic and immunological traits⁸².”

Toxicity Evaluation: The text mentions toxicity evaluation but does not provide an in-depth discussion of the potential clinical implications and safety considerations associated with the nanoformulation. This is crucial for translational research.

A: Thank you for your advice. We would like to apologize for the oversimplified discussions regarding the pharmacokinetic and toxicity properties of the liposomal systems and have substantially expanded the associated contents to highlight their translational potential. Specifically, we found that the blood circulation time of Lip@AUR-aptPD-L1 liposomes has increased by 5-fold than AUR and reached 8 h (Supplementary Figure 29). The liposome-enhanced blood retention capacity of the therapeutic contents is particularly important for their subsequent radio-immunotherapeutic activities, which may facilitate the interaction of the liposomes with melanoma tissues. Alternatively, the systemic distribution of the liposomes was determined by profiling the AUR content in major organs or tissues via ICP, and the results revealed that Lip@AUR-aptPD-L1 was predominantly enriched in melanoma site at 6 hours after administration while Lip@AUR was mainly detected in kidneys (Supplementary Figure 30), validating our hypothesis that the incorporation of aptPD-L1 ligands enables efficient and guided delivery of the liposomes to melanomas after systemic administration.

In addition, we have also expanded the discussions regarding the in vivo toxicity profiles of the liposomes. Notably, the mice in all groups showed no obvious body weight loss after various treatment, which not only

confirmed the non-toxicity of the liposomes and low-dose IR but also indicated that the liposome-enhanced radio-immunotherapy induced no serious systemic adverse immune reactions. Meanwhile, H&E staining-based histological inspections of major organs after various treatment showed that no obvious tissue damage occurred after liposome treatment regardless of the IR irritation status (Supplementary Figure 40). The histocompatibility of the liposome-enhanced radio-immunotherapy was manifold. On one hand, the melanoma-targeting effect of the liposomes and low IR dose could limit the collateral damage to non-specific organs and tissues. On the other hand, the multivariate-gated operation of the ACP assembly in TME could further improve the spatial-temporal controllability of the eCpG-dependent immunostimulatory effect and reduce the risk of systemic adverse immune responses. These above results suggested that Lip@AUR-ACP-aptPD-L1 liposomes could be a safe and effective option for melanoma radioimmunotherapy.

The related discussions are shown here below for your review:

“To monitor the pharmacokinetic properties of the liposomal systems in vivo, AUR, Lip@AUR or Lip@AUR-aptPD-L1 were injected into C57BL/6J mice through the tail vein with three mice per group, then AUR content in tail venous blood of mice was detected by HPLC at scheduled time points, which could quantitatively determine the AUR levels with molecular precision and thus indicate the blood retention time of the injected samples. The Lip@AUR-aptPD-L1 liposomes showed significantly longer blood circulation time compared with AUR, of which the blood half-life has increased by 5-fold and reached around 8 h (Supplementary Figure 29). The liposome-enhanced blood retention capacity of the therapeutic contents is particularly important for their subsequent radio-immunotherapeutic activities, which may facilitate the interaction of the liposomes with melanoma tissues.”

“Meanwhile, we also profiled the systemic distribution of the liposomes by measuring the AUR abundance in specific organs and tissues via ICP test in vivo using B16F10 tumor-bearing C57BL/6J mouse model, on account of the wide applicability of ICP analysis for tracking trace elements in complex samples. The comparative analysis of AUR deposition patterns immediately suggested that Lip@AUR-aptPD-L1

liposomes predominantly accumulated in the B16F10 tumors with a relative ratio of around 46% after 24 h of administration (Supplementary Figure 30). In contrast, non-targeting Lip@AUR liposomes were mostly detected in mouse kidney, attributing to the liposome clearance capacity of the mononuclear phagocyte system (MPS) therein. The observations validated our hypothesis that the incorporation of aptPD-L1 ligands enables efficient and guided delivery of the liposomes to melanomas after systemic administration.”

“Notably, mice receiving combinational liposome+IR treatment showed no significant weight loss compared to the PBS-only control group, which not only confirmed the non-toxicity of the liposomes and low-dose IR but also indicated that the liposome-enhanced radio-immunotherapy induced no serious systemic adverse immune reactions (Supplementary Figure 39). Alternatively, various samples were injected into the B16F10 tumor-bearing C57BL/6J mice through intravenous route for 15 days of treatment, afterwards the slices of major organs were stained by hematoxylin and eosin (H&E) for histological inspections, which revealed that the liposomal samples did not induce obvious damage to major mouse organs regardless of the IR treatment conditions (Supplementary Figure 40). The histocompatibility of the liposome-enhanced radio-immunotherapy was manifold. On one hand, the melanoma-targeting effect of the liposomes and low IR dose could limit the collateral damage to non-specific organs and tissues. On the other hand, the multivariate-gated operation of the ACP assembly in TME could further improve the spatial-temporal controllability of the eCpG-dependent immunostimulatory effect and reduce the risk of systemic adverse immune responses. These above results suggested that Lip@AUR-ACP-aptPD-L1 liposomes could be a safe and effective option for melanoma radioimmunotherapy.”

Limitations and Challenges: Include a thorough discussion of potential limitations and challenges associated with the experimental approach. Addressing these aspects will contribute to a more nuanced interpretation of the results.

A: Thank you for your advice. We agree with the reviewer that a thorough discussion of the limitations and challenges of this study could provide a more objective assessment on the translational potential of this

study and have made corrections as requested. Firstly, although we have demonstrated that Lip@AUR-ACP-aptPD-L1 liposomes can be anchored onto melanoma cell membranes for sufficiently long time to potentiate enhanced radio-immunotherapeutic potency, it is worth mentioning the delivery of the liposomes on real-life patients is a highly complex and dynamic process, and rigorous investigations of the bio-nanointeractions of the liposomes under clinically relevant conditions could substantially facilitate the optimization of treatment schedule. Secondly, it is important to note that phenotypic heterogeneity is a hallmark of solid tumors such as melanoma. Despite the potent melanoma inhibition efficacy of the Lip@AUR-ACP-aptPD-L1-enhanced radio-immunotherapy in vitro and in vivo, its efficacy against those minor melanoma cell populations with intrinsically low PD-L1 expression and the potential mechanisms are worth investigation in follow-up research.

The related discussions are shown here below for your review:

“Our study reports that Lip@AUR-ACP-aptPD-L1 could substantially enhance the radio-immunotherapeutic efficacy against melanomas. However, more studies on melanoma samples from real-life patients are necessary to understand its pharmacological characteristics and general applicability, which will be the subjects of future research. Firstly, although we have demonstrated that Lip@AUR-ACP-aptPD-L1 liposomes can be anchored onto melanoma cell membranes for sufficiently long time to potentiate enhanced radio-immunotherapeutic potency, it is worth mentioning the delivery of the liposomes on real-life patients is a highly complex and dynamic process, and rigorous investigations of the bio-nanointeractions of the liposomes under clinically relevant conditions could substantially facilitate the optimization of treatment schedule. Secondly, it is important to note that phenotypic heterogeneity is a hallmark of solid tumors such as melanoma. Despite the potent melanoma inhibition efficacy of the Lip@AUR-ACP-aptPD-L1-enhanced radio-immunotherapy in vitro and in vivo, its efficacy against those minor melanoma cell populations with intrinsically low PD-L1 expression and the potential mechanisms are worth investigation in follow-up research.”

Clinical Translation: Discuss the potential challenges and considerations for translating the findings into clinical applications. Address scalability, reproducibility, and safety aspects of the proposed strategy.

A: Thank you for the advice. Based on your suggestions we have added extra discussions in the Discussion section to address key challenges and considerations for the translation of the liposome-enhanced radio-immunotherapy against melanomas, which are shown here below for your review:

“The variations in the cell membrane dynamics, pharmacokinetic characteristics and IR treatment parameters are major considerations regarding the potential clinical translation of the proposed Lip@AUR-ACP-aptPD-L1 systems for radio-immunotherapy against melanomas. Firstly, the fusion between the liposomes and melanoma cell membrane and the subsequent anchoring of ACP assemblies in real life patients are profoundly affected by the molecular dynamics of cell membranes such as lipid composition, symmetry and cytoskeleton organization, which could affect the scalability of the proposed therapeutic strategy and thus requires the personalized optimization of the liposome composition and treatment schedule to prolong the membrane retention of the anchored ACP constructs. Meanwhile, although the ACP constructs are assembled through the autonomous complexation between individual components, the potential competition from endogenous biomolecules remains to be investigated that may attenuate its tumor-specific biochemical reprogramming and TME remodeling efficacy, which could be solved through the rational optimization of the inter-component binding segments. Finally, the IR dose in this study was significantly lower than the ablative dose in the clinical setting, which may require further optimization for achieving balanced immunostimulatory effect, tumor cell damage and safety on melanoma patients. Addressing these considerations may substantially enhance the clinical relevance of the radio-immunotherapeutic strategy in the present study.”